# Targeting p53 and histone methyltransferases restores exhausted CD8+ T cells in HCV infection

Valeria Barili[1,2], Paola Fisicaro[2], Barbara Montanini [3,4], Greta Acerbi[1,2], Anita Filippi[2], Giovanna Forleo[2], Chiara Romualdi[5], Manuela Ferracin [6], Francesca Guerrieri [7], Giuseppe Pedrazzi [8], Carolina Boni[2], Marzia Rossi[1,2], Andrea Vecchi[1,2], Amalia Penna[2], Alessandra Zecca[2], Cristina Mori[2], Alessandra Orlandini[2], Elisa Negri[2], Marco Pesci[1,2], Marco Massari [9], Gabriele Missale[1,2], Massimo Levrero[7,10,11], Simone Ottonello[3,4] & Carlo Ferrari[1,2]*

Hepatitis C virus infection (HCV) represents a unique model to characterize, from early to late stages of infection, the T cell differentiation process leading to exhaustion of human CD8+ T cells. Here we show that in early HCV infection, exhaustion-committed virus-specific CD8+ T cells display a marked upregulation of transcription associated with impaired glycolytic and mitochondrial functions, that are linked to enhanced ataxia-telangiectasia mutated (ATM) and p53 signaling. After evolution to chronic infection, exhaustion of HCV-specific T cell responses is instead characterized by a broad gene downregulation associated with a wide metabolic and anti-viral function impairment, which can be rescued by histone methyltransferase inhibitors. These results have implications not only for treatment of HCV-positive patients not responding to last-generation antivirals, but also for other chronic pathologies associated with T cell dysfunction, including cancer.

[1] Department of Medicine and Surgery, University of Parma, Parma, Italy. [2] Unit of Infectious Diseases and Hepatology, Laboratory of Viral Immunopathology, Azienda Ospedaliero–Universitaria of Parma, Parma, Italy. [3] Biomolecular, Genomic and Biocomputational Sciences Unit, Department of Chemistry, Life Sciences and Environmental Sustainability, University of Parma, Parma, Italy. [4] Biopharmanet-Tec, University of Parma, Parma, Italy. [5] Department of Biology, University of Padova, Padova, Italy. [6] Department of Experimental, Diagnostic and Specialty Medicine—DIMES, University of Bologna, Bologna, Italy. [7] Cancer Research Center of Lyon (CRCL)-INSERM U1052, Lyon, France. [8] Unit of Neuroscience, Department of Medicine and Surgery, Robust Statistics Academy (Ro.S.A.), University of Parma, Parma, Italy. [9] Unit of Infectious Diseases, IRCCS–Azienda Ospedaliera S. Maria Nuova, Reggio Emilia, Italy. [10] Université Claude Bernard Lyon 1, Service d'Hepatologie et Gastroenterologie Hopital de la Croix-Rousse, Hospices Civils de Lyon, Lyon, France. [11] Center for Life Nano Science, Istituto Italiano di Tecnologia, Rome, Italy. *email: carlo.ferrari@unipr.it

Hepatitis C virus (HCV) infection can progress to resolution in around 20–40% of the cases; conversely, in 60–80% of infected patients the virus persists and the infection becomes chronic. Therefore, patients with different disease outcomes can be identified at early stages of infection making hepatitis C an ideal model to study the process of differentiation from early exhaustion-committed to late fully exhausted CD8+ T cells in humans. Spontaneous viral clearance is associated with efficient CD8+ T cell responses, whereas cytotoxic T cells are functionally impaired in chronically infected patients[1,2]. In chronic viral infections, including hepatitis C, persistent antigen stimulation is believed to represent a major determinant of CD8+ T cell dysfunction, which is characterized by upregulation of multiple inhibitory receptors, repressive transcriptional reprogramming, broad metabolic alterations and defective T cell effector function and memory development[3–9].

To improve exhausted T cell functions, regulatory pathway manipulation has been attempted in various experimental settings, but a full restoration has not been achieved so far[5,10–12]. It is thus becoming increasingly clear that T cell exhaustion results from a broader cellular alteration with an extensive involvement of crucial T cell functions, including metabolism and epigenetic regulation[13–16]. Indeed, the role of metabolic reprogramming in T cell differentiation is now well appreciated[6,15,17]. In particular, differentiation of naïve into effector T cells is accompanied by a marked upregulation of aerobic glycolysis, to meet the increased energetic demands associated with this transition[18–20]. Following successful control of infection, glycolysis is turned down, and mitochondrial fatty acid oxidation (FAO) becomes the major provider of metabolic energy to support memory CD8+ T cell generation, survival, and antigen recall responsiveness[21,22].

Severe defects in HBV-specific CD8+ T cell mitochondrial function (e.g., impaired FAO, ROS overproduction and mitochondrial depolarization) and the positive effect of their correction on CD8+ T cell exhaustion have recently been reported in chronic HBV infection[23,24]. In the LCMV mouse model of chronic infection, glycolysis downregulation and OXPHOS impairment are hallmarks of exhaustion and are more evident in the PD1[hi] than in the PD1[int] subset of exhausted CD8+ T cells[25]. A dysregulated gene expression profile consistent with these metabolic changes has recently been described in human HCV-specific CD8+ T cells committed to exhaustion[26], but a functional assessment of the actual contribution of metabolic and epigenetic state alteration to CD8+ T cell dysfunction and its causal association with exhaustion is still lacking[27].

To further delineate and functionally validate key dysregulated pathways associated with the development of exhausted T cell responses, we applied genome-wide transcriptome profiling, coupled with multiple functional validation assays and targeted rescue strategies, to virus-specific CD8+ T cells from chronically evolving or resolved HCV infections. We show, here, that differentiation from early exhaustion-committed to late fully exhausted CD8+ T cells is marked by a transition from a predominantly upregulated early gene expression profile to a massive late downregulation with global repression of core cellular processes and an extensive histone methylation-dependent repressive chromatin remodeling. Targeting dysfunctional signaling, metabolic and chromatin remodeling pathways can efficiently improve CD8+ T cell metabolism and antiviral function, and may thus represent a novel rational immunomodulatory strategy, perhaps complementary to PD-1 or other checkpoint inhibitor blockade, to correct T cell exhaustion. This may positively impact HCV infection, especially in 'Direct Acting Antivirals' (DAA) non-responder patients who still require effective therapies, but also other chronic diseases, including cancer, displaying T cell dysfunction caused by persistent CD8+ T cell stimulation.

## Results

**Early dysregulated metabolic and signaling pathways in HCV-specific CD8+ T cells.** We characterized CD8+ T cell gene expression profiles associated with either spontaneous HCV control and T cell memory generation or virus persistence and T cell exhaustion. HCV-specific CD8+ T cells were collected and sorted (Supplementary Fig. 1) from patients with T1/early self-limited or chronically evolving acute HCV infection immediately after clinical presentation (HCV RNA-positive patients, within one month from the ALT peak; T1/early time-point) and at later stages of resolution (HCV RNA-negative patients, at least 12 months after the ALT peak; T2/late time-point) or chronic evolution of infection (HCV RNA-positive patients, followed for at least one year after the first detection of ALT elevation; T2/late time-point). Patients were thus subdivided into four groups, designated as T1/early self-limited ($n = 5$), T1/early chronic ($n = 8$), T2/late resolved (late resolution phase of T1 self-limited patients; $n = 4$) and T2/late-chronic ($n = 7$).

Principal-component analysis of ANOVA-filtered expression data (Fig. 1a) revealed a significant overlap between the two groups at the T1/early-stage time-point and a substantial separation between T2 late-stage patients belonging to either the chronic (i.e., T cell-exhausted) or the fully resolved groups. This was confirmed by the statistical clustering of gene expression data shown in Fig. 1b, where the dysfunctional profiles of early and late exhausted CD8+ T cells appear to be distinct from that of FLU-specific CD8+ T cells from healthy donors, which served as fully functional memory T cell controls. Altogether, these results suggest a progressive, outcome-dependent accumulation of divergent gene expression signatures starting from more comparable acute phase transcriptional profiles.

A topology-based analysis, which relies not only on the assignment of dysregulated genes to a given pathway but also on gene connections derived from pathway annotation, was then applied to the T1 and T2 datasets. Twenty-nine dysregulated pathways (Benjamini-Hochberg corrected $q$-value $\leq 0.05$) were identified by this analysis in T1/early HCV-specific CD8+ T cells from chronically evolving compared to self-limited acute patients (Fig. 1c and Supplementary Data 1). By contrast, 277 dysregulated pathways were retrieved by the same analysis from the T2/late comparison of HCV-specific exhausted and memory CD8+ T cells from chronic and resolved patients, respectively. Fifteen of these pathways were shared by the T1 and the T2 time-points (Fig. 1c). Most of them (14 out of 15) were upregulated at the T1/early-stage of chronically evolving infection but downregulated (13 out of 15) at the T2/late-stage time-point (Fig. 1c). The 15 shared dysregulated pathways (Fig. 1c) include intracellular signaling downstream to TCR activation, NF-kB/TNF activation, DNA damage response and mitochondrion (MT)-related metabolic functions as the most represented cellular processes (see Supplementary Data 1). Half of the 14 T1/early-dysregulated pathways, which were not shared with the T2/late time-point, were downregulated. These comprised genes related to glucose transport (GLUT1 and GLUT4, also known as SLC2A1 and SLC2A4) and cytokine signaling, including the T cell-specific, glycolysis-regulatory CD28 signaling pathway (Fig. 1d). Other genes related to glucose homeostasis, including genes coding for gluconeogenesis enzymes known to be involved in the production of glycolytic intermediates under low glucose conditions[28] (see Supplementary Data 1), also emerged as enriched from topology-based analysis.

The entire T1 dataset was then subjected to unsupervised gene-set enrichment analysis (GSEA), which allows to reveal sets of differentially regulated, functionally related genes, rather than individual high-scoring genes, above an arbitrarily set cut-off. As shown in Fig. 1e and Supplementary Fig. 2 (see also the T1 sheets in Supplementary Data 2), all dysregulated pathways and gene

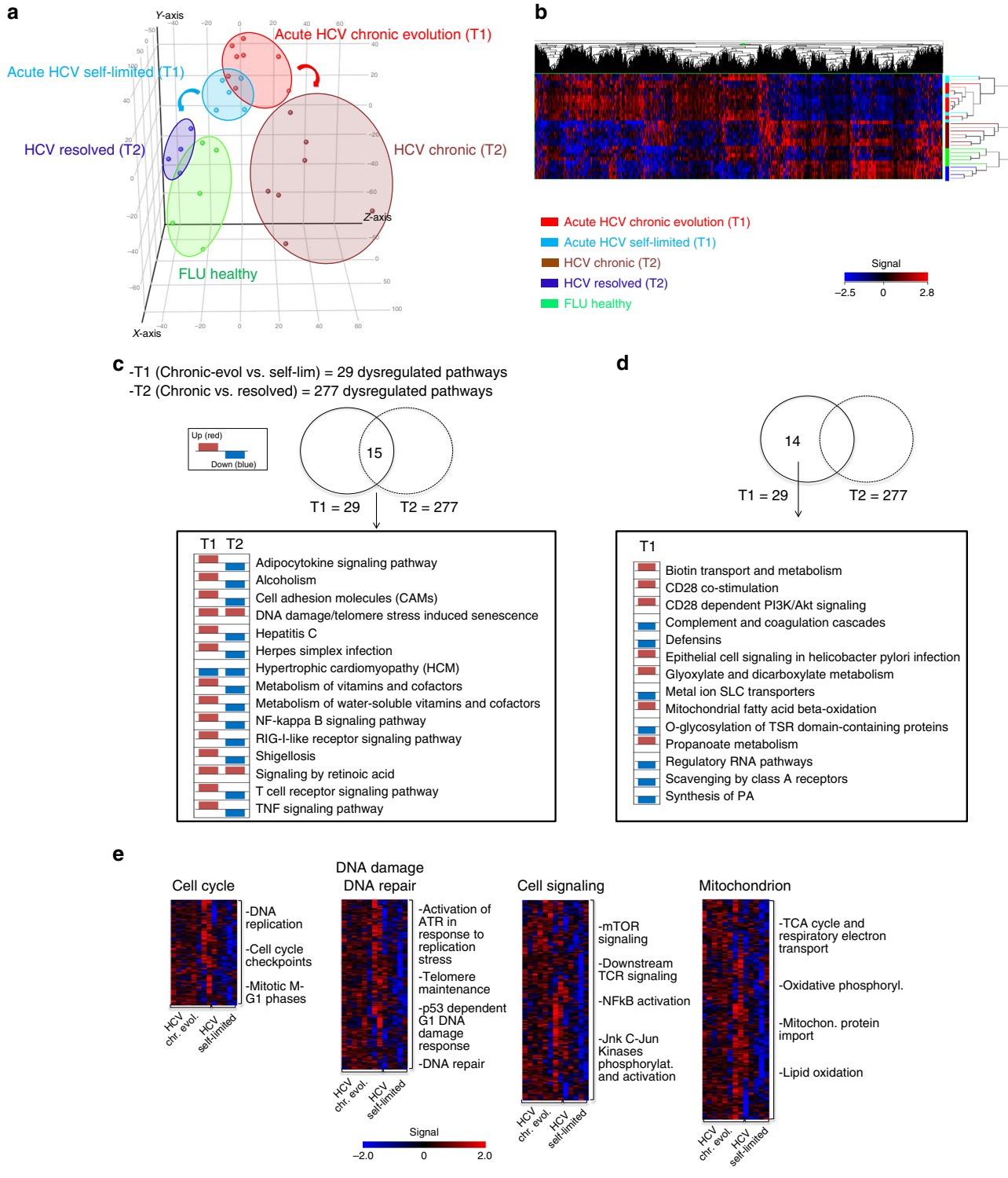

ontology categories identified by GSEA were upregulated in HCV-specific CD8+ T cells from T1/early chronic patients. In particular, GSEA confirmed a strong upregulation of multiple DNA damage response, intracellular signaling and cell-cycle control genes, including immune-checkpoint inhibitors, such as PD-1 (also known as PDCD1), CD160, CTLA4 and TIGIT (see the T1 sheets in Supplementary Data 2). Notably, GSEA also confirmed the upregulation of a large set of genes coding for mitochondrial oxidative phosphorylation (OXPHOS) and electron transport, reactive oxygen species (ROS) detoxification and fatty acid oxidation (FAO) components (see Fig. 1e and

Supplementary Data 2). Further analysis of GSEA data using the KEGG Reactome and GO databases allowed to group T1 upregulated genes into eight distinct functional categories matching the above mentioned dysregulated processes (Supplementary Fig. 2).

A generalized upregulation thus appears to be the predominant but not exclusive feature of acute-phase dysregulation.

**Impaired CD8+ T cell glucose and mitochondrial metabolism in early infection.** To gain insight into the relationship between

**Fig. 1 Gene-expression profiling of virus-specific CD8+ T cells in HCV infection. a** Principal-component analysis (PCA) of 4766 differentially expressed genes (DEGs) identified by ANOVA ($q$-value ≤ 0.05) in HCV-specific CD8+ T cells from patients with acute ($n = 13$), chronic ($n = 7$) and resolved ($n = 4$) HCV infections, as well as FLU-specific CD8+ T cells from healthy controls ($n = 5$). Data were normalized with the quantile method and filtered for probes detected in at least two-third of replicates for each condition. **b** Hierarchical-clustering representation of the 4766 DEGs. Data were median-normalized before clustering and expressed as single patient profiling. In red upregulated and in blue downregulated genes. **c** Transcriptome profiles of HCV-specific CD8+ T cells from chronically evolving and self-limited acute patients were compared by topological analysis at two different time-points (time of diagnosis/T1 and several months later/T2). Twenty-nine and 277 pathways were significantly dysregulated (Benjamini-Hochberg corrected $q$-value ≤ 0.05) at the T1/early and T2/late time-points, respectively. Venn-diagram distribution of pathways identified as dysregulated by comparative topological analysis of acute chronically-evolving vs. self-limited and chronic vs. resolved patients. The 15 pathways found to be significantly dysregulated in both comparisons, but with a largely predominant trend toward upregulation (red) at T1/early and the opposite trend (blue) at T2/late, are listed in the bottom panel; these include genes related to TCR signaling, DNA damage response and metabolism at the T1 comparison (each column shows the enrichment in upregulated or downregulated genes in each pathway derived from the calculation of the median gene expression fold change in the comparison of chronically evolving vs. self-limited–T1/early–and of late chronic vs. spontaneously resolved patients–T2/late). **d** List of the 14 T1/early-specific dysregulated pathways, half of which are upregulated (red), while the remaining half is downregulated (blue). **e** Heat-map of differentially expressed genes derived from GSEA (Molecular Signature Database, C2 canonical pathways and C5 gene ontology sets) at T1/early time-point, related to cell cycle, DNA damage/DNA repair, cell signaling, mitochondrion and metabolism. Upregulated genes in red; downregulated genes in blue.

the dysregulation of metabolism-related genes and progression toward the exhaustion phenotype, GSEA analysis was applied using the molecular signature identified in the murine model of chronic LCMV infection[25], obtained by merging the leading-edge genes belonging to the different metabolic KEGG pathways. As shown in Supplementary Fig. 3a, a significant overlap was observed between metabolism-related genes that appear to be dysregulated in early exhausted LCMV-specific lymphocytes and in T1/early exhausted HCV-specific CD8+ T cells. Importantly, in the case of the 'glycolysis and gluconeogenesis' and 'oxidative phosphorylation' pathways, and of the 'glucose deprivation signature'[25,29], the correlation of the two datasets was confirmed also when all genes (not only those restricted to the leading-edge set) were considered (Supplementary Fig. 3b).

Particularly evident was a parallel enrichment in upregulated genes related to mitochondrial metabolism and a concomitant enrichment in downregulated, glycolysis-related genes at the T1 time-point, both of which were significantly correlated with the relative expression levels of the same sets of genes in the LCMV model of chronic infection (Pearson $p$-values < 0.05). Based on these findings and on similar data reported for other models of lymphocyte exhaustion[29], we hypothesized that HCV-specific CD8+ T cells from chronically evolving T1/early patients are in shortage of glucose, which is required to meet the energy and biosynthetic demands of T cell expansion.

To functionally verify T1/early metabolic gene dysregulation, we initially focused on glucose metabolism and measured glucose import, the first limiting step of glycolysis, as well as glucose transporter 1 (GLUT1) protein levels. As shown in Fig. 2a–c, glucose uptake and GLUT1 levels were both significantly reduced in T1/early HCV-specific CD8+ T cells from chronically evolving compared to self-limited patients and FLU-specific CD8+ controls. In line with these findings, also glucose consumption was diminished in T1/early chronic CD8+ T cells compared to control CD8+ T cells (Supplementary Fig. 4a).

The basal extracellular acidification rate (ECAR), a quantitative indicator of glycolytic activity, as well as the maximum glycolytic capacity and glycolytic reserve were also reduced in HCV peptide-stimulated CD8+ T cells from chronically evolving T1/early patients compared to control peptide (FLU, CMV and EBV)-stimulated cells (see Fig. 2d and the cell energy phenotype plots in Supplementary Fig. 4b). Conversely, NS3-HCV peptide stimulated CD8+ T cells from T1/early acute self-limited patients displayed higher ECAR values, both at basal and maximal respiration levels, compared to CD8+ T cells derived from T1/early chronically-evolving patients (see Fig. 2e and the cell energy phenotype plots in Supplementary Fig. 4c).

Interestingly, in the same set of cells, expression levels of the PD-1 co-inhibitor were inversely related to glucose uptake capacity and were maximal in CD8+ T cells from T1/early chronically evolving patients, thus suggesting a possible link between PD-1 upregulation and altered glucose metabolism (Supplementary Fig. 4d).

HCV-specific CD8+ T cells from T1/early chronic patients also displayed an impaired mitochondrial function. In fact, a markedly reduced mitochondrial membrane potential (Fig. 3a) and abnormally elevated mitochondrial ROS levels (Fig. 3b, c) were detected in CD8+ T cells from chronically evolving patients. Further confirming a depressed mitochondrial respiratory capacity, a significantly lower basal oxygen consumption rate (OCR) and similarly reduced maximal and spare respiratory capacities were measured in purified HCV peptide-stimulated CD8+ T cells from chronically evolving patients (see Fig. 3d and the cell energy phenotype plots in Supplementary Fig. 4b). Again, NS3-HCV peptide stimulated CD8+ T cells from T1/early acute self-limited patients displayed higher OCR values, both at basal and maximal respiration levels, compared to CD8+ T cells from T1/early chronically-evolving patients (see Fig. 3e and the cell energy phenotype plots in Supplementary Fig. 4c).

Evidence that this difference in glycolytic and mitochondrial efficiency was not merely due to a lack of T cell activation was provided by the increased cytokine production detected in HCV peptide-stimulated compared to unstimulated cells (Supplementary Fig. 4e). Multiple lines of evidence also indicate that the observed metabolic differences cannot be simply ascribed to different frequencies of virus-specific CD8+ T cells within the total CD8+ population or to an insufficient sensitivity of the Seahorse analysis at the low HCV-specific CD8+ T cell frequencies commonly found in HCV infected patients. In fact, a similar difference in metabolic profiles between early self-limited and chronically evolving acute infections was observed when Seahorse analysis was performed in patients with different disease outcomes but comparable numbers of IFN-γ producing CD8+ T cells following HCV peptide stimulation (e.g., patient # 1 vs. # 7 in Supplementary Fig. 5). Moreover, when similar frequencies of IFN-γ producing CD8+ T cells were detected after overnight stimulation of individual patients lymphocyte samples with control and HCV-specific peptides, a better metabolic performance was again detected in control compared to HCV peptide-stimulated cells (e.g., patient #1 in Supplementary Fig. 5). Finally, when tested with graded frequencies of FLU matrix 58-66-specific CD8+ T cells, Seahorse sensitivity appeared to be sufficient to detect metabolic differences down to frequencies lower than 0.5% FLU-specific T cells with respect to total CD8+

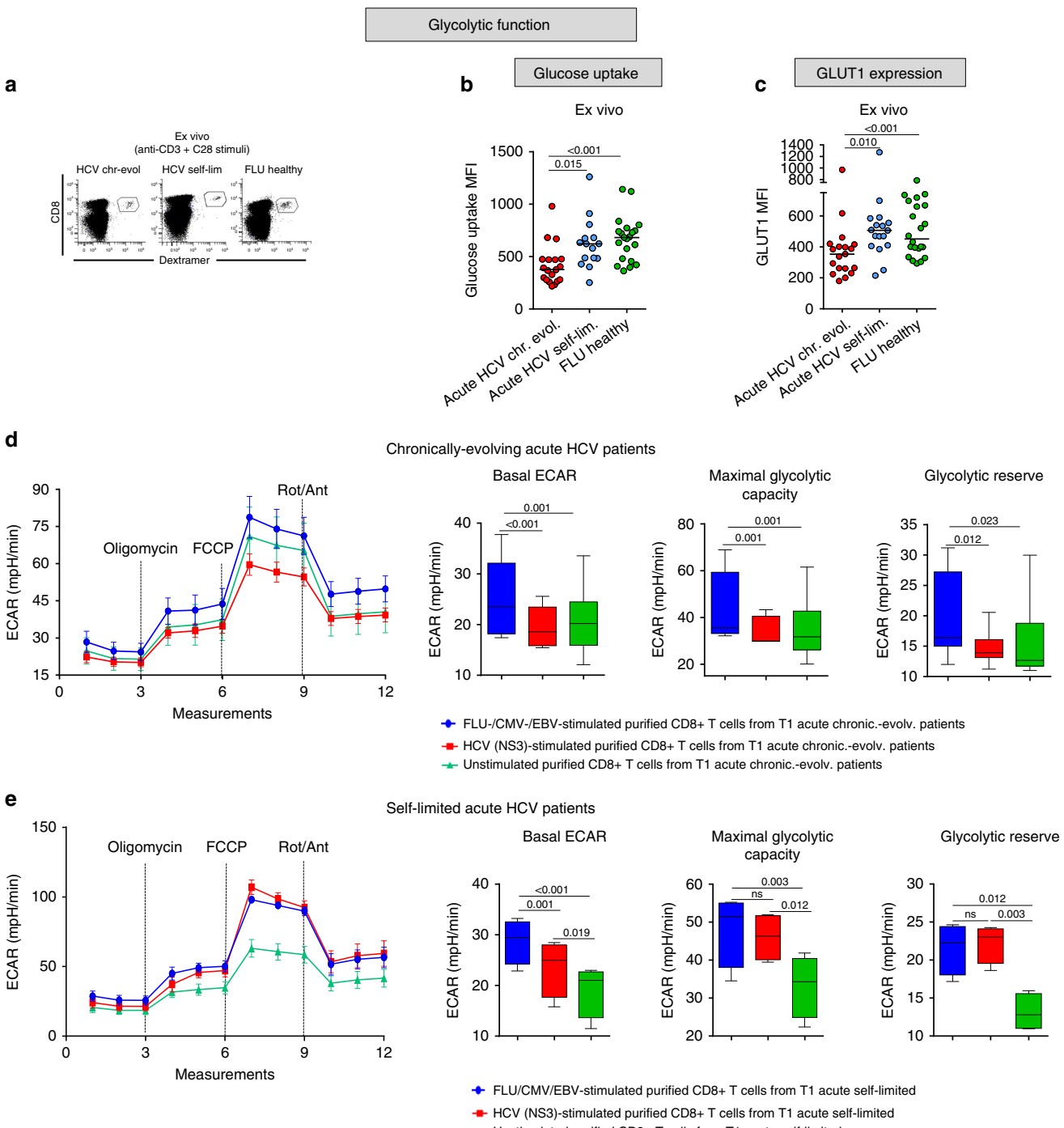

**Fig. 2 Glucose metabolism is impaired in HCV-specific CD8+ T cells from chronically evolving acute patients. a** Representative examples of virus-specific CD8+ T cells stained with HLA-A2+ dextramers ex vivo after overnight anti-CD3/anti-CD28 stimulation. Glucose uptake (**b**), measured by the incorporation of the glucose analog 2-NBDG (MFI), and Glut1 expression levels (**c**) in virus-specific CD8+ T cells from T1/early HCV patients and healthy controls stimulated as in **a**. Data are presented as median fluorescence intensity (MFI) values; median values are indicated by horizontal lines. Different numbers of patients (represented by individual dots) were tested in each assay depending on dextramer-positive cell frequencies. **b**–**c** Differences between multiple groups were evaluated with the non-parametric Kruskal-Wallis test; p-values were corrected for pairwise multiple comparisons with the Dunn's test. **d** metabolic flux profiling of purified CD8+ T cells from 6 T1/early chronically-evolving (acute) patients. Cells were stimulated overnight with either HCV-NS3 (red) or control (FLU-specific, CMV-specific and EBV-specific) peptides (blue), or were not stimulated (green). The extracellular acidification rate (ECAR) was measured in real-time ex vivo before (basal level) and after oligomycin treatment in order to determine the maximum glycolytic capacity (MGC) and glycolytic reserve (difference between MGC and baseline ECAR) (see Methods section for details on Seahorse analysis). **e** Metabolic flux profiling of purified CD8+ T cells from T1/early self-limited (acute) patients ($n = 4$) stimulated overnight as in **d**. ECAR, maximum glycolytic capacity and glycolytic reserve were measured as in **d**. In **d** and **e**, ECAR values are given as the mean ± SD in the left-side and are presented as box-and-whisker plots (with median and 5–95 percentile) in the right-side. **d**–**e** Statistical analysis was performed with the Friedman test to compare different stimuli; p-values have been corrected for pair-wise multiple comparisons with the Conover's test.

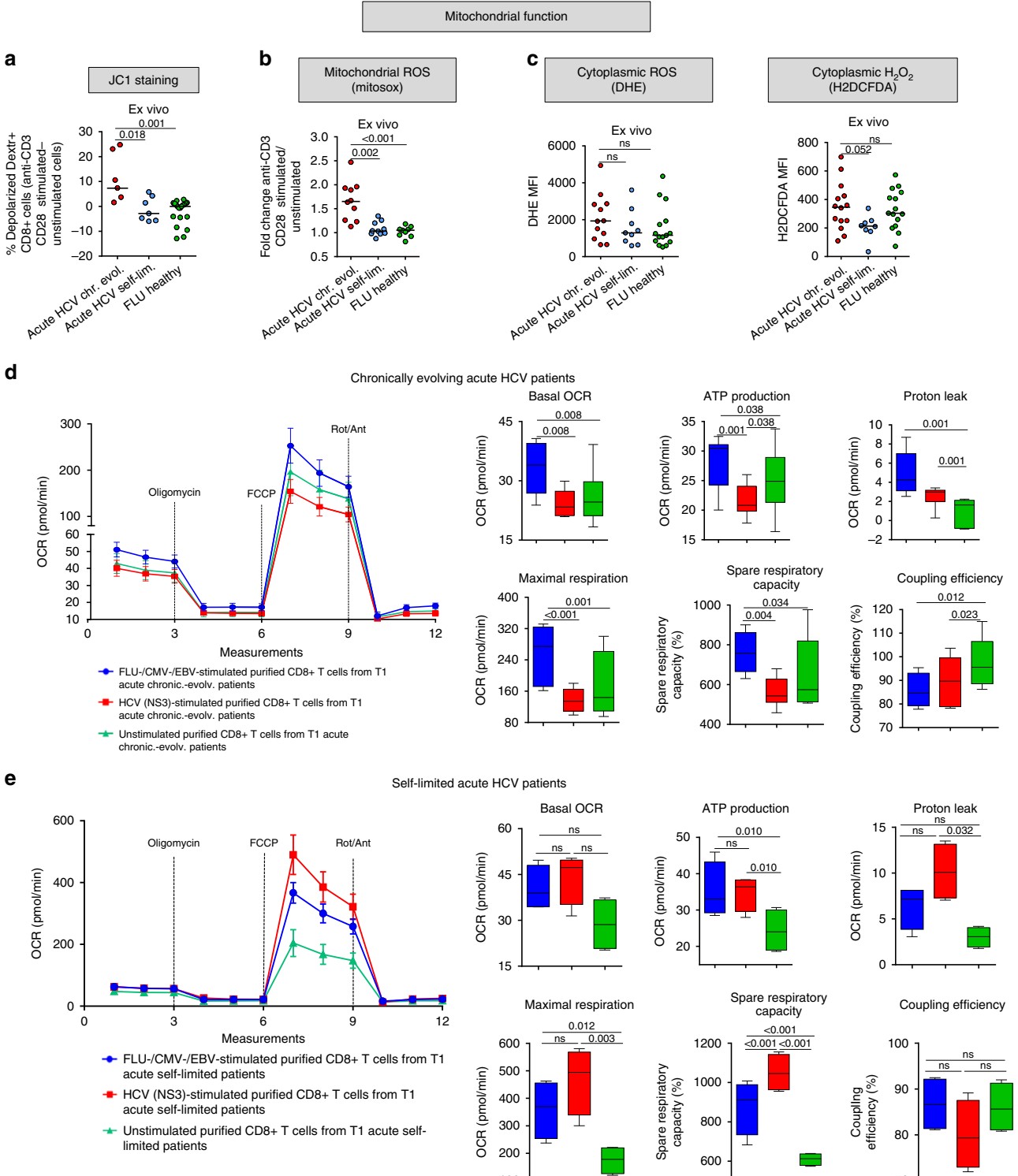

T cells, which are comparable to, or even lower than, the frequencies typically found in our acute HCV patients cohorts (data not shown).

**p53 drives early metabolic dysregulation in HCV-specific CD8+ T cells.** We then sought to elucidate the mechanisms responsible for glycolytic impairment and the functionally ineffective transcriptional upregulation of OXPHOS genes. To this end, we extracted a list of candidate regulatory genes (i.e., genes upregulated in at least five of the eight functional categories reported in Supplementary Fig. 2; see also the T1 sheets in Supplementary Data 2) and performed a mutual interaction relationship analysis. As shown by the interaction map in Fig. 4a, p53 (also known as TP53), a known negative regulator of glycolysis and an enhancer of OXPHOS[30,31], was central to this network of dysregulated genes. p53 upregulation, as well as enhanced expression of its phospho-activated (p-Ser15) form in HCV-specific CD8+ T cells from T1/early chronic compared to T1/early self-limited patients were confirmed at the protein level

**Fig. 3 Mitochondrial metabolism is impaired in HCV-specific CD8+ T cells from chronically evolving acute patients. a** Percentage of mitochondrial depolarized virus-specific CD8+ T cells, detected with HLA-A2+ dextramers ex vivo after overnight anti-CD3/anti-CD28 stimulation, by staining with the mitochondrial membrane potential (MMP) sensitive dye JC-1 (see Methods section for details). Dextramer-positive virus-specific depolarized cells were quantified by subtracting the percentage of FL1high/FL2low cells (JC-1 staining) detected in the unstimulated samples from the percentage of the corresponding cellular subsets detected in the stimulated samples, as previously reported[23]. **b** Mitochondrial superoxide levels determined ex vivo as in **a** with the MitoSOX Red dye. **c** Cytoplasmic reactive oxygen species (ROS) determined ex vivo, as in **a**, with the superoxide-specific dye DHE and the intracellular $H_2O_2$ specific dye H2DCFDA are shown on the left and on the right, respectively. **a–c** Data are presented as median fluorescence intensity (MFI) values; median values are indicated by horizontal lines. Different numbers of patients (represented by individual dots) were tested in each assay depending on dextramer-positive cell frequencies. **a–c** Differences between multiple groups were evaluated with the non-parametric Kruskal-Wallis test; p-values were corrected for pairwise multiple comparisons with the Dunn's test. **d** Oxygen consumption rate (OCR) data determined on the same samples ($n = 6$ T1/early chronically evolving acute patients) utilized for ECAR analysis (see Fig. 2d) before (basal level) and after addition of the mitochondrial stressors oligomycin, FCCP and rotenone/antimycin A, which were used to calculate ATP production, maximal respiration capacity, spare respiratory capacity, coupling efficiency and proton leak, as indicated. **e** OCR, ATP production, maximal respiration capacity and spare respiratory capacity were determined on the same samples ($n = 4$ T1/early self-limited acute patients) utilized for ECAR analysis in Fig. 2e, and were calculated as in **d**. In **d** and **e**, OCR values are given as the mean ± SD in the left-side and are presented as box-and-whisker plots (with median and 5–95 percentile) in the right-side. **d–e** Statistical analysis was performed with the Friedman test to compare different stimuli; p-values have been corrected for pair-wise multiple comparisons with the Conover's test.

(Fig. 4b, c). The Ataxia Telangiectasia Mutated (ATM) kinase, an upstream regulator of p53[32,33], was also upregulated in T1/early chronically evolving patients (Supplementary Data 1 and 2) and the phospho-activated (p-Ser1981) ATM protein was significantly increased in CD8+ T cells from T1/early chronic compared to T1/early self-limited HCV patients (Fig. 4d). ATM is responsible for the AMP-activated protein kinase (AMPK)-dependent activation of the stress-sensor MAPK p38a[34,35]. Interestingly, p38a (also known as MAPK14) and the AMPK subunits PRKAA1, PRKAB1 and PRKAG1 were also upregulated at the transcriptional level (Supplementary Data 1 and 2) and increased levels of the phospho-activated (p-Thr180) form of p38a were detected in CD8+ T cells from T1/early chronically evolving HCV patients (Fig. 4e).

ATM is activated by DNA damage, ROS overproduction and oxidative stress[36,37]. Notably, treatment with the ROS scavenger resveratrol[38,39] resulted in a reduction of mitochondrial reactive oxygen species, which was accompanied by a significant decrease of phospho-ATM and a smaller reduction of phosho-p53 and phospho-p38a (Fig. 5a). Similar results were obtained with the antioxidant N-acetyl-L-cysteine (NAC) in a smaller cohort of patients (data not shown). Conversely, no significant phospho-ATM decrease was observed when resveratrol was applied to CD8+ T cells from T1/early chronically-evolving patients stimulated with control viral peptides (Supplementary Fig. 6a).

To further probe the signal transduction network operating in T1/early-exhausted HCV-specific CD8+ T cells, we assessed the effect of different chemical inhibitors targeting ATM, p53, AMPK, and p38 on the metabolic profile and anti-viral function of CD8+ T cells (see Methods section for details). ATM, p53, and p38a blockade strongly increased GLUT1 expression and glucose uptake capacity, whereas a smaller effect was elicited by AMPK inhibition (Fig. 5b, c). Treatment with ATM, p53 and p38a inhibitors also reduced PD-1 levels (Fig. 5d), suggesting that PD-1 expression can be at least partially modified at an early stage of chronically evolving infection.

Interestingly, treatment with specific transducer inhibitors also enhanced the antiviral capacity of HCV-specific CD8+ T cells from T1/early chronically evolving patients, as revealed by the increased cytokine (IFN-γ, TNF-α, and IL-2) production observed in both ex vivo (Fig. 5e) and post 10-day culture (Fig. 5f). Instead, only a modest effect was induced by the same inhibitors on FLU-stimulated, CMV-stimulated, or EBV peptide-stimulated PBMC from T1/early chronically-evolving patients (Supplementary Fig. 6b–e), as well as on NS3-HCV-peptide-stimulated PBMC from T1/early patients with an acute

self-limited infection (Supplementary Fig. 7a, b). The above data suggest that transducer inhibitors may represent promising therapeutic tools to prevent progression and worsening of T cell exhaustion.

**Global transcriptional repression in T2/late HCV-specific CD8+ T cells.** To delineate the dysfunctional features of the T2/late-chronic stage, we focused on the comparison between HCV-specific CD8+ T cells from established chronic and late-resolved patients. Most functional categories identified as dysregulated in the T2/late stage by GSEA were downregulated, as indicated by the negative Normalized Enrichment Score (NES) values reported in Supplementary Data 2 (T2 sheets). Gene sets significantly enriched at both T1/early and T2/late time-points but oppositely regulated (i.e., upregulated at T1 and downregulated at T2) are outlined in Fig. 6a. T2/late-downregulated gene sets include genes coding for genome safeguard, cell cycle/checkpoint regulation, proteasomal degradation, mitochondrial metabolism and OXPHOS components, as well as multiple signal transducers acting downstream to TCR activation (Fig. 6a, b and T2 Sheets in Supplementary Data 2). qPCR validation, applied to the comparison of T2/late-chronic vs. resolved HCV patients and healthy FLU controls, confirmed the expression trends revealed by microarray analysis for a subset of T cell effector (e.g., ZAP70, LAT, AKT1, and Tbet, also known asTBX21) and regulator genes (e.g., ADCY4, BATF), whereas basal expression levels or a weak upregulation were observed for the TCR regulators PD-1 and CTLA-4, respectively (Supplementary Fig. 8)[4,40,41]. To further investigate these signaling pathways at the protein level, all patient categories were analyzed longitudinally (see Supplementary Fig. 9). Phosphorylated-ATM levels did not change in CD8+ T cells from T2/late chronic patients (Supplementary Fig. 9a), whereas total p53 levels increased in T2/late chronic compared to T2/late resolved patients and healthy controls (Supplementary Fig. 9b, on the left). In contrast, we did not observe any significant increase in Ser15 phosphorylated p53 levels (Supplementary Fig. 9b, lower graph), as it may be expected considering that phosphorylation at this position is mediated by the ATM, ATR, and AMPK kinases, all of which are transcriptionally down-regulated in T2/late chronic patients (see T2 sheets in Supplementary Data 2). However, since p53 can be phosphorylated by other protein kinases at different sites[42]—e.g., Ser46 phosphorylation by p38, whose phosphorylated form did not change appreciably between T2/late chronic and resolved patients (Supplementary Fig. 9c)—we cannot conclude that p53-dependent signaling is necessarily depressed.

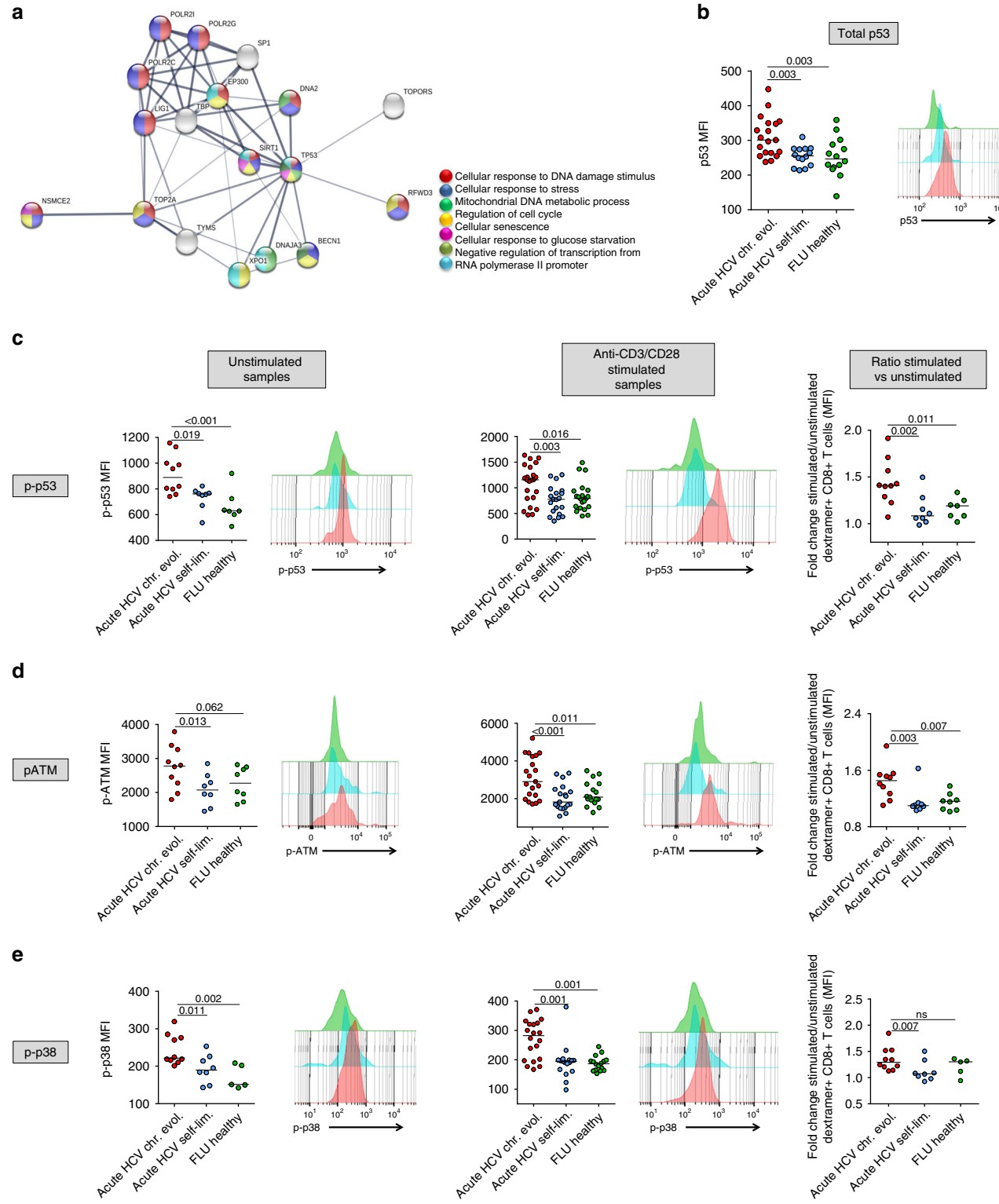

Furthermore, a significant reduction of mitochondrial polarization (Fig. 6c, left panel) and a marked accumulation of proteasome-undegraded aggregated proteins (Fig. 6c, right panel) were detected ex vivo in HCV-specific CD8+ T cells from T2/late chronic compared to T2/late resolved patients and healthy controls. Further investigation of cellular metabolism at the T1 and T2 time-points, revealed sustained glucose uptake and GLUT1 expression levels in T2/late chronic patients

(Supplementary Fig. 10a), associated with elevated PD-1 protein expression (Supplementary Fig. 10b), dysfunctional mitochondria and excess mitochondrial (but not cytoplasmic) ROS levels (Supplementary Fig. 10c, d). Mitochondrial defects were already evident in the T1/early acute phase of chronically evolving infection (Supplementary Fig. 10c). Proteasomal dysfunction, instead, was not detectable in the same group of T1/early chronically evolving patients at the T1 time-point

**Fig. 4 ATM and p53 pathways are activated in T1/early HCV-specific CD8+ T cells. a** Interaction network of genes involved in at least five of the eight dysregulated processes identified by GSEA (outlined in Supplementary Fig. 2). The network was generated using STRING v. 10.5. Node colors refer to enriched pathways associated with the proteins represented in the network. Line thickness indicates the degree of confidence prediction of the interactions. **b** Intracellular staining for total p53 of dextramer positive virus-specific CD8+ T cells from patients in the acute phase of HCV infection (chronically evolving T1/early $n = 19$ and HCV self-limited T1/early $n = 14$) or from healthy controls ($n = 14$), performed after overnight stimulation with anti-CD3/anti-CD28 (ex vivo staining). **c** Intracellular staining for phospho-p53 (Ser15) of dextramer positive virus-specific CD8+ T cells from PBMCs derived from patients in the acute phase of HCV infection or from healthy controls was performed with no stimulation and after overnight anti-CD3/anti-CD28 stimuli (left and middle panels, respectively). The plot on the right represents the ratio between anti-CD3/anti-CD28 stimulated and unstimulated virus-specific CD8+ T cells. **d** Intracellular staining for phospho-ATM (Ser1981) of dextramer positive virus-specific CD8+ T cells as in **c**. **e** Phospho-p38 (Thr180) intracellular staining of dextramer positive virus-specific CD8+ T cells as in **c**. Data in panels from **b** to **e** are presented as median fluorescence intensity (MFI), with median values indicated by horizontal lines. Different numbers of patients (represented by individual dots) were tested in each assay depending on dextramer-positive cell frequencies. Representative overlay histograms are shown next to each plot in panels from **b** to **e**. All data were analyzed with the Kolmogorov-Smirnov test. Differences between multiple groups were evaluated with the nonparametric Kruskal-Wallis test; *p*-values were corrected for pairwise multiple comparisons with the Dunn's test (JASP software).

(Supplementary Fig. 10e, red circles), in accordance with transcriptome data indicating a strong upregulation of proteasomal subunits genes at this stage of infection (Fig. 6a, Supplementary Fig. 2 and T1 sheets in Supplementary Data 2).

Taken together these results indicate that mitochondrial and proteasomal functions, despite their divergent behavior in the early phase of infection (T1), are profoundly impaired in HCV-specific CD8+ T cells from T2/ chronic patients.

**Epigenetic repressive mechanisms in fully exhausted HCV-specific CD8+ T cells.** To explain the massive transcriptional downregulation that accompanies and likely contributes to the chronic outcome of HCV infection, we hypothesized the involvement of chromatin-based silencing mechanisms. To test this hypothesis, we used the EpiFactors database of epigenetic regulators[43] as a reference to interrogate the T1/early and the T2/late transcriptome datasets for dysregulated chromatin modifiers. Genes comprised in the histone H2A/H2B deubiquitination (DUB) and the histone methyltransferases (HMT) complex denominations were consistently upregulated in chronic patients compared to resolvers and healthy controls (Fig. 6d). We focused on the HMTs G9a (also known as EHMT2), which methylates lysine 9 of histone 3, and EZH2, which methylates lysine 27 of histone 3, because of their repressive effects[44–46]. As shown in Fig. 6e, significantly higher levels of H3 K9 di-methylation were detected in HCV-specific CD8+ T cells from T2/late chronic patients compared to T2/late resolvers and healthy controls. Interestingly, under the same short-term culture conditions, H3 K9 di-acetylation, which marks relaxed, transcriptionally active chromatin and is mutually exclusive with the repressive H3 K27me3 mark, displayed an opposite behavior, being significantly more represented in CD8+ T cells from T2/late resolvers and healthy controls compared to T2/late chronic patients (Fig. 6e). Interestingly, repressive chromatin marks were already detectable in the early phase (T1) of chronically evolving infections compared to healthy controls (Supplementary Fig. 10f).

To further investigate the relationship between repressive histone methylation and T cell functionality, we used four different inhibitors to selectively target the T2/late-chronic phase upregulated HMTs G9a (UNC0638 and BIX01294) and EZH2 (GSK126 and EPZ005687). Treatment with HMT inhibitors increased GLUT1 levels and glucose import in HCV peptide-stimulated PBMC from T2/late chronic patients (Fig. 7a, b). HMT inhibitors also reduced PD-1 expression levels in T1/early but less efficiently in T2/late HCV-specific CD8+ T cells (Fig. 7c). Moreover, the number of single-positive and double-positive IFN-γ/TNF-α-producing CD8+ T cells was markedly increased by HMT inhibitor treatment upon peptide stimulation both ex vivo (Fig. 7d–f) and after short-term culture (Fig. 7g–j and

Supplementary Fig. 11), with a particularly significant effect on T2/late-chronic CD8+ T cells. As shown by the increased frequency of HCV dextramer-positive CD8+ T cells upon stimulation with HLA-A2 restricted peptides, HMT inhibitors, which caused a reduction of H3 K9 di-methylation detectable both ex vivo and after 10 days of culture in CD8+ T cells from T2/late chronic patients (Fig. 7k), also improved the expansion capacity of these late-exhausted cells (Supplementary Fig. 12). This effect was appreciably more potent on T2/late exhausted HCV-specific CD8+ T cells than on fully functional HCV-specific CD8+ T cells from T2/late resolved patients (Supplementary Fig. 13a) and functional CD8+ T cells of different virus-specificities (FLU, CMV, EBV in Supplementary Fig. 13b).

Given the persistent upregulation of p53 throughout the different phases of HCV infection, the effect of p53 inhibition on cytokine production, glucose uptake and PD-1 expression levels was also tested on T2/late chronic HCV-specific CD8+ T cells. As in the case of HMT inhibitors, all the tested parameters were positively affected by p53 blockade, with an increase in glucose uptake, a decline of PD-1 levels and an increased cytokine production (Fig. 7b, c, d–j, blue dots and Supplementary Fig. 11), thus pointing again to a key role of p53 as a central regulator of T cell exhaustion.

**CD8+ T cell function in DAA-treated chronic hepatitis C patients.** We finally asked if and to what extent HCV replication blockade and antigen decline may affect the metabolic and epigenetic dysregulation marks documented in CD8+ T cells from T2/late chronic patients. To address this question, we investigated glucose import, mitochondrial depolarization, PD-1 expression, proteasomal activity and repressive histone methylation in CD8+ T cells from T2/late chronic patients treated with direct acting antivirals (DAA; see Supplementary Data 3a, b). After HCV clearance, glucose import was significantly reduced (Fig. 8a), even though it did not reach the levels found in spontaneous resolvers. Mitochondrial depolarization declined in T2/late HCV-specific CD8+ T cells from some but not all patients at the end-of-therapy (EOT) (Fig. 8a), with a behavior similar to what we observed in the case of PD-1 expression levels (Fig. 8a). This contrasts with the recovery of proteasomal function revealed by the significant reduction in unfolded protein aggregates (Fig. 8a). Variable results were obtained with the repressive H3K9me2 mark, which at EOT remained higher than in spontaneous T2/late resolvers and healthy controls (Fig. 8a).

We also compared the effect of histone methylation and p53 inhibition on the antiviral activity of CD8+ T cells from T2/late chronic patients before and after DAA therapy (see Supplementary Data 3a, b). Treatment with p53 and HMTs inhibitors (pifithrin-alfa for p53, GSK126 and EPZ005687 for EZH2,

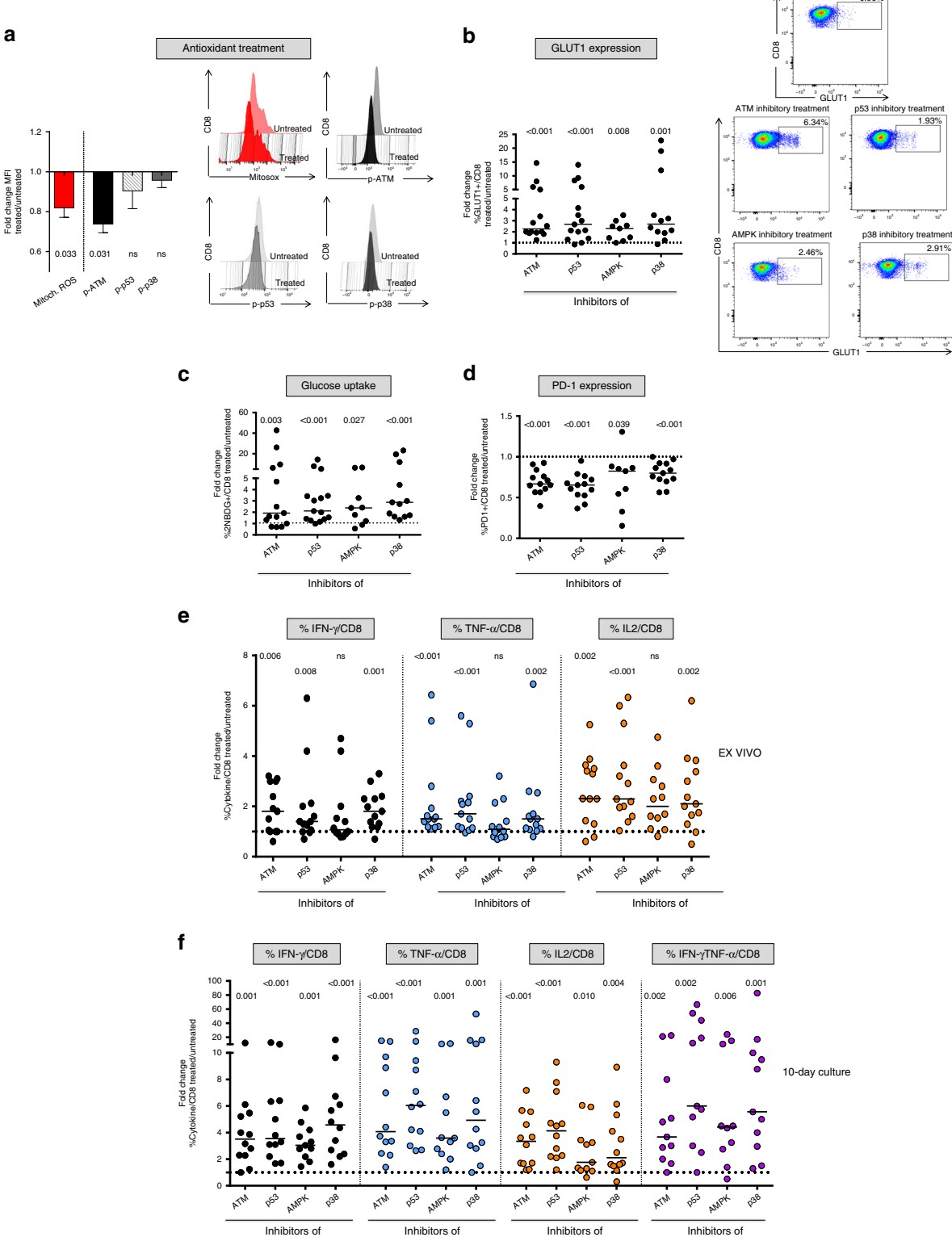

UNC0638 and BIX01294 for G9a) markedly increased the frequency of single-positive and double-positive IFNγ/TNFα-producing CD8+ T cells upon peptide stimulation both ex vivo (Fig. 8b) and after 10 days of culture (Fig. 8c). Conversely, the same set of inhibitors had a modest effect when supplied after therapy withdrawal (EOT), with a significant increase of anti-viral cytokine production only observed after 10 days of culture but not under ex vivo conditions (Fig. 8b, c and Supplementary Fig. 11).

## Discussion

Knowledge of the molecular and cellular features of the T cell responses associated with self-limited compared to persistent viral

**Fig. 5 Blocking dysregulated intracellular signaling pathways can reverse early metabolic and functional CD8+ T cell defects. a** PBMC from T1/early chronically-evolving patients were stimulated overnight with HCV-NS3 peptides in the presence or absence of the ROS scavenger resveratrol[38,39] (treated vs. untreated) and then stained with MitoSOX Red to assess mitochondrial superoxide content and with anti-phospho-ATM (Ser1981), phospho-p38 (Thr180), and phospho-p53 (Ser15) antibodies. Bars represent mean fold-change values + SEM derived from 6 patients. Representative overlay histograms are illustrated on the right. **b** PBMCs from T1/early chronically-evolving patients were stimulated for 40 h with HCV-NS3 peptides in the presence or absence of specific ATM (KU-55933), p53 (Pifithrin-α), AMPK (Dorsomorphin), and p38a (SB203580) inhibitors, followed by flow cytometry determination of GLUT-1 expression levels. Representative dot plots are illustrated on the right. Glucose uptake studied via incorporation of the glucose analog 2-NBDG (**c**) and PD-1 expression (**d**) have been measured as in **b**. **e** IFN-γ, TNF-α, and IL2 production by CD8+ T cells cultured as in **b**. Data are presented as the ratio between the percentage of cytokine positive CD8+ T cells detected in inhibitor-treated vs. untreated cultures (fold-change). **f** IFN-γ, TNF-α, IL2 single positive, as well as double-positive IFN-γ+/TNF-α+CD8+ T cells generated in short-term T cell lines upon 10-days stimulation with HCV-NS3 peptides in the presence or absence of the inhibitors specified in the legend to panel **b**. Data shown in all panels are presented as fold-change of treated vs. untreated CD8+ T cells. Horizontal lines in panels **b** to **f** represent median values; data were analyzed statistically with the Wilcoxon signed-rank test; NS = not significant.

infections has greatly advanced in recent years[1,6,12,13,17,19]. However, our understanding of the regulatory mechanisms responsible for these different outcomes remains largely incomplete, especially in the case of human infections[13,25]. Here, we combined transcriptome profiling, cell metabolic analyses and selective targeting of newly identified dysregulated pathways, to examine and compare the functional features of HCV-specific CD8+ T cells at different time-points along the transition from acute hepatitis to a resolved (memory T cell generation) or a chronic (exhausted T cells) outcome of infection. Short-term culture conditions in the presence of either HCV-specific peptides or anti-CD23/anti-CD28 antibodies were set-up in order to circumvent the strong limitations inherent to the extremely low frequency of HCV-specific CD8+ T cells. These conditions, which differ from the much more prolonged stimulation commonly applied in human studies of virus-specific CD8+ T cells, were also aimed to limit possible artifactual metabolic alterations associated with long-term culture and in vitro expansion.

We found that a predominantly upregulated gene expression profile marks HCV-specific CD8+ T cells committed to exhaustion at early times of infection, in patients who will ultimately develop chronic hepatitis C. However, this extensive upregulation, which involves DNA damage stress responses and multiple intracellular signaling pathways, including T cell receptor downstream effectors, the CD28-dependent PI3K/Akt anabolic pathway and several mitochondrial components, does not lead to an improved antiviral function but rather translates into a deep functional impairment at both the energetic and metabolic level. Such a dissociation between transcriptional and functional/metabolic outputs was particularly evident for OXPHOS impairment, which was associated with an altered MT membrane potential and ROS accumulation.

While OXPHOS was transcriptionally upregulated but functionally depressed, glucose utilization and related glycolytic functions were downregulated at both the gene expression and the functional level in T1/early chronic CD8+ T cells committed to exhaustion. Reduced glycolytic activity and inefficient OXPHOS have recently been reported for exhausted CD8+ T cells in the mouse LCMV model of persistent viral infection[25]. Thus, a common metabolic alteration with divergent transcriptional and functional responses likely reflecting a failed compensatory attempt, is shared by exhausted CD8+ T cells across different species. This profound metabolic derangement is expected to impact on the capacity of T cells to express efficient anti-viral effector functions[18–20] and to generate protective memory[21,22] suggesting that its selective targeting may provide new avenues for novel rational therapeutic interventions.

p53 is known to negatively affect glycolysis and to promote OXPHOS, thus resulting in ROS production[30,31]. In the case of

prolonged stimulation and/or irreparable DNA damage, p53 activation induces a sustained pro-oxidative state that may result in the repression of mitochondrial biogenesis and mitochondrial dysfunction[30,31]. Different lines of evidence point to p53 as a highly plausible driver of the initial transition to the exhaustion program in HCV infected patients. p53, which was found to be upregulated at both the transcript and protein level (including its phospho-activated form) in HCV-specific CD8+ T cells from early-chronic patients, emerged as a central hub in the network of regulatory genes retrieved from interaction analysis of our transcriptome data. Indeed, p53 is modulated transcriptionally by type I interferons (IFN)[31] that are actively engaged in the initial phase of HCV infection[47,48], as also indicated by the multiple IFN-stimulated genes we found to be upregulated in T1/early-exhausted HCV-specific CD8+ T cells. Notably, the DNA damage sensor kinase ATM, which activates p53 by Ser15 phosphorylation, was also upregulated at the transcript and protein level in T1/early exhaustion-committed CD8+ T cells. Despite the present lack of direct supporting evidence, it is conceivable to imagine that the DNA replication stress caused by elevated viral loads and the resulting persistent TCR signaling are responsible for ATM upregulation. Accordingly, transcripts coding for T cell receptor downstream effectors, as well as cell cycle machinery and DNA replication components were also found to be upregulated in T1/early-exhausted HCV-specific CD8+ T cells.

In addition to a role in the cellular metabolic homeostasis, p53 is primarily known to play a general role in promoting apoptosis and permanently blocking cell proliferation under various stress conditions[30]. In our study, only a very limited proportion of p53high HCV-specific CD8+ T cells from chronically evolving acute patients were found to be positive for Caspase 9, which has been reported to play a major role in apoptosis induction in the acute phase of HCV infection[49]. Conversely, nearly half of the p53high HCV-specific CD8+ T cells in our study expressed the anti-apoptotic BCL2 regulator at high levels (data not shown), thus suggesting that only a minor fraction of the p53-positive HCV-specific CD8+ T cell population we used for our functional analyses was actually undergoing apoptosis. Moreover, the actual amount of pre-apoptotic cells might have been further lowered by overnight stimulation of dextramer-positive CD8 T cells with anti-CD3 and anti-CD28 antibodies, a treatment that is expected to drive apoptosis-committed T cells toward apoptotic death[50,51]. Altogether, the above data strongly suggest that the p53 increase observed in chronically evolving HCV patients is related to less conventional (e.g., metabolic adaptation) p53-mediated functions, rather than to apoptosis promotion.

Importantly, a key role of p53 in the initial triggering of exhaustion is also supported by the restoration of both metabolic

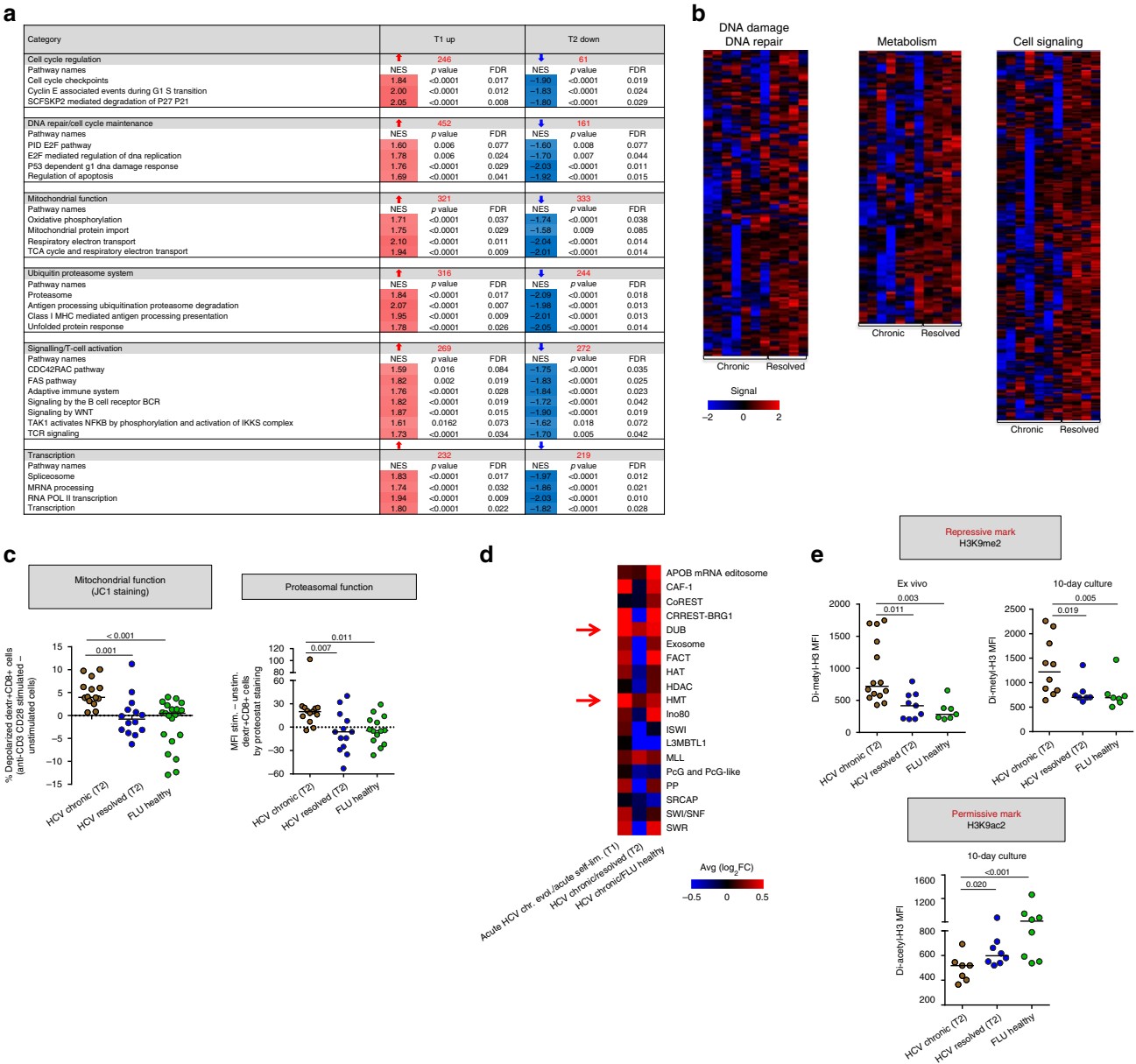

**Fig. 6 Epigenetic transcriptional repression in exhausted HCV specific CD8+ T cells from chronic patients. a** Six distinct functional groups of pathways enriched in upregulated genes (red) in T1/early and displaying the opposite trend (blue) in T2/late identified by GSEA (MSigDB, C2 canonical pathways and C5 Gene Ontology sets) in HCV-specific CD8+ T cells from chronically evolving patients. NES = normalized enrichment score; FDR = False Discovery Rate. In red, above the p-value columns, is shown the total number of genes significantly upregulated in T1/early and downregulated in T2/late in each group of pathways. **b** Heat-maps comparing the expression profiles of leading genes belonging to the DNA repair/damage response, metabolism and cell signaling pathways in chronic and resolved infections (see also the T2 sheets in Supplementary Data 2). **c** Mitochondrial (left panel, JC-1 staining) and proteasomal (right panel, ProteoStat staining) functions were assessed in dextramer-stained HCV-specific CD8+ T cells from T2/late chronic and T2/late self-limited HCV infection and in healthy controls following PBMC overnight stimulation with anti-CD3/CD28. MFI, Median Fluorescence Intensity. **d** Heat-map comparing the expression levels (average log2 fold change) of epigenetic regulatory complexes (derived from the EpiFactors database as detailed in Methods section) in chronic vs. self-limited infection (T1/early), chronic vs. resolved infection (T2/late) and chronic (T2/late) vs. healthy controls. **e** Repressive H3K9me2 (upper panels) and permissive H3K9ac2 (lower panel) histone marks determined by flow cytometry in dextramer-stained HCV specific CD8+ T cells from T2/late chronic and T2/late resolved HCV patients or healthy controls. PBMC were stimulated overnight with anti-CD3/CD28 (ex vivo staining) or for 10 days with HLA-A2-restricted HCV-specific or FLU-specific peptides. Horizontal lines in panels **c** and **e** represent median values. Differences between multiple groups in panels **c** and **e** were evaluated with the non-parametric Kruskal-Wallis test; p-values were corrected for pairwise multiple comparisons with the Dunn's test.

and antiviral functions, and the concomitant reduction of PD-1 levels, elicited by treatment with a specific p53 inhibitor. A link between p53 and immune-checkpoint regulators has recently been described in human cancer cells, which respond to genotoxic stress and DNA damage via p53-dependent upregulation of

PD-1 and its PD-L1 ligand[52]. The effect of p53 inhibition on PD-1 expression levels we observed in T1/early exhaustion-committed HCV-specific CD8+ T cells, thus provides further support to the existence of a regulatory link between p53 and PD-1, which may play an as yet unappreciated role in T cell exhaustion. The

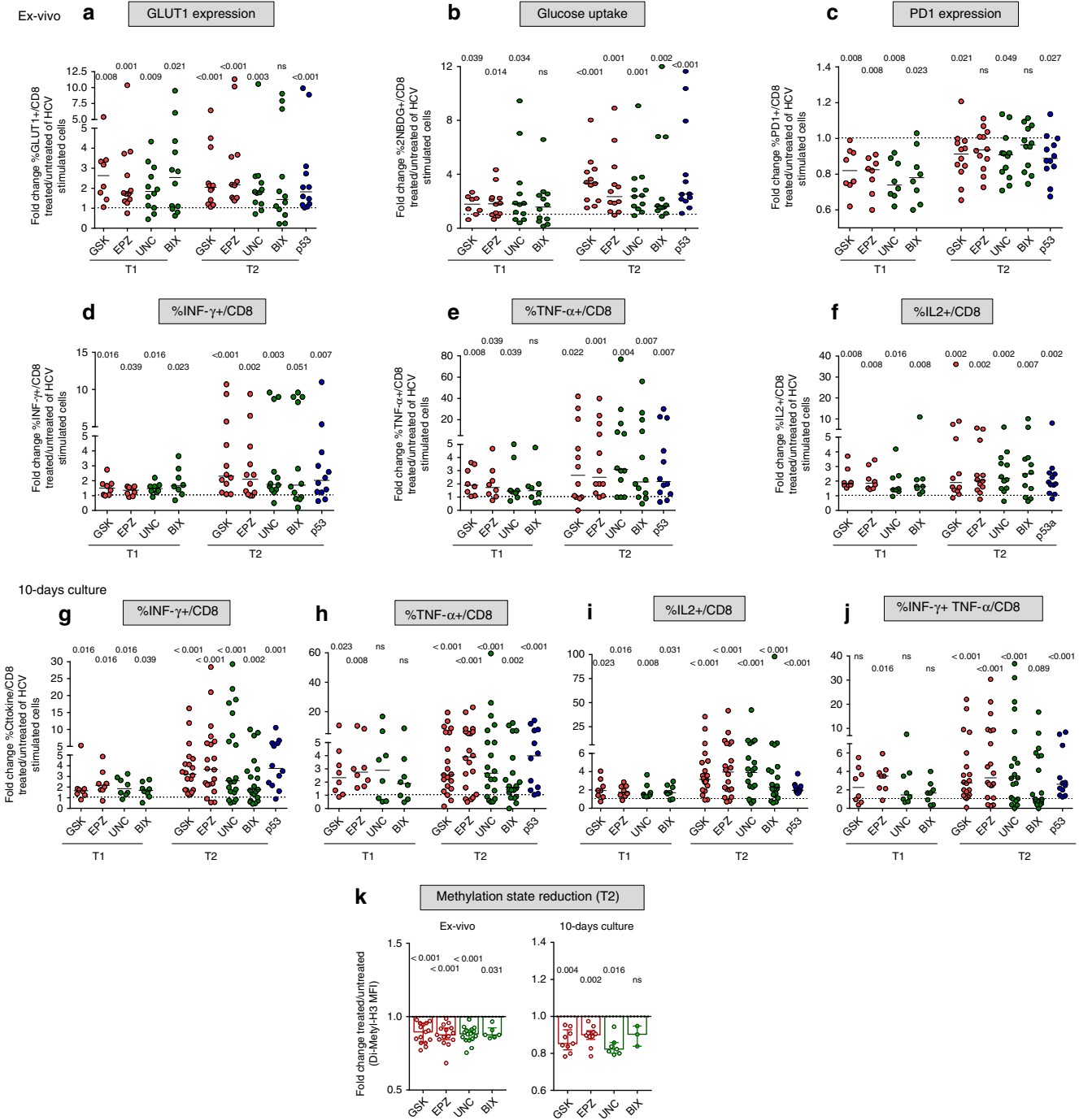

**Fig. 7 HMT and p53 inhibitors improve anti-viral and metabolic functions of exhausted HCV-specific CD8+ T cells.** PBMC from chronically evolving (T1/early) or chronic (T2/late) HCV patients were stimulated for 40 h with HCV-NS3 peptides in the presence or absence of the EZH2 inhibitors GSK126 (GSK) and EPZ005687 (EPZ) (red dots), of the EHMT2/G9a inhibitors UNC0638 (UNC) and BIX01294 (BIX) (green dots), and of the p53 inhibitor pifithrin-alfa (p53) (blue dots). HCV-stimulated CD8+ T cells were then tested in flow cytometry for GLUT-1 levels (**a**), glucose uptake (**b**), PD-1 expression (**c**), IFN-γ (**d**), TNF-α (**e**), and IL2 (**f**) production. **g–j** PBMC from chronically evolving (T1/early) or chronic (T2/late) HCV patients were stimulated for 10 days with HCV-NS3 peptides in the presence or absence of the inhibitors specified in **a** and CD8+ T cells were then tested for IFN-γ, TNF-α, IL2, and IFN-γ plus TNF-α production as indicated. Data are presented as the ratio (fold-change) between positive CD8+ T cells detected in treated vs. untreated cultures from individual patients. Statistical analysis was performed with the Wilcoxon signed-rank test; horizontal lines represent median values. **k** Reduction of the repressive H3K9me2 histone mark was assessed by flow cytometry on CD8+ T cells from T2/late chronic HCV patients stimulated for 40 h or 10 days as in **a**. Data are presented as the ratio (fold-change) between MFI (Median fluorescence intensity) of H3K9me2 CD8+ T cells detected in treated vs. untreated cultures from individual patients; statistical analysis was performed with the Wilcoxon signed-rank test; columns and dots represent median values and single patients, respectively.

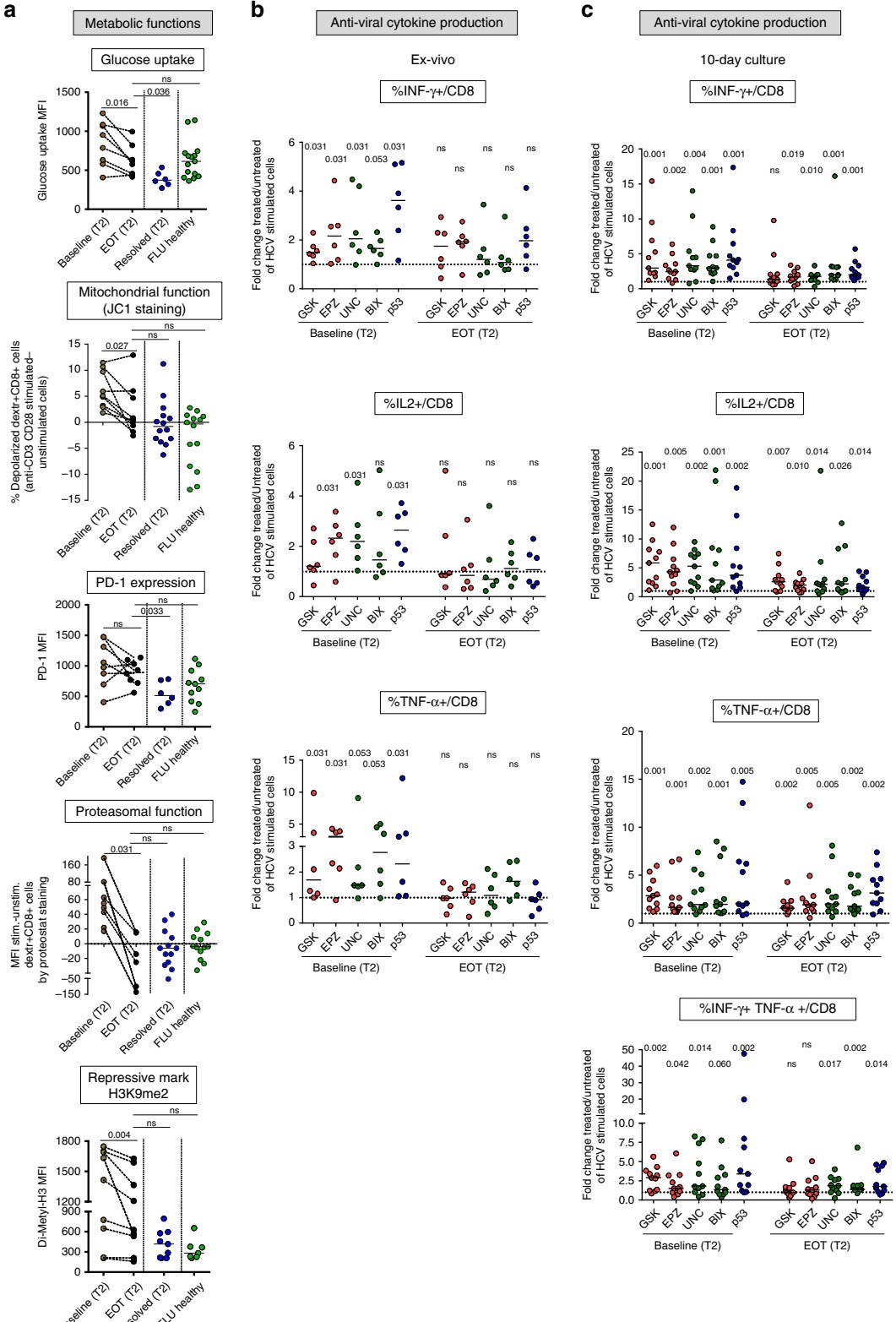

relevance of our model to other T cell exhaustion-associated pathologies, including cancer, is further corroborated by the increased effector function and enhanced glycolytic commitment exhibited by T cells derived from p53 knock-out mice[53]. In the same model, adoptive transfer of p53 knock-out T cells has been shown to improve the control of a subcutaneously established murine melanoma, thus highlighting a key role of p53 in modulating tumor-reactive T-cell responses with a potential translational significance in adoptive T-cell therapy[53].

ATM downstream targets upregulated in T1/early chronic CD8+ T cells also include AMPK (PRKAA1, PRKAB1, and PRKAG1 subunits) and the MAPK/stress sensor kinase p38a

**Fig. 8 Effect of DAA treatment and HMTs/p53 modulation on HCV-specific CD8+ T cells from chronic patients. a** Glucose uptake (measured by the incorporation of the glucose analog 2-NBDG), percentage of depolarized mitochondria (by staining with the mitochondrial membrane potential sensitive dye JC-1), PD-1 expression, proteasomal function (by ProteoStat staining) and repressive H3K9me2 mark were assessed in virus-specific, dextramer-stained CD8+ T cells from T2/late HCV patients ($n = 9$) at baseline and at EOT, from T2/late resolved patients and from healthy controls after overnight anti-CD3/anti-CD28 stimulation (see Methods section for details). All data were analyzed with the Kolmogorov-Smirnov test, followed by Wilcoxon matched-pairs signed rank test (paired for chronic patients at baseline vs. EOT). Conversely, differences between multiple patient groups were evaluated with the non-parametric Kruskal-Wallis test; $p$-values were corrected for pairwise multiple comparisons, with the Dunn's test. **b** PBMC from chronic HCV patients (T2/late; $n = 6$) at baseline and at EOT were stimulated for 40 h with HCV-NS3 peptides in the presence or absence of the EZH2 inhibitors GSK126 (GSK) and EPZ005687 (EPZ) (red dots), of the G9a inhibitors UNC0638 (UNC) and BIX01294 (BIX) (green dots), and of the p53 inhibitor pifithrin-alfa (p53) (blue dots), followed by co-staining for IFN-γ, IL2, and TNFα (left). **c** PBMC from baseline and EOT of T2/late chronic HCV patients ($n = 11$) were stimulated for 10 days with HCV-NS3 peptides as in **b**. Data in **b** and **c** are presented as the ratio (fold-change) between cytokine producing CD8+ T cells detected in treated vs. untreated cultures from individual patients; statistical analysis was performed with the Wilcoxon signed-rank test; horizontal lines represent median values.

(also known as MAPK14). The latter has been shown to be involved in T cell senescence with markedly negative effects on telomerase activity, cell proliferation and expression of key components of the TCR signalosome[34,35]. Accordingly, we found that T2/late exhausted HCV-specific CD8+ T cells display a profoundly reduced expression of intracellular transducers downstream to TCR activation, including the proximal kinases ZAP70 and CD3zeta chain (also known as CD247), AKT1, and PLCG1. Pharmacological inhibition of ATM, AMPK, and p38a rescued to varying extents the metabolic and antiviral activity defects associated to early exhaustion of T1/ chronically-evolving CD8+ T cells. Altogether, our data suggest that ATM, p53, and p38a cooperate in the establishment of a pre-senescent, exhaustion-oriented state.

Although we do not formally delineate a biochemical cascade leading to metabolic dysregulation in T1/early exhausted HCV-specific CD8+ T cells, ROS accumulation, resulting from OXPHOS derangement and impaired detoxification, links mitochondrial dysfunction to ATM and p53 activation; in line with this, antioxidant treatment led to a marked decline of phospho-ATM levels.

Contrasting with the predominant upregulation that characterizes exhaustion-committed CD8+ T cells in the T1/early phase of HCV infection, a massive transcriptional downregulation was observed in T2/late-exhausted CD8+ T cells. This involved a variety of genes and pathways related to core cellular processes such as cell cycle checkpoints, chromosome/telomere maintenance, transcription and mitochondrial function and to intracellular signaling downstream to TCR activation (see Supplementary Data 2 for a complete list of T2-downregulated genes). Despite progression towards a widespread downregulation, a number of genes were found to be upregulated in T2/late-exhausted CD8+ T cells (T2 sheets in Supplementary Data 1), including genes coding for negative regulators of TCR signaling (e.g., adenylate cyclase 1 and 4, also known as ADCY1 and 4; NFAT1 also known as NFATC2; and NFAT2 also known as NFATC1)[40,54], mRNA translation (e.g., the interferon-inducible protein kinase PKR also known as EIF2AK2; the GCN2 kinase also known as EIF2AK4; and 4EBP1 also known as EIF4EBP1)[55], and chromatin architectural components (e.g., the high-mobility group protein HMGA1)[56].

Chromatin remodeling via histone modification has recently been implicated in various cell fate transitions associated with T cell development[57–60]. The PRC2 repressive complex and its catalytic subunit, the H3K27 histone methyltransferase EZH2, have been shown to regulate terminal effector (and memory) CD8+ T cell differentiation[57–61], whereas the H3K9 methyltransferase G9a has been reported to promote the expression of inhibitory receptors during T cell exhaustion in chronic LCMV infection and to repress type I cytokine gene expression in Th2

cells[62,63]. Through EpiFactors database investigation, we found that both HMTs and additional PRC2 complex components (e.g., EED, Suz12, RBBP4, and Jarid2)[61] are upregulated in T2/late-exhausted HCV-specific CD8+ T cells compared to the same cells from T2/late resolved patients. Accordingly, we observed an increase in the repressive H3K9me2 mark, accompanied by reduced levels of the activating H3K9ac2 mark (mutually exclusive with the repressive H3K27me3 modification) in T2/late exhausted HCV-specific CD8+ T cells. Notably, EZH2 and G9a upregulation was already apparent at the T1/early time-point. Whether this reflects an early initiation of the repressive chromatin remodeling program or the presence of a qualitatively different methylation landscape in the two phases of the infection process remains to be determined. Also lacking, is direct evidence on the actual trigger of histone methylation-dependent silencing, although it is likely that, at least in the case of EZH2, upregulation is driven by DNA damage and the increased E2F1 activity associated with cell proliferation[64].

A causative role of H3K9 and H3K27 methylation in the modulation of CD8+ T cell exhaustion is strongly supported by the significant restoration of metabolic and antiviral functions, together with an improved expansion capacity, elicited by pharmacological inhibition of the G9a and EZH2 histone methyl transferases in T2/late-exhausted HCV-specific CD8+ T cells. A similar functional recovery was observed upon p53 blockade and it was paralleled by both an improved CD8+ T cell metabolic capacity, including increased glucose uptake and GLUT1 expression levels, and a significant decrease of PD-1 expression. Interestingly, EZH2 inhibition also positively affected GLUT1 expression but with a limited or no effect on PD-1. This suggests the existence of only a partial overlap between the transcriptional repression programs driven by the EZH2 and G9a HMTs and p53.

Given the functional recovery results obtained with multiple, rationally selected inhibitors, some of which are currently being tested in oncology clinical trials[65], it is conceivable to imagine that knowledge generated in this study holds considerable potential for translation into novel and more effective T cell reconstitution strategies.

The contribution of HCV-specific T cells to virus clearance promoted by DAA treatment is still a matter of debate[66,67]. The prevalent view, however, based on studies performed in resolved HCV infections, either spontaneously or after IFN-alpha therapy[68], and, more recently, in DAA-treated patients[12] is that resolution of HCV infection does not lead to a complete recovery of the HCV-specific T cell function. Specifically, resolution of infection in DAA treated patients does not appear to be associated with the acquisition of a conventional memory phenotype by HCV-specific CD8+ T cells. In fact, most virus-specific T cells differentiate into a memory-like (or "exhausted

memory") CD8+ T cell population (CD127+/TCF1+/BcL2+/ PD-1+) with an ability to produce anti-viral cytokines and to proliferate greater than terminally exhausted (CD127-/PD-1$^{hi}$) CD8+ T cells but lower than that of real memory CD8+ T cells (CD127+/PD-1-)[12].

In keeping with the partial DAA mediated recovery highlighted by some of the above studies, we did not observe a complete correction of dysfunctional metabolic and repressive histone methylation marks in virus-specific CD8+ T cells from T2/late DAA-treated patients. Interestingly, we also observed a higher efficacy of HMTs and p53 inhibitors when applied before the start of DAA therapy (i.e., in the presence of a high viremia) rather than after HCV clearance at DAA suspension.

Combined DAAs and HMTs/p53 inhibitors may thus be particularly valuable for the secondary treatment of HCV-positive patients not properly responding to last-generation antivirals. Also notable is the overlap (and co-enrichment) of the gene-sets found to be dysregulated in the present study and those of conceptually similar transcriptome profiling studies conducted in the LCMV model of chronic infection[69,70] (Supplementary Fig. 14a), in HIV (progressors vs. controllers; see Supplementary Fig. 14b)[41], in HBV (Supplementary Fig. 14c)[23] and in HCV (early chronically-evolving patients) infections (Supplementary Fig. 14d)[26]. This suggests that the HMTs/p53 inhibitors might also be effective in other viral and non-viral chronic pathologies (including cancer) sharing a T cell exhaustion phenotype.

## Methods

**Human subjects and samples**. The following groups of patients (Supplementary Data 3) were enrolled into the study at the Unit of Infectious Diseases and Hepatology of the Azienda Ospedaliero-Universitaria of Parma, Italy:

84 T2/late treatment-naive patients with chronic active hepatitis C (aged 20–67 years; median 45 years); diagnosis was based on the finding of positive serum HCV-RNA and alanine aminotransferase (ALT) elevation for at least 1 year; 20 T2/late chronic active hepatitis C genotype 1 patients were followed through IFNα-free DAA therapy (Supplementary Data 3b).
85 T1/early patients with acute hepatitis C, 52 of them with chronic evolution of infection (aged 18–73 years; median 41 years) and 33 with a self-limited outcome (aged 23–70 years; median 40 years), with clinical, biochemical and virological evidence of acute HCV infection (transaminase levels at least ten times higher than the upper limit of the normal range, anti-HCV antibody, and HCV-RNA positive);
26 T2/late subjects who spontaneously recovered from acute HCV infection (aged 19–64 years; median 33 years), recruited 8 to 18 months after the peak of ALT elevation;
28 healthy subjects (aged 23–54 years; median, 36 years), as controls.

All patients were negative for anti-hepatitis B virus, delta virus, human immunodeficiency virus type 1 (HIV-1) and type 2 (HIV-2) antibodies and for other markers of viral or autoimmune hepatitis. No randomization was used to determine patient groups, and during all experiments, investigators were not blinded to the group allocation. The study was approved by the Government of the Emilia-Romagna Region, Italy (protocol n. 1786/2012) and by the Ethical Committee of the Azienda Ospedaliero-Universitaria of Parma, Italy, (protocol n. 9787/2013 and protocol n. 9050/2017) and all subjects provided written, informed consent.

**Synthetic peptides, peptide-HLA class I dextramers, and antibodies**. A total of 130, 15-mer overlapping peptides, corresponding to the entire sequence of the HCV-NS3 protein of genotype 1 (Mimotopes, Victoria, Australia) were pooled into a single mixture and used in functional validation experiments performed in HLA-A2-negative patients. A pool of immunodominant peptides from CMV, EBV, and FLU sequences of various HLA-class I and II restriction was used in HLA-A2-negative patients as a control for other viral specificities. Peptides covering the HLA-A2-restricted genotype-specific HCV epitopes corresponding to NS3 amino-acids 1073–1081, 1406–1415, NS4 1992–2000, and NS5 2627–2635, and to the influenza virus (FLU) matrix amino-acids 58–66 (Mimotopes, Victoria, Australia) and the corresponding PE-labeled or APC-labeled dextramer peptide-HLA class I complexes from Immudex (Copenhagen, Denmark) were used for virus-specific T cell functional assays.
Anti-Human CD3 (BD Horizon™ PE-CF594, clone HCHT1, 1:200, Cat# 562280); Anti-Human CD3 (BD Horizon APC-R700, clone HCHT1, 1:200,Cat# 565119); Anti-Human CD3 (PE/Cy7, clone HIT3a, BioLegend, 1:100, Cat#300316); Anti-Human CD8 (PE-Cy™7, clone RPA-T8, BD Biosciences, 1:100, Cat#557746);

Anti-Human CD8 (CD8-AlexaFluor®700, clone RPA-T8, BD Biosciences, 1:50, Cat#557945); Anti-Human CD8 (PerCP, clone BW135/80, Miltenyi Biotec, 1:200, Cat#130-113-160); Anti-Human CD8 (BD Pharmingen™ APC-H7, clone SK1, BD Biosciences, 1:50, Cat# 560179); Anti-Human CD4, PE, clone RPA-T4, BD Biosciences, 1:50, Cat# 555347); Anti-Human CD4, AlexaFluor®647, clone RPA-T4, BD Biosciences, 1:50, Cat#557707); Anti-Human IFN-γ (APC-R700, clone B27, BD Biosciences 1:200, Cat# 564981); Anti-Human IL-2 (APC, Clone MQ1-17H12, BD Biosciences, 1:50, Cat# 554567); Anti-Human CD279 (PD-1) (PE/Cy7, clone EH12.2H7, BioLegend, 1:20, Cat# 329918); Anti-Human TNF-α (FITC, Clone cA2, Miltenyi Biotec, 1:200, Cat# 130-091-650); Anti-Human Glut1 (FITC, Clone # 202915, R&D Systems, 1:20, Cat# FAB1418F); Anti-Phosphorylated Human p38 MAPK (pT180/pY182) (PE-Cy7, Clone 36/p38, BD Biosciences, 1:20, Cat# 560241); Anti-Human p53 (FITC, Clone DO-7, BioLegend, 1:20, Cat# 645804); Anti-Human phospho-p53 (S15) (PE, Clone 261352, R&D Systems, 1:50, Cat# IC1839P); Anti-Human phospho-ATM (Ser1981) (PE, Clone 10H11.E12, BioLegend, 1:100, Cat# 651204); Anti-Human dimethyl-Histone H3 (Lys9) (AlexaFluor®488, Merck Millipore, 1:100, Cat# FCABS301A4); Anti-Human acetyl-Histone H3 (PE, Merck Millipore, 1:100, Cat# FCABS325PE); Anti-Human HLA-A2 (PE, Clone BB7.2, BioLegend, 1:200, Cat# 343306), and the viability probe 7-AAD (BD Biosciences) were used for T cell staining. LEAF purified anti-CD3 (clone HIT3a, BioLegend) and anti-CD28 (clone CD28.2, BioLegend) were used for overnight T cell stimulation. All dextramers and antibodies were used according to the manufacturer's instructions.

**Isolation of PBMCs and cell sorting**. PBMCs were isolated from fresh heparinized blood by Ficoll-Hypaque density gradient centrifugation and cryopreserved in liquid nitrogen until the day of analysis. After thawing, CD8+ T cells were isolated from PBMC with the CD8+ T Cell Isolation Kit (Miltenyi Biotec) and labeled with the viability probe 7-AAD, anti-CD3, anti-CD8, and HLA-Class I dextramers to identify antigen-specific T cell sub-populations. CD8-dextramer-positive T cells were subsequently sorted (yield 500–2000 cells per sample) with a FACSAria III Cell Sorter (BD Biosciences).

**Microarray data acquisition**. RNA was purified from dextramer-sorted CD8+ T cells with the Nucleospin RNA XS kit (Macherey Nagel, Duren, Germany), according to the manufacturer's instructions. Total RNA concentration was determined using a Nanodrop spectrophotometer and/or with the Ribogreen RNA quantification kit (ThermoFisher). RNA integrity was evaluated with the Bioana-lyzer 2100 traces system (Agilent Technologies). Total RNA was amplified with the Transplex Whole Transcriptome Amplification (WTA2) kit (Sigma) and purified with GenElute PCR Clean-Up silica spin-columns (Sigma) following manu-facturer's instructions. cDNA was labeled with the ULS Fluorescent Labeling kit (Agilent Technologies) and hybridized to 60-mer oligonucleotide whole-human-genome microarrays (Human GE 4 × 44K v2, Agilent Technologies), as indicated in the manufacturer's protocol. Microarray slides were scanned with an Agilent dual-laser DNA microarray scanner. The Agilent Feature Extraction software with default settings (user manual version 7.5) was used to obtain normalized expression values from the raw scans.

**Microarray data analysis**. The GeneSpring GX v11.5 software package (Agilent Technologies) was used for quality control checks, data normalization by the quantile method and initial microarray data analysis. Probes detectable in at least two-thirds of the replicates for each condition were retained for further analysis. ANOVA test with Benjamini–Hochberg correction for multiple testing (FDR ≤ 0.05) was used to select genes differentially expressed among all conditions. These were visualized as hierarchical clustering and used for principal component ana-lysis (PCA), to which 'mean-center' and 'scale to unit standard deviation' options were applied. Heat maps were generated using the MT4 Multi-Experiment Viewer[71].
Topological pathway analysis was performed separately on the T1 and T2 time-points datasets with Clipper method[72], a two-step empirical approach implemented in the GraphiteWeb tool[73], that uses Gaussian graphical models to identify pathways with means or covariance matrices significantly different between two experimental conditions. Specifically, following pathway structure conversion into nodes (i.e., genes) and edges (i.e., gene connections) graphical representation using biologically driven rules (KEGG and Reactome databases), mean and covariance matrices were calculated for chronic and self-limited patients at each time-point assuming a Gaussian distribution and taking into account the constraints imposed by the graph structure. After graph decomposition into cliques (i.e., small connected components), separate likelihood ratio tests were used to compare the means and covariance matrices of the two groups and to identify the most altered (i.e., dysregulated) portions (sub-networks) of each pathway. The p-values derived from this second step of analysis were corrected for multiple testing with the Benjamini-Hochberg method, thereby generating the corresponding q-values. Pathways with q-values ≤ 0.05 either in mean or covariance were deemed as significant and genes associated to individual dysregulated subnetworks were selected for further analysis.
Overrepresentation analysis was performed on log$_2$ expression data (all detected probes) using the Gene Set Enrichment Analysis (GSEA) software[74] to identify

molecular pathways significantly overrepresented among the upregulated and downregulated genes, and to compare the resulting expression profiles with those of previously published studies. To this end, we used the Molecular Signature Database v 6.0 (C2, canonical pathways; C5, Gene Ontology, GO, gene sets) available at http://www.broadinstitute.org/gsea/msigdb/index.jsp. Gene sets smaller than 15 or larger than 500 genes were excluded; gene sets passing this filter were considered as significantly enriched if the False Discovery Rate (FDR), calculated using Signal2Noise as metric and 1000 permutations of gene sets, was ≤0.1. The permutation type was set to 'gene set', instead of 'phenotype', as suggested for fewer than seven replicates; default settings were applied to all other options.

The leading-edge subset analysis was performed to identify the genes in the ranked list of a functional gene set that appear at, or before, the point where the running sum reaches its maximum deviation from zero (enrichment score)[74]. The genes within this subset can be interpreted as the most important in the enrichment of the functional gene set. Grouping of terms in categories was then performed by applying the Enrichment map in Cytoscape[75,76] to leading edge genes deriving from enriched terms with FDR ≤ 0.1. An edge-weighted force-directed layout method applied to the combined score attribute was used for graphical representation.

To identify candidate regulatory genes at T1 time point, leading-edge genes present in at least five of the eight categories were retained, and the Search Tool for the Retrieval of Interacting Genes (STRING v10.5) database was used to build the functional gene association network which was visualized by Cytoscape[76,77]. EpiFactors (http://epifactors.autosome.ru)[43], a web-accessible, manually curated database that provides information on human proteins and complexes involved in epigenetic regulation, was used to generate expression profiles of the main chromatin-regulator complexes.

In Fig. 6d the listed complexes are: APOB mRNA editosome also known as the Apolipoprotein B mRNA editing enzyme complex; CAF-1, Chromatin Assembly Factor-1; CoREST, REST Corepressor 1; CREST-BRG1, transcription activator BRG1; DUB, histone deubiquitination complex; Exosome, RNA exosome complex; FACT, histone chaperone that FAcilitates Chromatin Transcription; HAT, Histone acetyltransferases; HDAC, histone deacetylases; HMT, Histone Methyltransferases; Ino80, chromatin-remodeling ATPase INO80; ISWI, SWI/SNF-Related Matrix-Associated Actin-Dependent Regulator of Chromatin A5; L3MBTL1, Histone Methyl-Lysine Binding Protein; MLL, Histone-lysine N-methyltransferases; PcG and PcG-like, Polycomb Group complexes; PP, dephosporylation complex phosphatases; SRCAP, multiprotein chromatin-remodeling complex; SWI/SNF, SWItch/Sucrose Non-Fermentable nucleosome remodeling complex; and SWR, SWR1-like complex.

Expression data are available at NCBI GEO: GSE111449.

**Quantitative PCR**. Expression levels of a subset of modulated genes were independently determined by Taqman gene-expression assays (Life Technologies) using the same amplified cDNAs used for microarray analysis, as starting material. The selected genes included PDCD1 (assay Hs00169472_m1) and BATF (assay Hs00232390_m1); GAPDH (Hs02758991_g1) served as a loading and normalization control. The expression levels of nine additional genes—TBX21 (Hs.PT.56a.20216516), CTLA4 (5′UTR-EX1 custom: F: 5-TCCTTGATTCTGTGTGGGTTC-3, R: 5-TTTATGGGA GCGGTGTTCAG-3, probe: 5-ACACATTTCAAAGCTTCAGGATCCTGA-3), HeyL (Hs.PT.58.3223619), PCCB (Hs.PT.58.40514946), PGAP3 (Hs.PT.58.38411960), PRKAG1 (Hs.PT.58.40441800.gs), LAT (Hs.PT.58.39778311), AKT1 (Hs.PT.58.38430799.g), ZAP70 (Hs.PT.58.2646227.g), and ADCY4 (Hs.PT.58.39234436.g)—were determined by PrimeTime qPCR 5′ Nuclease Assays (IDT, Coralville, IA) using the relative quantification method; also in this case, GAPDH served as a reference housekeeping gene.

**Glucose uptake and GLUT1 assay**. Glucose uptake assays were done on PBMC stimulated overnight with coated anti-CD3 (10 μg/ml) and soluble anti-CD28 (2 μg/ml) for ex vivo analysis. To study inhibition by specific chemical compounds, PBMCs were stimulated for 40 h with virus-specific peptides. After specific stimulation, PBMCs were washed with PBS 1X to eliminate endogenous glucose and stained for 30 min at 37 ° C with the glucose analog 2-NBDG (2-deoxy-2-[(7-nitro-2,1,3-benzoxadiazol-4-yl) amino]-D-glucose, ThermoFisher; 40 μM) in glucose-free RPMI with 10% dialyzed FBS, and finally stained with the viability probe 7-AAD. Fluorescence generated by the glucose analog, which is proportional to glucose uptake, was measured with a FACSCANTO II multicolor flow cytometer and analyzed with DIVA (BD Biosciences) and FlowJo (Tree Star) softwares.

GLUT 1 assay was performed on PBMCs, after specific stimulation as described above, by using an anti-glucose transporter GLUT1 antibody (Glut1-FITC, R&D Systems). Cells were surface-stained, fixed with the fixation reagent-medium A (Nordic MUbio), then permeabilized with the permeabilization reagent-medium B (Nordic MUbio) and stained with the anti-glucose transporter GLUT1 antibody (R&D system), according to the manufacturer's instructions, and finally analyzed by flow cytometry.

**Glucose level measurement**. The glucose level was measured with the Glucose Assay kit (Beckman Coulter) in cell culture media from total CD8+ T cells purified with the CD8+ T Cell Isolation Kit (Miltenyi Biotec). CD8+ T cells were cultured

overnight with or without virus-specific peptides (NS3-specific or FLU-/EBV-/CMV-specific peptides). A Beckman Coulter AU5400 analyzer was used, in accordance with the manufacturer's instructions.

**Mitochondrial membrane potential and mitochondrial superoxide assays**. MMP was determined on PBMCs stimulated overnight with coated anti-CD3 (10 μg/ml) and soluble anti-CD28 (2 μg/ml) for ex vivo analysis by using the potentiometric probes JC-1 (Molecular Probes, Life Technologies). After surface staining, cells were incubated with JC-1 (2.5 μg/ml) for 15 min at 37 °C and protected from light before flow-cytometry analysis. Samples were then stained with the viability probe 7-AAD and finally acquired on a FACSCANTO II multicolor flow cytometer and analyzed with the DIVA software (BD Biosciences). The decrease in JC-1 fluorescence caused by co-treatment (15 min at 37 °C) with the protonophores valinomycin or carbonyl cyanide m-chlorophenyl hydrazine (CCCP) (Molecular Probes) served as a positive control for MMP depolarization. Dextramer-positive virus-specific depolarized cells were quantified by subtracting the percentage of FL1high/FL2low cells (JC-1 staining) detected in the unstimulated samples from the percentage of the corresponding cell subsets detected in the stimulated samples[23].

Mitochondrial superoxide levels were measured on PBMCs incubated overnight with coated-anti-CD3 (10 μg/ml) and soluble anti-CD28 (2 μg/ml) or with NS3-specific peptides in the presence or absence of the antioxidant resveratrol (10 μM). ROS levels were also measured after 10 days of culture with virus-specific peptides. After specific stimulation, PBMCs were washed, analyzed for viability with the with the LIVE/DEAD Far Red Dead Cell Stain Kit (ThermoFisher), subsequently stained for cell surface markers and then incubated with the MitoSOX Red dye (5 μM; ThermoFisher) for 15 min at 37 °C, according to the manufacturer's protocol, before flow cytometry acquisition[23].

**Cytoplasmic ROS (DHE, H₂DCFDA staining)**. Cytoplasmic ROS levels were determined on PBMC stimulated at the same culture conditions described for mitochondrial ROS. Cells were analyzed for viability with the LIVE/DEAD Far Red Dead Cell Stain Kit (ThermoFisher), stained for cell surface markers, then incubated with the superoxide sensitive dye dihydroethidium (DHE, 1 μM, Thermo-Fisher) or with the H₂O₂ sensitive dye 2′,7′-dichlorodihydrofluorescein diacetate (H₂DCFDA, 2 μM, ThermoFisher), according to the manufacturer's protocol, and finally analyzed with FACSCANTO II multicolor flow cytometer. Data were analyzed with DIVA (BD Biosciences) and FlowJo (Tree Star) softwares.

**Phosphoprotein staining**. Phosphoproteins and nuclear factors were measured on PBMC either unstimulated or incubated overnight with coated-anti-CD3 (10 μg/ml) and soluble anti-CD28 (2 μg/ml) or with NS3-/FLU-CMV-EBV-specific peptides in the presence or absence of the antioxidant resveratrol (10 μM).

For anti-phospho-ATM, anti-dimethyl-Histone H3 (Lys9), anti-acetyl-Histone H3 intracellular antibody detection, intracellular staining BD Phosflow™ Fix and Perm/Wash Buffer I (BD) was used, according to the manufacturer's instructions. For anti-phospho-p38, anti-total p53 and anti-phospho-p53 antibody detection, the intracellular staining Fixation/Permeabilization Solution Kit (Cytofix/Cytoperm BD) was used, according to the manufacturer's instructions.

Data were collected with a FACSCANTO II multicolor flow cytometer and analyzed with DIVA (BD Biosciences) and FlowJo (Tree Star) softwares.

**Seahorse analysis**. Seahorse experiments were performed on total CD8+ T cells purified with the CD8+ T Cell Isolation Kit (Miltenyi Biotec). Multiple stimuli (i.e, Flu-/EBV-/CMV-specific and NS3-specific HCV peptides, plus an unstimulated control) were applied to CD8+ T cells from each individual patient and each culture condition was analyzed in the same experimental session in order to limit possible batch effects. These were further minimized through the internal normalization of maximal to basal OCR and ECAR values that was applied to each sample. Specifically, purified CD8+ T cells were cultured overnight with or without virus-specific peptides and then seeded on Seahorse XFp culture miniplates (300,000 cells/well). Briefly, cells were plated on poly-D-lysine-(100 μg/ml) coated 8-well polystyrene Seahorse miniplates and assayed for OCR (pmol/min) and ECAR (mpH/min) at the basal level and after sequential addition of oligomycin (1 μM), FCCP (2 μM), and antimycin A/rotenone (1 μM/0.1 μM), according to the manufacturer's protocol. To determine the maximum glycolytic capacity (MGC) and glycolytic reserve (GR; i.e., the difference between MGC and baseline ECAR) values of basal and oligomycin-modulated ECAR were used. Specifically, oligomycin blocks mitochondrial complex V, providing indication of the amount of oxygen utilized for ATP synthesis; FCCP uncouples ATP synthesis from the electron transport chain by transporting protons across the inner mitochondrial membrane, thereby allowing the calculation of the maximal and the spare respiratory capacity; rotenone/antimycin A shuts down mitochondrial respiration by blocking complexes I and III, allowing to calculate the ATP production, the maximal (MRC), the spare respiratory capacity (SRC), the proton leak and the coupling efficiency of virus-specific CD8+ T cells. Results were obtained with the Agilent Seahorse XF Report Generator of the Wave 2.3 software, according to the manufacturer's instructions.

**Aggresome detection (ProteoStat staining).** For the detection of protein aggregates, after overnight PBMC stimulation with or without coated anti-CD3 (10 μg/ml) and soluble anti-CD28 (2 μg/ml), cells were surface stained and then the ProteoStat Aggresome Detection Kit (Enzo Life Sciences, New York, NY) was used according to the manufacturer's protocol, before flow cytometry acquisition[25]. Cells treated with 5 μM of proteasome inhibitor (MG-132) served as positive controls. Aggresome levels were quantified by subtracting ProteoStat median fluorescence intensity (MFI) in the unstimulated from the stimulated samples. Data were collected with a FACSCANTO II multicolor flow cytometer and analyzed with DIVA (BD Biosciences) and FlowJo (Tree Star) softwares.

**T cell expansion, inhibitory compound treatment, and cytokine production assays.** Short-term T cell lines were generated by 10 days PBMC stimulation with HCV-NS3 or with HLA-A2-restricted HCV peptides (each at the final 1 μM concentration) in the presence or absence of drug inhibitors. For ex vivo analysis of inhibitor activity, PBMCs were stimulated for 40 h with HCV-NS3 peptides. FLU, CMV and EBV peptides were used as controls. At the end of the culture period, cytokine determination (IFN-γ, TNF-α and IL-2) was performed by intracellular cytokine staining[23]. Specifically, brefeldin-A (10 μg/ml; Sigma-Aldrich) was added to the cells and left throughout the time of stimulation for ex-vivo assays and for the last 4 h of the 10-days cultures. After washing, cells were stained with anti-CD8, anti-CD4 and anti-CD3, fixed and permeabilized (FIX&PERM Cell Fixation and Permeabilization Kit, Nordic-MUBio) to allow intracellular detection of IFN-γ, TNF-α, and IL-2.

**Inhibitors tested.** The following compounds were used to restore intracellular signaling, metabolic functions and anti-viral activities: the kinase ATM inhibitor KU-55933 (tested at concentrations of 0.01–0.1 μM, Sigma), the p53 inhibitor Pifithrin-α (tested at concentrations of 10–30 μM, Sigma), the p38 inhibitor SB203580 (tested at concentrations of 0.01–0.1 μM, Sigma), the AMPK inhibitor Dorsomorphin (tested at concentrations of 0.1–1 μM, Sigma), the EZH2 inhibitor GSK126 (0.1–0.05 μM, Selleckchem), the EZH2 inhibitor EPZ005687 (0.1–0.05 μM, Selleckchem), the G9A inhibitor UNC0638 (0.1–0.5 μM, Selleckchem), the G9A inhibitor BIX01294 (0.5–1 μM, Selleckchem), and Resveratrol[38,39] (5–10 μM, Sigma). Cell toxicity by the inhibitory compounds was assessed by analyzing increased cell death by flow cytometry.

**Flow cytometry measurement controls.** As a negative control (in gray in Supplementary Fig. 15) for all cytofluorimetric measurements, a sample without the staining of interest has always been acquired in order to define the level of autofluorescence related to background signals. This provides a measure of the fluorescence spread from the other staining parameters into the channel of interest and allows to determine accurately the threshold for positive staining, in accordance with accepted guidelines[78].

**Statistical analysis.** The GraphPad Prism 6.0, JASP 0.10.2 (https://jasp-stats.org/) and Jamovi 1.0.7 (https://www.jamovi.org) softwares were used for statistical analysis. The $F$ test of variance was applied to the different groups; normality distribution of data was tested by the Kolmogorov-Smirnov test. Differences between multiple patient groups have been evaluated by Kruskal-Wallis non-parametric test and the $p$-values have been calculated and corrected for pair-wise multiple comparisons, according to the methods of Dunn.

Moreover, to detect differences in treatments the Friedman non-parametric test has been applied and corrected for pair-wise multiple comparisons, according to the Conover test. In the experiments with the different inhibitory compounds, the significance of the fold change (FC) upon drug treatments was evaluated by the Wilcoxon signed-rank tests (different from a theoretical median = 1).

**Reporting summary.** Further information on research design is available in the Nature Research Reporting Summary linked to this article.

## Data availability

All transcriptomic data from this study have been deposited in the National Center for Biotechnology Information Gene Expression Omnibus (GEO) and are accessible through the GEO Series accession code GSE111449. Raw data are provided as Source Data File and all other relevant data are available from the corresponding author on request.

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

## Acknowledgements

We thank Prof. Riccardo Bonadonna (Unit of Endocrinology, University of Parma, Italy) for helpful discussions and the Microarray Facility at the University of Ferrara-Italy (http://ltta.tecnopoloferrara.it/bioinformatica.php) for help in the initial phase of bioinformatic analysis. Access to the CoreLab (Azienda Ospedaiero-Universitaria of Parma, Italy) and to the CRBa (University of Bologna, Italy) central facility instrumentations and technical advice by Dr. Vilma Mantovani, Dr. Francesca Bianchi, Dr. Rosalia Aloe, and Dr. Roberta Musa are also gratefully acknowledged. This work also benefited from support by the Biotechnology Interuniversity Consortium (CIB) and the bioinformatics expertize framework available within the COMP-HUB Initiative, funded by the 'Departments of Excellence' program of the Italian Ministry for Education, University and Research (MIUR, 2018–2022). This work was supported by the European Commission grant HepAcute (FP7-HEALTH-2010), by a grant from Regione Emilia-Romagna, Italy (Programma di Ricerca Regione-Università 2010–2012; PRUa1RI-2012-006 to C.F., by a FIRB grant (RBAP10TPXK to C.F.) from the Italian Ministry of Education, University and Research (MIUR to C.F.), by a grant from "Agence Nationale pour la Recherche sur le SIDA et les hepatites virales" (ANRS) to M.L. (n. ECTZ66014) and by a grant from the Agence National de la Recherche (ANR@TRACTION) to M.L.

## Author contributions

V.B.: study design, design, and execution of the experiments; data acquisition, analysis, and interpretation; writing of the paper; P.F.: execution of experiments and statistical analysis; B.M.: microarray data handling, including GSEA and network analysis; G.A., A.F., G.F. and F.G.: execution of experiments; C.R.: microarray data handling, including topology-based analysis; M.F.: microarray data analysis; G.P.: statistical analysis; C.B., M.R., A.V., A.P., A.Z. and C.M.: analysis of data from cell-based assays and patients' characterization; A.O., E.N., M.P. and M.M.: recruitment and characterization of patients; G.M.: critical revision of the paper; M.L. and S.O.: study design, data mining, and interpretation, writing and revising the paper; C.F.: study concept and supervision, data analysis, and interpretation, writing and revising the paper and funding retrieval.

## Competing interests

M.L.: Consultant for Gilead, Jansen, BMS, Arbutus, Galapagos, Assembly Pharma, Sanofi/Aventis. C.F.: Consultant for Gilead Srl, Abbvie, Arrowhead, Humabs, Abivax, MSD; grants from Gilead Srl, Bristol Squibb, Roche Spa, Abbvie, Janssen Cilag.
