## [Peer Review File · Nature Communications]

Reviewers' comments:

Reviewer #1 (Remarks to the Author):

The findings of this manuscript by Barili et al can be summarized as follows: the transcriptional signatures of HCV-specific CD8+ T cells differ between self-limited and chronic patients at early and late time points of infection suggesting specific, outcome-dependent genetic patterns. A general transcriptional upregulation indicates a metabolic dysregulation in early chronic patients correlated with an upregulation of p53 and ATM, known regulators of glycolysis and oxidative phosphorylation. Inhibition of p53, ATM and related targets of the concerning pathway restores the glycolytic activity and antiviral capacity of the CD8+ T cells of the early chronic patients. Furthermore, a downregulation of T cell-specific and other cellular processes in the HCV-specific CD8+ T cells of the late-stage chronic patients is correlated with diverse cellular and functional impairments. A link of these impairments to epigenetic silencing has been demonstrated by inhibition of selected repressive, transcriptionally upregulated HMTs. The observed restoration of the metabolic and antiviral functionality of the exhausted CD8+ T cells can also be achieved by inhibition of the constantly upregulated p53.

Of note, it has recently been published by Wolski et al. (Immunity 2017) that the metabolism of HCV-specific CD8+ T cells early in chronic infection is dysregulated although in the current manuscript of Barili et al, they clearly deepened the understanding of the dysregulated metabolism in HCV-specific CD8+ T cells in chronic infection by using a large repertoire of functional assays. By this, the involvements of upregulated p53 and HMTs in the altered expression pattern leading to exhausted CD8+ T cells has been clearly underlined.

However, several concerns with respect to data interpretation and presentation are listed below:

Major points:

For a better evaluation of all statements, the separation between the data of the self-limiting cohort and the chronic cohort should be clearly stated.

Can antigen recognition of the used CD8+ T cells be guaranteed and is a viral escape excluded?

In all assays, the groups stated at the beginning (T1: self-limiting, T1: chronic-evolving, T2: resolved and T2: chronic) should be included. This would allow for a better evaluation of the data, for instance concerning the assays investigating altered methylation patterns in chronically infected patients, the self-limiting patients should be severed as a control group, as well as the late-phase T2 samples should serve as control group for the metabolism assays.

In all figures containing MFI data, the MFI of a negative control would be helpful to better evaluate the results, especially concerning altered epigenetic patterns depicted in figure 5. Additionally, regarding the same aspect, representative plots would be helpful.

Furthermore, to evaluate interindividual differences among the patient groups, please show for the relevant figures the differentially expressed genes of all patients not only the average of each group.

Concerning figure 1a, the graphical representation is not comprehensible and the overlap between T1 and T2 resolved and the healthy control is not clearly evident.

Please integrate the supplementary figure 2 in the main text or integrate the corresponding paragraph into another one.

Please show the data proving that the difference in glycolytic efficiency is not caused by lack of T cell activation.

Concerning the epigenetic patterns in figure 5e, please consider not only the proposed negative MFI control but also the investigation on CD8+ T cell bulk level and other controls as Flu-specific CD8+ T cells.

Please state if there is an effect of the HMT inhibitors leading to upregulated TCR or increase in dextramer-specific CD8+ T cells which can be assumed based on the representative dot plots in supplementary figure 5b.

Due to the unique property of DAA treatment and the here proposed possibility of a restoration of exhausted HCV-specific CD8+ T cells, experiments regarding the metabolic dysregulation and the potential effects of the inhibitors targeting ATM, p53, AMPK, p38 on HCV-specific CD8+ T cells from chronic patients after/during DAA-mediated treatment would be of great importance to estimate the efficiency of the here proposed strategies.

Minor points:

Please use a consistent nomenclature for CD8+ T cells throughout the publication.

Please use consistent nomenclature regarding the declaration of sample numbers throughout the publication.

Regarding supplementary fig. 5, the figure legend is not separated between figure S5a and S5b.

Reviewer #2 (Remarks to the Author):

Barili et al. have analyzed HCV-specific CD8 T cells during acute HCV infection with different outcomes, using microarrays and a series of functional assays. They find early transcriptional differences between T cells in resolving versus chronic infection, impaired glycolytic and mitochondrial functions, and later broad impairment of T cell metabolism and functionality. This

exhausted phenotype can be improved in vitro by histone methyltransferase inhibitors. They conclude that the findings are of importance for development of treatments for patients failing DAA HCV therapy and beyond for all pathologies associated with dysfunction.

Overall, I fully agree with the authors that the scenario of acute HCV infection, with a clear dichotomy in outcome represents a unique opportunity to study the difference between the generation of fully functional memory versus exhausted T cell immunity in humans. This opportunity to analyze and directly compare in a human model infection two dichotomous fates of the T cell response that are, individually, also observed in other diseases, makes this study of potentially very high impact. In comparison, the mentioned issue of DAA failure is less relevant, but this does not lessen the overall importance of this kind of study.

Overall, the manuscript reports about a large number of different experiments, with several intriguing findings. I especially appreciate the attempt to identify/test specific mechanisms in a human model disease. Having said that, I think the manuscript as is has several weaknesses related to several of the assays, the lack of important controls, limited information about experiments and result details, and is overall a bit convoluted. I also find the discussion too extensive and speculative. Nevertheless there is a lot of interesting data here and I have the following questions and suggestions for the authors, hoping that they will allow to further improve manuscript :

1) There is extremely little to no information about the patients that were studied. There must be more information about the timepoints selected for analysis. Roughly when after estimated time of infection were patients studied? Based on what information? What is the time difference between samples categorized as T1 and T2? This is of critical importance, especially given the small sample numbers. T1 being "immediately after clinical presentation" is extremely vague, as is "later stages of infection" for T2. What were the ALT values and viral loads at the timepoints studied? What about symptoms? What is the male/female ratio? All these clinical factors can have strong impact on the data (viral load at T2 being the most obvious).

2) It should also be made clear which patients are studied in which experiment, since it seems that almost every experiment was performed using cells from different patients. How similar were the different patients clinically? And should one not compare for each experiment whether differences in the patient groups (e.g. viral load, time post infection) were comparable to the patients/samples in other experiments?

3) There is very limited detail about the analysis of the transcriptional data and its analysis. The tables do not fully allow to judge the strength of the observations and to compare results, which is especially an issue given the very small sample sizes in some of the groups (as low as 4). Throughout the manuscript, later experiments are often justified by the transcriptional data, but in some cases I could not find the mentioned results in the tables and it is not clear whether the selected pathways and molecules were actually selected because of them being highly significant top hits in the transcriptional analysis, or whether they were selected based on the existing literature and then

justified by some observed difference in the transcriptional data. Examples are the GSEA results, which are impossible to interpret from the table alone. Certainly the adjusted FDR/q values are almost universally not very strong. How were the best/most promising pathways picked here? In addition, the authors mention upregulation of inhibitory receptors PD-1, CD160 etc supposedly shown in suppl table 2, but this table does not contain any individual gene expression data. Based on gene expression data in suppl table 1, these genes are not downregulated in chronic samples at T2 (unless the results in table 1 are logarithmic, which is not indicated).

4) The experiments in figure 2b, while significantly different between groups, show results with a wide overlap between chronic and resolving cells. Did the authors test whether the results were correlated with other parameters than outcome, most importantly viral load? Also, it would be nice to see some original flow data and to see the results for total CD8 populations.

5) According to the methods section the seahorse experiments from figure 2 were performed using total CD8 T cells, not specific cell populations. Would this not lead to a major impact of the relative population size of the specific cells within the total CD8s, with the populations being much smaller in chronic evolution, especially compared to EBV/CMV? And should unstimulated cells really respond as shown here, with almost as much as a response than stimulated healthy memory cells? And why are no results shown for acute resolving HCV infection, this being the critical comparison to chronic HCV.

6) In figure 2e, the results between chronic and resolving patients are connected with lines. How can these data from different patient groups be paired?

7) The PD-1 data in suppl figure 3b are rather surprising. Usually PD-1 expression during the acute phase is extremely high in all HCV-specific CD8 T cells irrespective of outcome. In contrast, flu-specific cells are usually PD-1 negative. Here the differences are extremely small and resolving patients show no PD-1 upregulation at all. Overall the PD-1 MFI values are so low that one wonders whether any cells were actually positive. These are extremely unusual results and impossible to interpret without seeing the staining for bulk CD8 T cells in dot plots and seeing any clinical context for the patients, such as time after infection and viral load.

8) The results in figure 3 are not really convincing. First, differences are rather subtle, and in 6/8 of the assays flu results are not different from those for chronic HCV. I am also not convinced that all of the p values are correct (most notably the left panels in 3b and 3d). Finally, are the differences for example for p53 really biologically relevant? These values look similar to the PD-1 staining (very low MFI, <300 ex vivo) and total CD8 results should be shown in dot plots. Similar issues apply to the phosphoflow staining, which is also lacking the data from unstimulated cells as key control. Usually phosphoflow is reported showing both unstimulated and stimulated cells together, and differences tend to be substantial.

9) The data in figures 4 and 6 are very interesting, but difficult to evaluate without control experiments using T cells from resolving HCV and flu. Are all these inhibitors only affecting chronic cells?

10) The results in figure 5a, where these pathways selected for being the most differentially regulated between T1 and T2, or what was the reason to select these? From the results as presented it is not clear which pathways were the most significant by transcriptional analysis.

Reviewer #3 (Remarks to the Author):

In this paper the authors compared T cells from HCV infected patients that were destined to have a chronic infection involving T cell exhaustion with ones that would resolve. They compared gene expression in the groups and also compared different cell functions including glycolysis and glucose uptake, oxidative phosphorylation, IFN and cytokine production, ROS levels, ATM and p53 phosphorylation and activation. It was found that chronically evolving T cells had increased ROS and ATM and p53 activation, reduced glycolysis and oxphos, and reduced anti-tumor type responses compared to cells where infection would resolve. They found an overall decreased transcription in chronically evolving T cells coincident with increased H3K9me2 repressive transcription mark. They showed HMT inhibitors of G9a and EZH2 which should H3K9me2 and H3K27me3 levels, also restored anti-infection phenotypes of the T cells and inhibitor of p53 also did. The authors suggest inhibitors such as these to increase T cell activity to fight HCV infections and possibly also cancer. Overall I thought the paper was written very well and was quite good. A strength of this study was the use and comparison of T cells from early and late infections that do and don't resolve. There are some questions;

1) Some of the findings or related findings have been made already showing p53 loss enhances anti-tumor activity of T cells associated with increased glycolysis and cytokine production and antitumor activity in a mouse study (e.g. Banjeree from Cancer Research Paper).

2) The authors mention H3K27me3 in the text but data in Fig 5 only shows H3K9me2. Did they look at H3K27me3 levels, or other H3K9 methylations?

3) Do they have evidence the H3K9 methylation levels were reduced in Fig 6.

Reviewer #4 (Remarks to the Author):

In this manuscript, Barili et al, by applying sophisticated transcriptomic, metabolic, biochemical and bioinformatic tools to study a well-defined cohort of HCV infected patients, demonstrated that chronic HCV results from profound perturbations of transcriptional, metabolic, and signaling pathways that are associated with effective immune response against viral infection. This study

applies state-of-the-art technologies and the data generated and analyses conducted are comprehensive and convincing. The findings are consistent with what have been reported in other model systems of chronic viral infection and ineffective immune response. Overall, the paper, while being somewhat descriptive and lacking mechanistic insight, provides rather valuable information to the scientific community regarding chronic viral infection and immune response.

The major comment by this reviewer is the timing of the samples. The T1 point, so-called the early samples, were obtained during the ALT peak of acute infection. As reported in the study, the T cells from those with acute HCV chronic evolution already demonstrated early markers of immune dysregulation, which then progressed to a phenotype of full immune exhaustion or nonresponse during the chronic phase (T2). It is the opinion of the reviewer that samples obtained prior to this point (during the acute asymptomatic infection stage) are essential to unravel the mechanism that lead to the described immune and metabolic dysfunctions. Also it is not clear whether the same patients were followed for the T1 early vs T2 late phase. Longitudinal study of same patients would have been much more compelling and eliminated much of the individual variations. Otherwise, this paper merely describes the natural evolution of immune exhaustion leading to ineffective antiviral immune response and chronic viral infection.

Response to the Reviewers' comments

Reviewer #1

Major points:

1. For a better evaluation of all statements, the separation between the data of the self-limiting cohort and the chronic cohort should be clearly stated.

We agree with the Reviewer and in the revised manuscript data derived from self-limiting and chronic patient cohorts have been more clearly separated and described (see for example Fig. 1a, Fig.2c and 2f, and highlighted in yellow the legend to Figs. 1, 4, 5 and 6).

2. Can antigen recognition of the used CD8+ T cells be guaranteed and is a viral escape excluded?

We thank Reviewer 1 for this comment. As a first response to a possible 'viral escape', we would like to point out that one of the HCV-specific peptides we used to capture and purify HCV-specific CD8+ T cells -namely, the HCV NS3 CINGVCWTV peptide epitope- was recognized by CD8+ T cells from T1/early chronically evolving acute patients in 7 out of 8 cases at the T1 time-point and in 6 out of 7 cases at the T2/late chronic time-point. Previous studies (e.g., Söderholm et al, Gut 2006, 55: 266-274)¹ indicate that this epitope does not tolerate mutations at five positions of its sequence, because they strongly impair protease activity and RNA replication. This suggests that viral fitness strongly limits the variability of this epitope, thus making selection of escape variants as major viral species in infected hosts most unlikely.

Second, to further address the escape issue, we retrieved all HCV polypeptide sequences from the Uniprot database (a total of 2332 sequences for the Hepatitis C virus genotype 1 NS3-NS5 polyprotein, UniRef100 database, October 2018 release), using the NP_671491.1 NS3-NS5 polypeptide as a query sequence with an e-value threshold of 10^{-4} , and analyzed the percentage of amino acid substitutions within the entire NS3-5 sequence. In the regions corresponding to the four epitopes (NS3 1073-1081, NS3 10406-1415, NS4B 1992-2000 and NS5 2594-2602) we used to isolate virus-specific CD8+ T cells by cell-sorting, the substitution rate was always lower than 5% and practically negligible for peptides NS3 1073-1081 and NS4B 1992-2000 (0.89% and 0.02%, respectively), (see below for an outline of this analysis).

Third, despite detectable *ex vivo* frequencies of dextramer+ CD8+ T cells specific for the peptides we sequenced and used for sorting, their capacity of expansion was always very low even if the sequenced peptides perfectly matched the sequence of the infecting virus, in line with the expected behavior of exhausted T cells.

Finally, we also followed-up to this issue by sequencing viral RNA in the patient sera which were still available from 5 of the 8 chronically-evolving acute patients at the T1/early time-point,

and from 3 of the 7 chronic patients (T2/late time-point) in order to exclude the presence of variant/escape sequences in the samples we used for transcriptome analysis (see Urbani S. et al, Journal of Immunology 2005 for a description of the HCV sequencing method)². Sequencing of two epitopes (HCV NS3 CINGVCWTV and HCV NS4 VLSDFKTWL), which were the only ones recognized by the five chronically evolving acute (T1/early time-point) and by the 3 tested chronic patients (T2 time-point), did not reveal any mutation. The fact that the peptides utilized for HCV-specific CD8+ T cell capture do indeed match the autologous viral sequences rules out viral escape in these patients. However, due to sample limitations, sequencing analysis could not cover the entire set of patients subjected to transcriptome analysis; thus, we would prefer not to include these data in the revised manuscript, but we are open to add them in the revised manuscript, in case of Reviewer's request.

Peptides used to isolate virus-specific CD8 T cells for transcriptomic analysis:

CINGVCWTV NS3 aa: 1073-1081, average substitution = 0.89%
 KLVALGINAV NS3 aa: 1406-1415, average substitution = 4.58%
 VLSDFKTWL NS4B aa: 1992-2000, average substitution = 0.02%
 ALYDVVTKL NS5 aa: 2594-2602, average substitution = 1.45%

3. In all assays, the groups stated at the beginning (T1: self-limiting, T1: chronic-evolving, T2: resolved and T2: chronic) should be included. This would allow for a better evaluation of the data, for instance concerning the assays investigating altered methylation patterns in chronically infected patients, the self-limiting patients should be served as a control group, as well as the late-phase T2 samples should serve as control group for the metabolism assays.

As suggested by the Reviewer, we expanded the number of patients tested in the different assays, as illustrated in the table below (see also **Supplementary table 3a and 3b**).

PATIENTS (n=144→207)	Patient number (old manuscript)	Patient number (new manuscript)	Patient number (HLA-A2+)
Acute HCV Chronic.-evolving (T1)	30	38	8→19
Acute HCV Self-limited (T1)	17	25	13→16
Chronic HCV (T2)	68	84	31→42
Resolved HCV (T2)	15	20	12→13
DAA-treated Chronic HCV (T2)	-	20	0→9
Healthy controls	14	20	13→20

This was not an easy task due to the following experimental hurdles:

- i) the difficulty in recruiting new patients, due to the fact that T1/early acute HCV patients undergo early pharmacological therapy prior to the definition of the infection outcome;
- ii) the need to select only HLA-A2 positive individuals because metabolic and epigenetic assays must be performed on virus-specific dextramer positive CD8+ T cells;
- iii) the relatively large number of PBMCs needed for our experiments, because each assay (requiring at least 4 million PBMCs) must be performed separately due the different staining procedures utilized for individual measurements (e.g. glucose uptake, GLUT1 expression, JC1, ROS and proteasome stainings).

Despite these drawbacks, we managed to perform the requested longitudinal analysis of all patient groups by flow cytometry using previously stored cell samples. The results of these additional experiments are illustrated in the **new Supplementary Figure 6 and 7** and are discussed in the main text at page 13 lines 5 and 23, respectively. The new data indicate that T2/late chronic patients display the highest glycolytic activity, as measured by glucose uptake and GLUT1 expression (**Supplementary Fig. 7a**). This metabolic profile was observed previously also in chronic HBV infection, where exhausted HBV-specific T cells, which were unable to meet their energy demands by oxidative phosphorylation due to dysfunctional mitochondria, were found to be strongly dependent on glycolysis (Schurich *et al*, Cell Rep. 2016 Aug 2;16(5):1243-1252; Fiscaro *et al*, Nat Med. 2017 Mar;23(3):327-336)^{3, 4}. Similarly to exhausted HBV-specific T cells from chronic HBV patients, also T2/late chronic HCV patients' PBMCs featured functionally deficient mitochondria with an increased ROS production, already apparent in the T1/early acute phase of chronically evolving infection (**Supplementary Fig. 7c**). A chromatin dysregulation mark was also observed at an early stage of infection (T1/early) at both methylation and acetylation levels (**Supplementary Fig. 7f**). In contrast, no proteasomal dysfunction could be detected at the T1/early stage of chronically-evolving acute infection (**Supplementary Fig. 7e in red**), and this was associated with a concomitant transcriptional upregulation of multiple proteasome-related genes (**Supplementary Fig. 2b, Fig. 5a** and T1 Sheets in **Supplementary Table 2**).

New Seahorse experiments were performed also in T1/early acute self-limited patients and the resulting data were added to **Supplementary Fig. 3** (panels **3d, 3e and 3f**) and discussed in the main text (page 10, line 12). In particular, NS3-HCV-stimulated CD8+ T cells from T1/early

acute self-limited patients displayed higher Extra Cellular Acidification Rate (ECAR) and Oxygen Consumption rate (OCR) values compared to chronically-evolving patients (**Supplementary Fig. 3e to 3f**), at both basal and maximal levels as shown by the cell-energy phenotype plots in **Supplementary Fig. 3 (panel 3d)**.

4. In all figures containing MFI data, the MFI of a negative control would be helpful to better evaluate the results, especially concerning altered epigenetic patterns depicted in figure 5. Additionally, regarding the same aspect, representative plots would be helpful.

According to the reviewer's request, negative control MFI data have been included as fluorescence-minus-one (FMO) stainings in all flow cytometry experiments.

For this purpose, a sample without the staining of interest has always been acquired in order to define the level of autofluorescence related to background signals. This provides a measure of the fluorescence spread from the other staining parameters into the channel of interest and allows to determine accurately the threshold for positive staining, in accordance with accepted guidelines (e.g., Cossarizza *et al.*, Guidelines for the use of flow cytometry and cell sorting in immunological studies. Eur J Immunol. 2017 Oct;47(10):1584-1797)⁵.

Following-up to the Reviewer's request, some of these negative control data have been included in **Supplementary Fig.11** of the revised manuscript, where negative controls allow to clearly identify positive signals, especially for epigenetic marks.

Some representative dot-plots related to glucose uptake and GLUT1 expression measurements performed on dextramer (+) (*black*) and on total CD8+ T cells (*green*) are illustrated below for Reviewer's perusal, but due to space constraints we would prefer not to incorporate them in the revised manuscript.

5. Furthermore, to evaluate inter individual differences among the patient groups, please show for the relevant figures the differentially expressed genes of all patients not only the average of each group. Concerning figure 1a, the graphical representation is not comprehensible and the overlap between T1 and T2 resolved and the healthy control is not clearly evident.

As suggested by the Reviewer, the heat-map in **Fig. 1b** now shows gene expression data referred to single patients. It should be noted, however, that the results of GSEA analysis were already presented as single patient heat-maps in the original manuscript (see **Supplementary Fig. 2a** and **Fig. 5b**).

Regarding the principal component analysis in **Fig. 1a**, the different patients' groupings were created according to the statistical clustering tree shown in **Fig. 1b** (generated by the Genespring software, with the Hierarchical Clustering Algorithm plus Squared Euclidean Similarity Measure plus Complete Linkage Rule as settings) where both acute patient groups are partially overlapping at T1/early and resolved patients (T2/late) are statistically closer to FLU healthy controls than HCV chronics (T2/late). To further improve figure readability, especially with regard to the stage of infection, patient groups in **Fig.1a** have been renamed.

Although the original graphical form of **Fig. 1a** has been maintained in the revised manuscript, a different graphical representation of the same data is reported below for Reviewer's evaluation. We are open to incorporate this alternative PCA data representation in the revised manuscript, in case the Reviewer finds it more understandable.

6. Please integrate the supplementary figure 2 in the main text or integrate the corresponding paragraph into another one.

According to the reviewer's comment, the paragraph relative to the **Supplementary Fig. 2** has been integrated into the following paragraph (please, see main text on page 8, line 21).

7. Please show the data proving that the difference in glycolytic efficiency is not caused by lack of T cell activation.

Additional dedicated experiments have been conducted to further address this issue. These included the measurement of cytokine production, which was found to be increased in HCV peptide-stimulated compared to unstimulated cultured cells, performed in parallel to Seahorse analyses. The results of these control experiments assessing HCV-specific T cell activation upon peptide stimulation (only mentioned as ‘data not shown’ in the previous manuscript) are now displayed in **Supplementary Fig. 3b** and discussed in the text (page 9, line 23), indicating that the observed difference in glycolytic efficiency is not merely due to a lack of T cell activation. As revealed by these experiments, IFN- γ production is strongly induced by FLU-, EBV- and CMV-specific peptides and a less intense but still highly significant activation is also induced by HCV-specific peptides. These data (some of which are shown below as representative examples) clearly demonstrate that overnight stimulation with FLU or HCV peptide mixtures prior to Seahorse analysis activates not only FLU- but also HCV-specific CD8⁺ T cells, thus allowing to link metabolic changes with virus-specific T cell activation.

8. Concerning the epigenetic patterns in figure 5e, please consider not only the proposed negative MFI control but also the investigation on CD8+ T cell bulk level and other controls as Flu-specific CD8+ T cells.

As shown in **panel A** below, MFI data points measured in total CD8+ T cells are quite scattered in all patient groups. Moreover, a different plot distribution of virus-specific and total CD8+ T cells is apparent in **panel B**, where representative MFI histograms for these two different CD8+ T cell populations are shown in *blue/brown* and *black*, respectively. As mentioned in our response to points #3 and #4 above, the overall analysis has been further expanded and now also includes the T1/early time-point for both acute patients cohorts (see the new **Supplementary Fig. 7f**).

A

B

The negative control data shown in **panel B** have been incorporated into the revised manuscript (**Supplementary Fig. 11**).

9. Please state if there is an effect of the HMT inhibitors leading to upregulated TCR or increase in dextramer-specific CD8+ T cells which can be assumed based on the representative dot plots in supplementary figure 5b.

To address this Reviewer's comment, we further investigated TCR expression after HMT inhibitors treatment. TCR expression was analyzed by measuring dextramer MFI in the same experimental set-up in which we previously observed a significant, HMT inhibitor-dependent increase in dextramer-positive cell frequencies. As shown below, only G9a HMT inhibition resulted in a significant increase of TCR expression (Wilcoxon signed rank test), while no appreciable effect could be detected in the case of EZH2 inhibitors. Thus, HMT inhibition appears to mainly affect CD8+ T cell expansion and, as shown in **Supplementary Fig. 8b**, it leads to a generalized and significant increase of HCV-specific CD8+ T cell frequency.

10. Due to the unique property of DAA treatment and the here proposed possibility of a restauration of exhausted HCV-specific CD8+ T cells, experiments regarding the metabolic dysregulation and the potential effects of the inhibitors targeting ATM, p53, AMPK, p38 on HCV-specific CD8+ T cells from chronic patients after/during DAA-mediated treatment would be of great importance to estimate the efficiency of the here proposed strategies.

To address this relevant but quite demanding question, we enrolled a new group of 20 DAA-treated chronic HCV patients (see **Supplementary Table 3b**) and performed metabolic as well as HMT/p53 inhibition experiments before and after DAA treatment. ATM, AMPK and p38 inhibition was not tested because these intracellular transducers appear to be up-regulated only at the T1/early but not at the T2/late time-point. Results included in the new **Fig. 7a** show that glucose import, albeit significantly reduced after DAA treatment and HCV clearance (end of therapy-EOT), did not reach the levels measured in spontaneously T2/late resolved patients (**first graph from the top**). Similarly, and in keeping with the PD-1 expression pattern (**Fig 7a, third graph from the top**), there was a recovery of mitochondrial depolarization in T2/late HCV-specific CD8+ T cells from some but not all patients at EOT (**Fig 7a, second graph from the top**). The repressive H3K9 di-methylation mark was also reversed to different extents in virus-specific CD8+ T cells from T2/late DAA-treated patients, and at EOT it was overall significantly higher than in spontaneously T2/late resolved patients and healthy controls (**Fig 7a, bottom graph**). Instead, a fairly strong and consistent recovery was observed for proteasomal activity (**Fig.7a, fourth graph from the top**). Thus, DAA treatment and virus clearance does not appear to afford a complete reversion of the metabolic and epigenetic alterations associated to virus-specific CD8+ T cells from T2/late chronic patients - an observation that might explain the variable levels of recovery of anti-viral responses previously reported for HCV-specific CD8+ T cells from T2/late DAA-treated patients (Martin *et al.*, J Hepatol. 2014 Sep;61(3):538-43; Callendret *et al.*, Hepatology. 2016 May;63(5):1442-54; van der Ree *et al.*, Antiviral Res. 2017 Oct;146:139-145)^{6,7,8}.

To further investigate the impact of DAA therapy on the functionality of HCV-specific T cells from T2/late chronic patients (**Supplementary Table 3b**), we tested the modulatory activity of four different histone methylation inhibitors (the G9a inhibitors UNC0638 and BIX01294, and the EZH2 inhibitors GSK126 and EPZ005687) plus the p53 inhibitor pifithrin- α . Interestingly, there was a significant response to the inhibitors - measured as number of single- and double-positive IFN- γ /TNF- α -producing CD8+ T cells upon peptide stimulation under both *ex vivo* conditions (**Fig.7b, left-side-graphs**) and after short-term culture (**Fig.7c, left-side graphs**) - when they were applied right before treatment. In contrast, a markedly reduced efficacy of the same inhibitors was observed when they were applied to CD8+ T cells collected after therapy withdrawal (i.e., at EOT) (**Figs. 7b and 7c, right-side graphs**), even if a significant increase was evident especially after 10 days of culture (**Fig. 7c, right-side graphs**).

All the above results are illustrated in the **new Fig. 7** and briefly described in the main text on page 16, line 13.

Minor points:

Please use a consistent nomenclature for CD8+ T cells throughout the publication.

The CD8+ T cell nomenclature was made consistent throughout the text.

Please use consistent nomenclature regarding the declaration of sample numbers throughout the publication.

We added the patient numbers in all figures where the results were not represented as single-patient dots.

Regarding supplementary fig. 5, the figure legend is not separated between figure S5a and S5b.

This formatting mistake has been corrected.

Reviewer #2

1) There is extremely little to no information about the patients that were studied. There must be more information about the timepoints selected for analysis. Roughly when after estimated time of infection were patients studied? Based on what information? What is the time difference between samples categorized as T1 and T2? This is of critical importance, especially given the small sample numbers. T1 being “immediately after clinical presentation” is extremely vague, as is “later stages of infection” for T2. What were the ALT values and viral loads at the timepoints studied? What about symptoms? What is the male/female ratio? All these clinical factors can have strong impact on the data (viral load at T2 being the most obvious).

According to the reviewer’s request, the clinical features of the 187 patients enrolled in our study have been detailed in the new **Supplementary Tables 3a** and **3b** and better described in the ‘Methods’ section. Some explanatory notes regarding the specific issues raised by the Reviewer are reported below.

- a. *Estimated time of infection*: patients were assigned to the T1/early time-point based on highly stringent and widely accepted clinical criteria: most patients were enrolled within 3 months (a few of them 3 to 6 months) from the ALT peak (at least 10-fold above the upper limit of the normal ALT range) and they were all HCV-RNA positive. In particular, all transcriptome analyses were performed on samples collected within the first month. Details on clinical parameters have been introduced in the new **Supplementary Table 3a**. Without evidence on the timing of anti-HCV seroconversion prior to clinical presentation and in the absence of specific risk factors, a reliable estimate of the time of infection cannot be made.
- b. *Time difference between samples classified as T1 and T2*: for patients with a self-limited acute infection, samples designated as T2/late were obtained 8 to 18 months after the ALT peak (generally more than 12 months), at a time when all patients were HCV-RNA negative. T2/late samples in the case of a chronically evolving infection were derived from patients followed in our clinic for at least one year after the first detection of ALT elevation; they were all HCV-RNA positive and for some (but not all) of them the time of acute infection was known. T2/late chronic HCV patients who underwent DAA treatment were followed-up for at least two years from the first detection of ALT elevation before starting therapy.

We agree with the Reviewer that the above data are of “critical importance, especially given the small sample numbers” used for transcriptional profiling. For this reason, in order to maximize the quality/reliability of our molecular and functional validation experiments, we enrolled the highest possible number of patients (a total of 187) belonging to the different clinical groups. They were distributed as follows: 38 and 25 patients in the T1/early acute phase of chronically-evolving and self-limited infections, respectively; 84 and 20 patients in the T2/late chronic and

resolved phase, respectively; plus 20 T2/late chronic HCV patients undergoing DAA treatment (see table below).

PATIENTS (n=144→207)	Patient number (old manuscript)	Patient number (new manuscript)	Patient number (HLA-A2+)
Acute HCV Chronic.-evolving (T1)	30	38	8→19
Acute HCV Self-limited (T1)	17	25	13→16
Chronic HCV (T2)	68	84	31→42
Resolved HCV (T2)	15	20	12→13
DAA-treated Chronic HCV (T2)	-	20	0→9
Healthy controls	14	20	13→20

ALT values, viremia levels and male/female ratios for each patient group have been included in the new **Supplementary Tables 3a** and **3b**.

With respect to symptoms, all patients were asymptomatic, as it is generally observed in acute and chronic HCV infections, and were identified only through ALT testing for unrelated reasons.

Although, as correctly stated by the Reviewer, the above “clinical factors” and other specific patient cohort features can potentially have “a strong impact on the data”, we were unable to detect any significant correlation between the above parameters and our functional profiling data.

2) It should also be made clear which patients are studied in which experiment, since it seems that almost every experiment was performed using cells from different patients. How similar where the different patients clinically? And should one not compare for each experiment whether differences in the patient groups (e.g. viral load, time post infection) where comparable to the patients/samples in other experiments?

Most of the experiments were conducted on the same patients, even though for some patients there were not enough cells to perform all the different assays. As specified in our answer to point #3 of Reviewer #1, metabolic and epigenetic assays must be performed on virus-specific, dextramer-positive cells and each assay, which typically requires at least 4 million PBMCs, must be performed separately due to the use of technically incompatible staining procedures (e.g. glucose uptake, GLUT1 expression, JC1, ROS and proteasome stainings).

Specifically, most of the patients used for transcriptomic analysis have also been employed for validation assays. In particular, metabolic assays on dextramer-positive CD8+ T cells have been performed in all T1/early acute patients (Fig. 2b, 2d and 2e). Regarding T2/late patients, metabolic and epigenetic assays on dextramer-positive CD8+ T cells have been done on most of them (Fig. 5c and 5e), with the exception of T2/late chronic patients because PBMC samples from some of them were totally used for transcriptome analysis, given the very low frequency of HCV-specific CD8+ T cells (not more than 500-1000 virus-specific CD8 T cells were retrieved from 40-50 millions of PBMCs).

In addition, following-up to this Reviewer's remark, in the revision work we have tried to expand as much as possible the number of patients' samples employed for each assay. To do so, we have used left-over frozen cell samples, whenever available, to expand assays that were only partially covered in the original analyses.

Finally, regarding the issue of the clinical homogeneity of the patients' groups, in the new **Supplementary Table 3a** we have depicted how patients were grouped based on highly stringent clinical parameters, suitable to guarantee a clear diagnostic discrimination of the different clinical categories (acute stage vs resolution stage vs chronic stage).

3) There is very limited detail about the analysis of the transcriptional data and its analysis. The tables do not fully allow to judge the strength of the observations and to compare results, which is especially an issue given the very small sample sizes in some of the groups (as low as 4). Throughout the manuscript, later experiments are often justified by the transcriptional data, but in some cases I could not find the mentioned results in the tables and it is not clear whether the selected pathways and molecules were actually selected because of them being highly significant top hits in the transcriptional analysis, or whether they were selected based on the existing literature and then justified by some observed difference in the transcriptional data. Examples are the GSEA results, which are impossible to interpret from the table alone. Certainly the adjusted FDR/q values are almost universally not very strong. How were the best/most promising pathways picked here? In addition, the authors mention upregulation of inhibitory receptors PD-1, CD160 etc supposedly shown in suppl table 2, but this table does not contain any individual gene expression data. Based on gene expression data in suppl table 1, these genes are not downregulated in chronic samples at T2 (unless the results in table 1 are logarithmic, which is not indicated).

We apologize for the not so clear presentation of transcriptome profiling data. Part of this lack of clarity we believe is due to the use of different gene name/gene product designations in the text and in **Supplementary Table 2**, for example, the reference name p38 in the text and the gene symbol MAPK14 in **Supplementary Table 2** for the same gene/transcript or gene product. In the revised manuscript, we have tried to solve this problem by using both designations for all the mentioned genes.

Regarding transcriptional analyses applied to our transcriptome data, two methods have been implemented. A topology-based analysis, which relies not only on the assignment of dysregulated genes to a given pathway but also on gene connections pointed out by pathway annotation, was applied to the T1/early and T2/late datasets. Pathway analysis was performed using a topology-based approach (with a multivariate statistics) with Reactome and the KEGG Pathway Database as annotation sources^{9, 10}. Then, we used gene-set enrichment analysis (GSEA) to further investigate with an unsupervised tool without any fold-change cutoff^{11, 12} the processes underlying T cell exhaustion.

Thus, specific pathways/biological functions were selected for functional validation assays based on their statistical significance, which, given the relatively small number of patients that were subjected to transcriptome profiling, was assessed by both multivariate topology-based pathway analysis (**Supplementary Table 1**) and GSEA (**Supplementary Table 2**). Genes related to intracellular signaling downstream to TCR activation, DNA damage response and mitochondrion (MT)/metabolic functions were identified as the most significantly dysregulated by both statistics (**Supplementary Table 1** and **Supplementary Table 2**).

Furthermore, more comprehensive (and hopefully easier to read) transcriptional data tables have been added, which include all relevant gene ontology (GO) terms and pathways ordered by p-value and FDR significance (see the T1 and T2 datasheets named as ‘FDR for Significance’ in the extensively revised **Supplementary Table 2**). In addition to individual gene expression data (see the datasheets named ‘T1 and T2 genes’ in **Supplementary Table 2**), also displayed in the new **Supplementary Table 2** are the mean expression fold-change values for the significantly modulated genes derived from GSEA leading-edge analysis, previously only shown in the GSEA heat-maps (**Supplementary Fig. 2a** and **Fig. 5b**). We hope that this new mode of presentation, which displays pathways and GO terms ordered by significance and by biological function (separate datasheets in **Supplementary Table 2**), will improve readability compared to the previous manuscript, where only statistically supported pathways and GO terms identified by GSEA (grouped by biological significance as in **Supplementary Fig. 2b**) were reported.

As to the adjusted FDR/q values, although we agree with the Reviewer that they “are almost universally not very strong”, we would like to point out that we used significance cut-off levels for FDR and p-value of <0.1 and <0.05, respectively, which are more stringent than or equal to those recommended by the BROAD Institute (FDR and p-values of <0.25 and <0.05, respectively)^{11, 12}.

Regarding specifically PD-1, CD160 and other inhibitory receptors, we referred to them in the ‘Results’ section entitled ‘Metabolic and signaling pathways are dysregulated in virus-specific CD8+ T cells from the acute phase of HCV infection’ as upregulated at the T1/early time-point in virus-specific CD8+ T cells from chronically-evolving compared to self-limited patients. Indeed, these and other co-inhibitory receptors (e.g. CTLA4) were listed among the 202 significantly dysregulated genes belonging to the ‘REACTOME_ADAPTIVE_IMMUNE_SYSTEM’ pathway (see the **T1 datasheet in Supplementary Table 2**). Notably, this pathway features a positive ‘Normalized Enrichment Scores’ (NES) value of 1.76 and only contains genes (ranked according to the hypergeometrical score, HGS) that are significantly upregulated in T1/early acute chronically-evolving patients. Finally, as requested by the Reviewer, individual expression values for all the significantly modulated genes displayed in the heat-maps of **Supplementary Fig. 2a** and **Fig. 5b** are now reported in the **new Supplementary Table 2** (see the ‘T1 genes’ data-sheet)

4) The experiments in figure 2b, while significantly different between groups, show results with a wide overlap between chronic and resolving cells. Did the authors test whether the results were correlated with other parameters than outcome, most importantly viral load? Also, it would be nice to see some original flow data and to see the results for total CD8 populations.

We agree with the Reviewer on the importance of identifying possible correlations between clinical parameters and functional results, but based on available clinical data (see also our response to point #1) only trends without a clear significance have been observed. For instance, as shown by the representative example reported below, an apparent but statistically non-significant inverse relationship was observed between HCV-RNA levels and glucose uptake in virus-specific CD8⁺ T cells. This point is, therefore, not explicitly mentioned in the revised manuscript.

With regard to **Fig. 2b**, the plots below (attached to this letter for Reviewer's assessment) illustrate some original flow data on glucose uptake (**A**) and Glut-1 expression (**B**) for both dextramer-positive (*black*) and total CD8⁺ T cells (*green*). Some representative MFI histograms and dot-plots are similarly shown in **panels C** and **D**, where patient groups are subdivided as in **Fig. 2b**. No significant MFI differences between individual patient groups could be observed in the case of total CD8⁺ T cells.

A

B

C

D

5) According to the methods section the seahorse experiments from figure 2 were performed using total CD8 T cells, not specific cell populations. Would this not lead to a major impact of the relative population size of the specific cells within the total CD8s, with the populations being much smaller in chronic evolution, especially compared to EBV/CMV? And should unstimulated cells really respond as shown here, with almost as much as a response than stimulated healthy memory cells? And why are no results shown for acute resolving HCV infection, this being the critical comparison to chronic HCV.

We agree with the Reviewer on the importance of the relative size of the virus-specific T cell pool within the total CD8+ T cell population. Control experiments assessing HCV-specific T cell activation upon peptide stimulation (only mentioned as ‘data not shown’ in the previous manuscript) are now displayed in **Supplementary Fig. 3b** and discussed in the text (page 9, line 23). As revealed by these experiments, IFN- γ production is strongly induced by FLU-, EBV- and CMV-specific peptides and a less intense but still highly significant activation compared to unstimulated samples is also induced by HCV-specific peptides. These data (some of which are shown below as representative examples) clearly demonstrate that overnight stimulation with FLU or HCV peptide mixtures prior to Seahorse analysis activates not only FLU- but also HCV-specific CD8+ T cells, thus allowing to link metabolic changes with virus-specific T cell activation.

As suggested by the Reviewer, we also added some additional Seahorse results (figures below and **Fig. 2f**) obtained in T1/early acute patients with chronic evolution of infection, that confirm the impaired mitochondrial functionality of HCV-specific cells, compared to FLU-stimulated or unstimulated CD8+ T cells.

In particular, data derived from ‘spare respiratory capacity’ (SRC) measurements, in which the respiratory electron chain is forced to operate at maximum rate, closely reflect the ability of a cell to respond to a sudden energetic demand and thus represent fairly reliable indicators of cell fitness and/or metabolic flexibility. Cumulative results obtained from the analysis of the SRC (Maximal vs Basal ‘Oxygen Consumption Rate’ ratio) are illustrated above. Importantly, by normalizing OCR values associated to the Maximal Respiration with respect to Basal Respiration OCR values measured in the same sample as in **Fig. 2f**, the result becomes independent from the relative numbers and activation levels of virus-specific cells present in the total CD8 population. Again, this kind of analysis clearly shows that stimulated HCV-specific CD8+ T cells display a reduced respiratory capacity and mitochondrial fitness compared to FLU-specific and total CD8+ T cells, while, as expected, the maximum respiratory capacity of FLU-specific and total CD8+ T cells does not differ significantly.

The above conclusion is also supported by the results of “Proton leak” and “Coupling efficiency” measurements, as defined in Jastroch et al (Essays Biochem. 2010, 47:53-67)¹³. As shown below, ‘proton leak’, which is voltage-dependent, is higher in stimulated compared to unstimulated cells, in agreement with the notion that mitochondrial membrane potential increases upon T cell activation. Also, ‘proton leak’ is maximal in FLU-specific cells, which feature the highest mitochondrial membrane potential increase, as revealed by JC-1 staining (see **Fig. 2d**).

The opposite situation is observed for “Coupling efficiency”, a measure of the functional link between mitochondrial ATP production and respiration, calculated as ATP production rate / basal respiration rate x 100 (see Brand and Nicholls, Biochem J. 2011, 435: 297–312)¹⁴. In this case, the lower “Coupling efficiency” value measured in FLU-stimulated cells is expected since, as shown in **Fig. 2f**, these cells display the highest ATP production and basal OCR levels.

We hope that these additional data may convince the Reviewer that multiple Seahorse measurements performed on total CD8⁺ T cells from T1/early chronically-evolving acute patients are really representative of diverse metabolic functions that distinguish two differently stimulated T cell subsets (Flu vs HCV) as well as unstimulated T cells.

Finally, as requested by the Reviewer, we performed Seahorse experiments on CD8⁺ T cells from T1/early acute self-limited patients. As shown in the revised **Supplementary Fig. 3** (panels **d**, **e** and **f**) and briefly mentioned in the text (page 10 line 12), these showed an opposite behavior compared to CD8⁺ T cells from T1/early chronically evolving acute patients. In particular, HCV NS3-stimulated CD8⁺ T cells from T1/early acute self-limited patients displayed higher ECAR and OCR values compared to CD8⁺ T cells from T1/early chronically-evolving patients, both at basal and maximal levels. These data are summarized in the cell-energy phenotype plots shown in **Supplementary Fig. 3d** for T1/early acute chronically evolving and acute self-limited patients.

6) In figure 2e, the results between chronic and resolving patients are connected with lines. How can these data from different patient groups be paired?

CD8+ T cells from chronic and resolving patients were tested in parallel, in the same experimental session to generate the data shown in **Fig. 2e**. However, as correctly pointed out by the Reviewer, this does not warrant their pairwise combination. This has been fixed in the new version of **Fig. 2**.

7) The PD-1 data in suppl figure 3b are rather surprising. Usually PD-1 expression during the acute phase is extremely high in all HCV-specific CD8 T cells irrespective of outcome. In contrast, flu-specific cells are usually PD-1 negative. Here the differences are extremely small and resolving patients show no PD-1 upregulation at all. Overall the PD-1 MFI values are so low that one wonders whether any cells were actually positive. These are extremely unusual results and impossible to interpret without seeing the staining for bulk CD8 T cells in dot plots and seeing any clinical context for the patients, such as time after infection and viral load.

We agree with the general PD-1 expression trend delineated by the reviewer. We would like to point out however that the results in **Supplementary Fig. 3** illustrate median values of PD-1 expression measured after anti-CD3+/anti-CD28 overnight cell stimulation, showing increased PD-1 expression. Therefore, what we observe also in the case of Flu-specific CD8+ T cells is the increased PD-1 expression expected after *in vitro* T cell activation. We also note that these assays were performed to assess the relation between glucose uptake and PD-1 expression under identical conditions of T cell stimulation and culture.

Below is an example of overnight stimulated FLU-specific CD8+ T cells, illustrating the flow cytometric measurement of PD-1 expression (with the appropriate negative controls) and documenting how PD-1 positivity can be reliably defined both in virus specific (**A**) and in total CD8+ T cells (**B**).

A**B**
The reliability of our PD-1 expression data was further supported by additional measurements (new **Supplementary Fig. 7b**) conducted on a new set of T2/late infected patients not included in the previous version of our manuscript.

Representative histograms of PD-1 MFI (median fluorescence intensity) values are attached below.

We agree with the Reviewer that the “PD-1-MFI values are overall quite low”. We would like to point out, however, that they are comparable to those previously reported in other papers dealing with the cytofluorimetric evaluation of PD-1 expression. For example, if we plot the data in **Supplementary Fig. 3c** (median MFI values of total dextramer-positive CD8+ T cells) in the form of MFI of the PD1+ dextramer+ CD8+ T cell subset, as in Wieland et al, (Nat Commun. 2017 May 3;8:15050)¹⁵ our MFI values become comparable to or even higher than those reported by these authors, as shown in the figure below.

Even though, as shown below, the plotted data are clearly significant, given the low frequency of this fairly rare T cell sub-population, we feel more confident in showing data derived from total dextramer-positive CD8+ T cells.

8) The results in figure 3 are not really convincing. First, differences are rather subtle, and in 6/8 of the assays flu results are not different from those for chronic HCV. I am also not convinced that all of the p values are correct (most notably the left panels in 3b and 3d). Finally, are the differences for example for p53 really biologically relevant? These values look similar to the PD-1 staining (very low MFI, <300 ex vivo) and total CD8 results should be shown in dot plots. Similar issues apply to the phosphoflow staining, which is also lacking the data from unstimulated cells as key control. Usually phosphoflow is reported showing both unstimulated and stimulated cells together, and differences tend to be substantial.

Regarding the assays in **Fig. 3**, we would like to point out that they have been performed on dextramer-positive, virus-specific CD8+ T cells. Due to the extremely low frequency of such cells, these assays required a minimum of 4 million PBMCs to allow the recovery of sufficient numbers of virus-specific T cell effectors, and each of them was performed on separate samples because of the different staining procedures required for each measurement. That is why relatively few data were reported in the previous version of the manuscript. As shown in the **revised Fig. 3** (T1/early time-point) and **Supplementary Fig. 6** (longitudinal comparison between T1/early and T2/late), despite these limitations, we managed to include additional patients at different stages of infection (patient number increased from 4 up to 11 for some groups; see revised **panels 3b and 3d**), in order to increase the robustness of the data (see also the legend of **Fig. 3**). To try to compensate for the *ex vivo* variability of signaling effectors expression levels, we also performed additional validation experiments by testing the expression of individual signaling proteins after 10-days of T cell stimulation in order to increase the frequency of dextramer+ cells. Furthermore, as also requested by Reviewer #1 (see our response to his/her point #3), we analyzed all patient groups longitudinally in order to compare the expression of each intracellular effector protein at the T1/early and T2/late time-points (see the **revised Supplementary Fig. 6**). T2/late chronic patients showed a decline in ATM-mediated signaling, as indicated by the decreased phosphorylated-ATM MFI values measured after 10 days of culture (**Supplementary Fig. 6a upper panel**). Conversely, and in line with the results of transcriptional analysis, total p53 was increased in chronic compared to resolved patients both at T1/early and T2/late time-points (see **Fig. 3b** and **Supplementary Fig. 6b upper plots**, respectively). Although a similar increase was not observed for phosphorylated p53 (a likely consequence of ATM kinase downregulation in T2/late patients), we cannot exclude p53 phosphorylation at different sites mediated by alternative signaling kinases (see, for example, Meek and Anderson 2009 Cold Spring Harb Perspect Biol. 2009 Dec;1(6):a000950)¹⁶. Indeed, p53 inhibitor treatment was found to be highly effective in restoring both glucose metabolism (**Fig. 6b**) as well as anti-viral functions (**Fig. 6a** and **6c**) in exhausted T cells from T2/late chronic patients, thus confirming a central role of p53 in T cell exhaustion maintenance.

Although we agree with the Reviewer's remark on the relatively low MFI values, we would like to point out that fluorescence intensity is strongly affected by flow cytometer (especially

photomultiplier) settings and that comparison of target proteins expression levels between patients groups, rather than absolute MFI values, was the main goal of our cytofluorimetric measurements.

To better document this point, we attach below representative examples of the different assays to show how they have been set up and how they can allow to detect positive expression in all patient categories; we also include a plot depicting the results of the above assays, as applied to total CD8+ T cells from a chronically evolving patient.

Regarding phospho-flow staining analyses, we would like to point out that these were preceded by set-up experiments (performed on all signaling proteins), aimed at determining optimized ('best response') conditions of anti-CD3/CD28 stimulation, as originally delineated by Finlay et al (J Exp Med 2012, 209:2441-53)¹⁷. Overnight anti-CD3/CD28 stimulation was thus found to yield the maximal increase in phosphoprotein MFI values compared to unstimulated controls (see the representative graphs attached below). However, due to cell amounts limitations (see also our response to point #3 above), this kind of comparison could not be performed for all phospho-assays. That's why we decided to focus on the comparison of the phosphoprotein MFI values measured for the different signaling proteins in CD8+ T cells from the different patient groups, rather than on internal ('stimulated' vs. 'unstimulated') comparisons for individual cell samples and signaling proteins. We also note that this kind of comparative (inter-CD8+ cell samples) mode of data presentation has been utilized before in related papers, in which the percentage of phospho-effector-positive cells was measured and compared in virus-specific CD8+ T cells at different stages of LCMV infection (see, for example, Bengsch et al. *Immunity* 2016, 45:358-73; Staron et al. *Immunity* 2014, 41:802-14; Chang et al. *Cell* 2013, 153:1239-51)^{18, 19, 20}. In addition to these considerations, the use of a data presentation format similar to that employed in the above papers may also facilitate the comparison of our phospho-assay results (the first of this kind to be reported in humans) with those obtained in the murine LCMV model of chronic infection and T cell exhaustion.

Phosphorylation assays

9) The data in figures 4 and 6 are very interesting, but difficult to evaluate without control experiments using T cells from resolving HCV and flu. Are all these inhibitors only affecting chronic cells?

Following-up to the Reviewer's comment, we performed all the requested controls and these are shown in the new **Supplementary Figs. 4 and 9** and briefly mentioned in the revised text (pages 12 and 15, lines 7 and 21, respectively). As apparent in **Supplementary Fig. 4a**, specific ATM, p53, AMPK and p38 inhibitors had only a modest effect on HCV NS3 peptide-stimulated PBMCs from six patients with T1/early acute self-limited infection as well as on FLU-/CMV-/EBV-peptide-stimulated PBMCs from six T1/early chronically-evolving patients that were utilized as fully functional CD8+ T cell controls (see **Supplementary Fig. 4b**). Moreover, the effect of inhibitors selectively targeting HMTs G9a (UNC0638 and BIX01294) and EZH2 (GSK126 and EPZ005687) or p53 (pifithrin-alfa), which are upregulated in the T2/late chronic stage, was significantly more potent on exhausted HCV-specific CD8+ T cells than on fully functional HCV-specific CD8+ T cells from T2/late resolved HCV patients (**Supplementary Fig. 9a**) and on functional CD8+ T cells of different virus-specificities (FLU, CMV, EBV, **Supplementary Fig. 9b**). Based on these results, we assume that the above inhibitors preferentially target overexpressed intracellular effectors and thus act more efficiently on dysfunctional T cells.

10) The results in figure 5a, where these pathways selected for being the most differentially regulated between T1 and T2, or what was the reason to select these? From the results as presented it is not clear which pathways were the most significant by transcriptional analysis.

We apologize for not having made this point sufficiently clear. The pathway/GO list in **Fig. 5a** was only meant to show the most significant pathways shared by T1/early and T2/late CD8+ T cells, that, as highlighted by their opposite-sign NES (Normalized Enrichment Scores) values, are divergently modulated at these time-points.

More comprehensive gene expression data tables, including all significant pathways ordered by p-value and FDR significance, have been added to the revised manuscript (see the T1 and T2 data-sheets named 'FDR for Significance' in **Supplementary Table 2**).

Reviewer #3 (Remarks to the Author):

1) Some of the findings or related findings have been made already showing p53 loss enhances anti-tumor activity of T cells associated with increased glycolysis and cytokine production and antitumor activity in a mouse study (e.g. Banjeree from Cancer Research Paper).

We thank the Reviewer for this insightful comment, which corroborates the rationale of our study and extends its possible implications to other T cell exhaustion-related diseases including cancer. The suggested reference²¹ has been incorporated into the revised manuscript.

2) The authors mention H3K27me3 in the text but data in Fig 5 only shows H3K9me2. Did they look at H3K27me3 levels, or other H3K9 methylations?

Unfortunately, although we mention H3K27me3, we could not address this particular histone-tail modification due to the lack of suitable antibodies validated for use in flow cytometry.

3) Do they have evidence the H3K9 methylation levels were reduced in Fig 6.

We thank the Reviewer for this comment. To properly address it, we have added in **Fig. 6d** new data showing a reduction of H3K9 methylation levels after HMT inhibitor treatment, both *ex vivo* and after 10 days of culture.

Reviewer #4 (Remarks to the Author):

The major comment by this reviewer is the timing of the samples.

1. The T1 point, so-called the early samples, were obtained during the ALT peak of acute infection. As reported in the study, the T cells from those with acute HCV chronic evolution already demonstrated early markers of immune dysregulation, which then progressed to a phenotype of full immune exhaustion or nonresponse during the chronic phase (T2). It is the opinion of the reviewer that samples obtained prior to this point (during the acute asymptomatic infection stage) are essential to unravel the mechanism that lead to the described immune and metabolic dysfunctions.

The point raised by the Reviewer is formally correct, but identification of prospective chronic patients in an early incubation phase (i.e., prior to ALT elevation) would require an extensive and extremely prolonged monitoring (likely lasting for years) of subjects belonging to specific risk groups. In fact, HCV infection is generally asymptomatic, especially before clinical presentation. In the past we have tried to follow cohorts of drug addicted subjects longitudinally and to find early infections by collaborating with physicians working in blood banks but efforts to identify patients immediately after infection and before ALT peak have been completely unsuccessful.

2. Also it is not clear whether the same patients were followed for the T1 early vs T2 late phase. Longitudinal study of same patients would have been much more compelling and eliminated much of the individual variations.

In the original study design, the same patients had to be followed from the T1/early acute phase to the late stages of resolution or chronic evolution of disease. While we have been able to identify a cohort of patients showing a rapid and spontaneous resolution of infection without therapy, patients with longer duration of the acute stage of infection underwent therapy based on present clinical guidelines. This prevented the possibility of establishing whether in the absence of therapy the outcome would have been control of infection and recovery or virus persistence and chronic disease. For this reason, self-limited infection samples derive from the same patients at both the T1/early and T2/late time-points. Instead, T1/early samples from acute infection with chronic evolution derive from patients' PBMCs that were stored and classified at a time when early therapy was not yet mandatory, while the T2/late stage was studied in PBMCs derived from late-diagnosed chronic patients, most of whom with a well characterized history of disease that allowed to reliably define the time of acute infection.

An additional table (**Supplementary Table 3**) reporting the main features of the 187 patients investigated in this work has been added to the revised manuscript (see also our response to point #1 by Reviewer #2).

3. Otherwise, this paper merely describes the natural evolution of immune exhaustion leading to ineffective antiviral immune response and chronic viral infection.

We believe that our study provides novel and important mechanistic insights into CD8⁺ T cell exhaustion, that go well beyond a simple transcriptional analysis. In fact, in-depth functional data-mining allowed us to identify and validate multiple signaling, metabolic and epigenetic defects associated with the dysregulated transcriptional profile of exhausted CD8⁺ T cells, thus highlighting novel candidate targets suitable for new immunomodulatory strategies aimed at reinvigorating exhausted CD8⁺ T cells. This study required an enormous effort in patient selection and enrollment in order to solve the problem of the low frequency of circulating HCV-specific CD8⁺ T cells, which are often insufficient to allow the analysis of multiple functional parameters on the same blood sample. For this very reason a large amount of work has been devoted to the optimization of the sensitivity and reproducibility of the different molecular and functional assays, so to enable the study of most patients *ex vivo* –i.e., without long-term CD8⁺ T cell *in vitro* stimulation (as in most studies on HCV adaptive immunity published so far) and the inherent risk of expansion/selection of cell populations that are not prevalent *in vivo*.

Finally, the revised manuscript also contains important new data on the effect of DAA treatment on metabolic, signaling and epigenetic parameters, which further enhance the overall novelty of our study.

Although we agree with the Reviewer that characterization of very early post-infection time-points and sequential monitoring of the same chronically evolving acute patients would have further improved the quality of our manuscript, we are nevertheless convinced that our study, because of the novel pathogenetic insights and new potential therapeutic avenues it provides, represents an important step forward in our understanding of T cell exhaustion and can thus be of interest to a large audience of scientists working not only on chronic hepatitis C but more generally on T cell exhaustion associated diseases, including cancer.

References

1. Söderholm J et al., Relation between viral fitness and immune escape within the hepatitis C virus protease. *Gut*. 2006 Feb;55(2):266-74.
2. Urbani S et al., The impairment of CD8 responses limits the selection of escape mutations in acute hepatitis C virus infection. *J Immunol*. 2005 Dec 1;175(11):7519-29.
3. Schurich et al., Distinct Metabolic Requirements of Exhausted and Functional Virus-Specific CD8 T Cells in the Same Host. *Cell Rep*. 2016 Aug 2;16(5):1243-1252;
4. Fisicaro et al., Targeting mitochondrial dysfunction can restore antiviral activity of exhausted HBV-specific CD8 T cells in chronic hepatitis B. *Nat Med*. 2017 Mar;23(3):327-336.

5. Cossarizza A et al., Guidelines for the use of flow cytometry and cell sorting in immunological studies. *Eur J Immunol.* 2017 Oct;47(10):1584-1797.
6. Martin B et al., Restoration of HCV-specific CD8+ T cell function by interferon-free therapy. *J Hepatol.* 2014 Sep;61(3):538-43.
7. Callendret B et al., Persistent hepatitis C viral replication despite priming of functional CD8+ T cells by combined therapy with a vaccine and a direct-acting antiviral. *Hepatology.* 2016 May;63(5):1442-54.
8. Van der Ree MH et al., Immune responses in DAA treated chronic hepatitis C patients with and without prior RG-101 dosing. *Antiviral Res.* 2017 Oct;146:139-145.
9. Martini, P., Sales, G., Massa, M. S., Chiogna, M. & Romualdi, C. Along signal paths: An empirical gene set approach exploiting pathway topology. *Nucleic Acids Res.* 41, (2013).
10. Sales, G., Calura, E., Martini, P. & Romualdi, C. Graphite Web: Web tool for gene set analysis exploiting pathway topology. *Nucleic Acids Res.* 41, (2013).
11. Subramanian A et al., Gene set enrichment analysis: a knowledge-based approach for interpreting genome-wide expression profiles. *Proc Natl Acad Sci U S A.* 2005 Oct 25;102(43):15545-50.
12. <https://software.broadinstitute.org/gsea/doc/GSEAUserGuideFrame.html>
13. Jastroch M et al., Mitochondrial proton and electron leaks. *Essays Biochem.* 2010;47:53-67.
14. Brand MD, Nicholls DG. Assessing mitochondrial dysfunction in cells. *Biochem J.* 2011 Apr 15;435(2):297-312.
15. Wieland D et al., TCF1+ hepatitis C virus-specific CD8+ T cells are maintained after cessation of chronic antigen stimulation. *Nat Commun.* 2017 May 3;8:15050.
16. Meek DW, Anderson CW. Posttranslational modification of p53: cooperative integrators of function. *Cold Spring Harb Perspect Biol.* 2009 Dec;1(6):a000950.
17. Finlay DK et al., PDK1 regulation of mTOR and hypoxia-inducible factor 1 integrate metabolism and migration of CD8+ T cells. *J Exp Med.* 2012 Dec 17;209(13):2441-53.
18. Bengsch B et al., Bioenergetic Insufficiencies Due to Metabolic Alterations Regulated by the Inhibitory Receptor PD-1 Are an Early Driver of CD8(+) T Cell Exhaustion. *Immunity.* 2016 Aug 16;45(2):358-73.
19. Staron MM et al., The transcription factor FoxO1 sustains expression of the inhibitory receptor PD-1 and survival of antiviral CD8(+) T cells during chronic infection. *Immunity.* 2014 Nov 20;41(5):802-14.
20. Chang CH et al., Posttranscriptional control of T cell effector function by aerobic glycolysis. *Cell.* 2013 Jun 6;153(6):1239-51.
21. Banerjee A et al., Lack of p53 Augments Antitumor Functions in Cytolytic T Cells. *Cancer Res.* 2016 Sep 15;76(18):5229-5240.

Reviewers' comments:

>Reviewer #1 (Remarks to the Author):

Comments#

The authors have carefully addressed most of my concerns. Still, there are some concerns that should be addressed prior publication:

- Figure 2 – it is unclear how many patients were analyzed in the Seahorse experiment. It seems as if the statistical tests are performed on technical replicates rather than biological replicates. Since the purified cells were CD8+ T cells then stimulated with individual peptides for control vs. HCV specific responses, were the baseline virus-specific T cell responses similar? Otherwise, the differences in the virus-specific frequencies could account for the differences observed independent of qualitative differences between HCV-specific and control T cell responses. Seahorse data should also be normalized to baseline ECAR and OCR.

- Figure 3 – p53 plays a major role in deciding cell fate. High apoptosis of HCV-specific T cells was described by Radziejewicz et al in acute infection (J Virology 2008). How strict were apoptotic and early apoptotic T cells excluded from the analysis performed? This could bias the analysis towards the reported phenotype.

- Figure 3b – differences in pATM MFI between acute chr evolving HCV and FLU seem minimal, yet high significance is reported. What is the statistical test used here? Representative overlay histograms should be added to allow the reader to judge the magnitude of change.

- Figure 4ab – A complex assay was performed in which PBMCs were stimulated with HCV peptides in presence or absence of antioxidants or signaling inhibitors. Are these effects HCV-specific or do they also occur after stimulation with control peptides? Representative plots should be added to the main figure.

- In the revised manuscript the authors show novel data regarding the effect of DAA therapy showing partial correction of some metabolic marks. In the discussion, they discuss these findings and state that “there is controversial results on the extent of virus-specific T cell function...after DAA therapy.” It is probably more correct to cite the paper by Wieland et al, published in Nature communications in 2017, that also supports partial recovery of virus-specific CD8+ T cell responses after DAA mediated antigen removal. There is consensus in the field that at least in humans (the authors cite one chimpanzee paper) that DAA mediated antigen removal leads indeed to partial restoration.

- Along the same line, considering the current publications, the phenotypical characteristics of HCV-specific CD8+ T cells after DAA-mediated therapy were not addressed satisfactorily and the influence of the heterogeneity within the HCV-specific CD8+ T cell population was not addressed. This needs at least to be discussed.

- Minor points:

- In legend of figure 2c there is a mistake: Cells were stimulated overnight with either HCVNS3 peptides (red) or control (FLU-, CMV- and EBV-specific) peptides (blue), or were not unstimulated (green). This should clearly mean “not stimulated”.

- In figure 2d, the ex vivo data points of the acute HCV chronic evolving patients seems to be completely different from the same data set of the first draft of the publication. Please check for mistakes regarding graph generation.

>Reviewer #2 (Remarks to the Author):

I greatly appreciate the massive efforts by the authors to clarify previously raised issues and especially to add more data to this already extremely complex manuscript. Several of the issues have been successfully addressed, but substantial questions remain about some of the data, their presentation and the analysis and interpretation of the data. Some new questions have also come up based on additional data that were provided in the revision.

Below are my most significant concerns:

1) Some of the statistical methods remain unexplained, most notably in figures 2 and 3 the Wilcoxon paired sample tes. It is unclear to me how cross sectional data from persisting versus resolving infection can be analyzed with a paired sample test. This inappropriate use use of a paired difference test explains why many of the results appear rather similar from looking at the graphs, yet supposedly have highly significant differences. This issue was raised previously and was not addressed.

2) The sequence analysis has only partially resolved the question about potential impact of viral escape. No sequencing data was provided for two of the 4 epitopes used in the transcriptomics analysis, though these two are known to escape more often than the other two that were studied. It also seems that none of the many other patients that were studied were sequenced.

3) The analysis of the transcriptional data is still hard to follow and interpret. Notably I cannot see how the results shown in figure 1d would point specifically towards glycolysis and glucose transport

as stated in the manuscript. These pathways are also not dominating the subsequent gene enrichment analysis. Overall I still fail to see how the transcriptomics data actually led towards the subsequent experiments. In addition, the newly provided gene expression lists raise some additional questions, the most obvious why PD-1, TIGIT and CTLA-4, while upregulated in chronic versus resolving patients at T1, show an inverse pattern at T2 (being higher expressed in resolved patients). This is rather different from what we know about the expression of these inhibitory receptors that are known to be most differentially regulated in exhausted (late chronic) cells versus memory (resolved infection) T cells. To me this raises serious questions about the validity of the gene expression data.

4) Regarding the Seahorse analyses the authors have provided data about the size of the antigen-specific populations for both HCV- and CEF-specific cells. While it is correct that HCV-specific responses are indeed also detectable, they are also with few exceptions less than 0.5% of total CD8s, which is about one log lower than the CEF populations. In my opinion this can fully explain the absence of reactivity for HCV responses in this assay.

5) I remain unconvinced by most of the data in figure 3. If the appropriate unpaired test is used, will the differences remain significant? In addition, while the authors are correct that phosphoflow experiments have been directly compared between different outcomes in LCMV using inbred mice, this seems more questionable to do in a human cohort, given the high genetic variability. For this reason I still think one needs the reference of unstimulated samples for each patient in these assays.

6) The new data in figure S4 are not useful to compare with the results in figure 4 as they are shown in a completely different format. Figure 4 shows fold changes per patient, figure S4 aggregate results for all samples.

7) The data in figure S5b for PD-1 and CTLA-4 directly contradict the microarray results in table 2. In S5b expression is higher in chronic patients and the opposite is shown in the microarray data, as already mentioned under 3)

8) A major new concern comes from the new supplemental data in figures S4/8/9. Here the ICS data show higher percentages for untreated cells of HCV-specific CD8 T cells in chronic compared to resolver patients at T2 directly ex vivo and even more stunningly post in vitro expansion. This is puzzling, as all published literature finds much stronger T cell responses directly ex vivo and also better expansion of T cells in resolved versus chronic infection. In addition, I am somewhat surprised by the fact that many somewhat modest looking differences in chronic patients are all highly significant, but none of the differences the graphs seem to indicate for resolvers. Why were that data not shown in the same way as in figure 4e/f .

9) It would be critical to see more raw data and more detailed results for the assays shown in figures 6 and 7. In the chronic stage of HCV infection the direct ex vivo ICS assay is typically negative for interferon-gamma and even more so for IL-2 in practically all patients. What is shown in figure 6c is feasible in acute patients, but is decidedly not a representative result for this assay in chronic (T2) patients. How many PBMC were studied and how big were the clouds of functional cells?

>Reviewer #3 (Remarks to the Author):

The authors addressed my previous concerns and appear to have addressed most or all the concerns of the other reviewers. Thus it is acceptable for publication.

>Reviewer #4 (Remarks to the Author)

None

>Reviewer #5 (Remarks to the Author):

1. General comments.

This is not an easy paper to review from a data analytic viewpoint. There are a huge number of analyses presented, and, in general I do not think these analyses are described very clearly. The underlying essence gets completely lost in a mass of experimental technical detail.

In general, the sample sizes are small and given the likely experimental variation in the variables being measured it seems unlikely that the findings are statistically robust.

Many P-values are reported to 4 significant figures. This is not appropriate precision.

2. The description of the statistical methods is very limited. Reveiwer #2 raised this issue (point #3) but clearly it has not been adequately addressed.

For example in first paragraph of p8 a topology-based analysis is mentioned, but there is almost no description in the statistical methods of what that actually means. I cannot understand how this has generated a q-value.

Similarly, the description of the gene-set enrichment analysis is inadequate.

3. Reviewer #1, point #5.

The relevant section is p7, para 2. The first sentence states that “Principal component analysis of ANOVA-filtered expression data ...”. This analysis is not described in the statistical methods and it is unclear to me what filtering has been applied. There are four patient groups with 5, 8, 4 and 7 patients. Parametric statistical methods are unlikely to be appropriate with such small numbers.

4. Figure 2/3. When comparing three groups it is inappropriate to compare group 1 with group 2 and then to compare group 1 with group 3. A general heterogeneity test for between group differences should be applied.

Of all the P-values reported in these figures $P=0.0039$ on seven occasions. This seems unlikely - though possible - with a rank based test (i.e. they would need to have observed the same ranking of different outcomes between the two groups on seven occasions).

5. Summary. I would not quibble very much with the individual statistical tests that have been applied, but the description of each of these and the inferences drawn from the findings of each needs to be clearer.

Paul Pharoah

Reviewers' comments:

>Reviewer #1 (Remarks to the Author):

Comments#

The authors have carefully addressed most of my concerns. Still, there are some concerns that should be addressed prior publication:

1.- Figure 2 – it is unclear how many patients were analyzed in the Seahorse experiment. It seems as if the statistical tests are performed on technical replicates rather than biological replicates. Since the purified cells were CD8+ T cells then stimulated with individual peptides for control vs. HCV specific responses, were the baseline virus-specific T cell responses similar? Otherwise, the differences in the virus-specific frequencies could account for the differences observed independent of qualitative differences between HCV-specific and control T cell responses. Seahorse data should also be normalized to baseline ECAR and OCR.

As mentioned in the legend to the revised Figure 2 and 3, we performed Seahorse experiments on six different patients with a chronically-evolving acute HCV infection (i.e. 6 biological replicates, were used for statistical analysis – each of them representing the average of duplicates performed in individual patients). Enriched purified CD8+ T cells from each patient were stimulated overnight with a total of 130, overlapping 15-mer HCV peptides covering the entire NS3 sequence, and with a pool of 50 immuno-dominant peptides from CMV, EBV and FLU epitopes of various HLA restrictions, as controls. Following-up to the indications of Reviewer #5 (contacted by the Editor as expert in statistics) we reconsidered and improved statistical analysis, by employing a heterogeneity test for multiple comparisons. Specifically, the new statistical analysis was performed by the non-parametric Friedman test to detect differences among various stimuli, corrected for multiple comparisons with the Conover comparison test. The data are displayed as box-and-whisker plots to highlight more clearly the differences between the distinct stimuli applied for Seahorse analyses. Please note that the spare respiratory capacity in Fig.3e is now expressed as percent instead of delta (previous Supplementary Fig.3f), to make identical the graphical representation of panels 3d and 3e.

The other point, regarding baseline normalization of Seahorse results, was addressed in the 'response letter' (Reviewer #2) attached to our previous resubmission, but we are pleased to clarify it again following the same reasoning as before but with the addition of some new results. Specifically, in the response to point 5 of Reviewer #2, we explained that the "Spare Respiratory Capacity" and the "Coupling Efficiency" are by definition normalized to basal respiration values. Indeed, the 'spare respiratory capacity' represents the ratio (expressed as %) between maximal and basal respiration, while the 'coupling efficiency' is the ratio between 'ATP production rate' and 'basal respiration'. This makes the Seahorse results independent from the frequency and activation level of the virus-specific cells present in the total CD8+ T cell population. 'Spare respiratory capacity' data show that stimulated HCV-specific CD8+ T cells display a reduced respiratory capacity and mitochondrial fitness compared to both total unstimulated and FLU-peptide stimulated CD8+ T cells. As shown in the new Figure 3d, FLU-stimulated CD8+ T cells, which display the highest ATP production and basal OCR values, also display the lowest "Coupling efficiency". Instead, proton leak (measured after oligomycin addition), which is voltage-dependent, is higher in stimulated compared to unstimulated cells. In agreement with the notion that mitochondrial membrane potential increases upon T cell activation, this is maximal in FLU-specific cells, which feature the highest mitochondrial membrane potential increase, as revealed by JC-1 staining (please see the new **Figure 3a**). The only parameter which is not shown as normalized to basal values in the new figure 3d is proton leak which is attached below for Reviewer's perusal (**Reviewer #1, Figure 1**).

Reviewer #1, Figure 1: Proton leak measurements normalized to baseline values. Experiments were performed with a Seahorse extracellular flux analyzer following oligomycin treatment. Metabolic flux profiling of purified CD8+ T cells from 6 T1/early chronically-evolving acute patients, was performed after overnight stimulation with either HCV-NS3 (red) or control (FLU-, CMV- and EBV-specific) peptides (blue), or no stimulation (green). Proton leak represents residual basal respiration, not coupled to ATP production, measured after oligomycin injection and is calculated by subtracting non-mitochondrial respiration from minimum rate respiration measured after oligomycin addition. Results were obtained with the Agilent Seahorse XF Report Generator of the Wave software, according to the manufacturer's instructions.

Altogether, these measurements indicate that the Seahorse results really reflect the diverse metabolic functions that distinguish two differently stimulated T cell subsets (HCV vs. Flu) as well as stimulated vs unstimulated total CD8+ T cells.

This conclusion is also supported by the Seahorse results we obtained with acute self-limited patients, that were included in our previous revision as an additional control patient cohort (in response to a specific request of this same Reviewer). Indeed, HCV NS3-stimulated CD8+ T cells from early acute self-limited patients (T1) displayed an opposite behavior, in terms of ECAR and OCR values measured at both basal and maximal respiration levels, compared to CD8+ T cells from early chronically evolving acute patients. This opposite behavior was reproducibly observed in all patients belonging to the two cohorts. Remarkably, the same discordant metabolic profiles were also observed when Seahorse analysis was performed on patients with different disease outcomes (self-limited vs chronically evolving infection) but comparable numbers of IFN-g producing CD8+ T cells following HCV peptide stimulation (**Reviewer #1, Figure 2;** patient #1 compared to patient #7), thus confirming once again that the different metabolic behavior really reflects a qualitative difference between self-limited and chronically evolving patients. Moreover, as apparent in the plots below (**Reviewer #1, Figure 2;** patient #1), also when comparable numbers of control and HCV peptide-specific CD8+ cells were used for Seahorse analysis, a difference in mitochondrial and glycolytic activity between HCV and control peptide stimulated cells was detected. In our view, this is an additional and quite convincing indication of a real qualitative metabolic difference, totally independent from the relative size of the virus-specific CD8+ T cell pool within the overall CD8+ T cell population.

ACUTE CHRONICALLY-EVOLVING

ACUTE SELF-LIMITED

Reviewer #1, Figure 2: IFN- γ production (detected by ICS in flow cytometry, left plots) and metabolic flux profiling (showing both ECAR and OCR) in CD8+ T cells either unstimulated (green) or stimulated overnight with HCV-specific peptides (spanning the complete NS3 sequence; red) or control peptides (FLU/CMV/EBV specificities; blue) in a chronically-evolving patient (#1) and in a self-limited acute patient (#7).

Finally, additional control experiments showed that Seahorse analysis has a sensitivity sufficient to distinguish between different metabolic conditions even at frequencies of virus-specific CD8+ cells below 0.5% of the total number of CD8+ T cells, i.e., within a range comparable to most of the HCV-specific CD8+ T cell frequencies we measured in our patient populations (please see our response to point #4 of Reviewer #2 for further details on this issue).

2.- Figure 3 – p53 plays a major role in deciding cell fate. High apoptosis of HCV-specific T cells was described by Radziejewicz et al in acute infection (*J Virology* 2008). How strict were apoptotic and early apoptotic T cells excluded from the analysis performed? This could bias the analysis towards the reported phenotype.

Although we regret that this comment was not made in the previous evaluation of our manuscript, we thank the Reviewer for having raised this point. We agree with the Reviewer that the major and more general role of p53 is to promote apoptosis and/or to permanently inhibit cell proliferation under various cell stress conditions. Nevertheless, it is becoming increasingly clear that p53 can also contribute to cell survival and to the maintenance of metabolic homeostasis (so called ‘metabolic adaptation or remodelling’). For example, a key and widely recognized function of p53 is to regulate energy metabolism by lowering glycolysis while augmenting mitochondrial respiration (see Kruijswijk F et al. P53 in survival, death and metabolic health: A lifeguard with a license to

kill. 2015 Nature Reviews Molecular Cell Biology 16, 393–405).

We mainly relied on 7AAD and LIVE/DEAD markers to exclude contamination by dead cells in our experiments, without the use of specific biomarkers to detect early apoptosis committed T cells. We note, however, that, at variance with the experimental strategy of Radziejewicz et al., our phenotypic and functional assays were all performed on dextramer-positive cells after overnight anti-CD3 and anti-CD28 stimulation. This experimental condition is expected to drive apoptosis-committed T cells toward apoptotic death and subsequent exclusion from analysis by 7AAD and LIVE/DEAD staining, which was always performed before all flow-cytometry assays.

As a second point, we would like to remark that our phenotypic analyses, which included different time-points of chronically evolving HCV infection, were characterized by persistently high p53 levels compared to resolving patients, long after the early acute stage of infection when, according to Radziejewicz et al, the apoptotic process is expected to be substantially attenuated.

Third, and most important, to directly evaluate whether the higher p53 levels measured in chronically evolving acute patients can actually result in an increased apoptotic death, we determined the expression levels of the apoptosis-associated marker Caspase 9 (as in Radziejewicz et al) and of the anti-apoptotic regulator BCL2 (known to be associated with T cell memory generation) in 10 additional patients. As revealed by these new experiments, reported in the attached **Reviewer #1, Figure 3**, approximately 50% of dextramer positive CD8+ T cells in chronically evolving acute patients are p53^{high}. However, only a very limited proportion of these cells (<2%) are Caspase 9 positive, while approximately half of them express BCL2 at high levels. Therefore, there is no evidence that the p53-positive HCV-specific CD8+ T cell population we analyzed was actually undergoing apoptosis. Instead, the observed divergence between p53 and Caspase 9 levels strongly suggests that the p53 increase is likely related to less conventional (e.g., metabolic adaptation) p53-mediated functions, rather than to apoptosis promotion.

C

Reviewer #1, Figure 3: A. Healthy control PBMC treated with staurosporine (1 μ M) for 12 hours to induce apoptotic cell death via caspase 9 induction. The upper dot plot shows caspase 9 positive cells (stained with a fluorescein active caspase-9 probe) in both dextramer-stained FLU (matrix)-specific (*blue*) and total (*grey*) CD8+ T cells); the lower dot plot shows apoptosis levels in untreated samples. B. Representative dot plots showing HCV-specific CD8+ T cells stained with a FITC active caspase-9 probe to detect apoptotic cells ex-vivo. C. Caspase 9, BCL2 and p53 expression levels in dextramer+ CD8+ T cells from self-limited and chronically evolving acute HCV patients; median values are indicated by horizontal black lines.

As correctly pointed out by this reviewer's request, the apoptosis results, shown above, were mentioned and described in the discussion paragraph with the reference of Radziejewicz et al.

3.-Figure 3b – differences in pATM MFI between acute chr evolving HCV and FLU seem minimal, yet high significance is reported. What is the statistical test used here? Representative overlay histograms should be added to allow the reader to judge the magnitude of change.

Following-up to the indications of Reviewer #5 (contacted by the Editor as expert in statistics) we reconsidered and improved statistical analysis. Regarding Figure 3b (now Figure 4), as specified in the revised figure legend, we employed a heterogeneity test for multiple comparisons in order to compare the different patient groups. Specifically, the new statistical analysis was performed by using the Kruskal Wallis one-way analysis of variance test corrected for multiple comparisons with the Dunn's Multiple Comparison Test. In addition, new patients have been included to address the concerns of Reviewer #2 as to the absence of unstimulated controls in phosphoflow experiments. We believe that this increase in the number of tested patients along with the use of a more appropriate statistical tests has significantly improved our analysis.

Due to all these changes, Figure 4 (previously Figure 3) has been completely reorganized to include also unstimulated controls and the ratio between stimulated and unstimulated cells. Moreover, as requested by this Reviewer, representative overlay histograms have been displayed next to each dot-graph.

4.- Figure 4ab – A complex assay was performed in which PBMCs were stimulated with HCV peptides in presence or absence of antioxidants or signaling inhibitors. Are these effects HCV-specific or do they also occur after stimulation with control peptides? Representative plots should be added to the main figure.

As suggested by the Reviewer, we performed additional experiments with control peptides to assess the HCV specificity of the modulatory effects elicited by antioxidants.

A weak and statistically non-significant effect of resveratrol was detected upon stimulation of CD8+ T cells with control FLU, CMV and EBV peptides (please see the modified Supplementary Fig. 6a). In contrast, as shown in the new Figure 5a (previously Figure 4), a more pronounced and statistically significant reduction in mitochondrial ROS content and ATM phosphorylation was observed upon stimulation with HCV peptides.

Moreover, additional control experiments have been done to investigate glucose uptake, GLUT1 and PD-1 levels following CD8+ T cell stimulation with FLU, CMV and EBV specific peptides in the presence or absence of specific signaling-inhibitory compounds (please see the modified panels b, c and d in Supplementary Fig. 6).

Due to space limitation in the main text figures, these additional data are now illustrated as a supplementary figure (new Supplementary Fig. 6) with the same graphical representation used for the main text Figure 5 (previous Figure 4). Please note that panels a, b, c and d in Supplementary Figure 6 are new, while panel e corresponds to panel b of the previous Supplementary Figure 4. All data are now expressed as dots, as requested by Reviewer #2, including the data represented in panel e, that were displayed as Whisker plots in the previous Supplementary Figure 4b.

Moreover, as requested by this Reviewer, representative overlay histograms and dot plots have been displayed in the new figure 5a and 5b (previous Fig. 4a,b).

5.- In the revised manuscript the authors show novel data regarding the effect of DAA therapy showing partial correction of some metabolic marks. In the discussion, they discuss these findings and state that “there is controversial results on the extent of virus-specific T cell function...after DAA therapy.” It is probably more correct to cite the paper by Wieland et al, published in Nature communications in 2017, that also supports partial recovery of virus-specific CD8+ T cell responses after DAA mediated antigen removal. There is consensus in the field that at least in humans (the authors cite one chimpanzee paper) that DAA mediated antigen removal leads indeed to partial restoration.

We thank the Reviewer for this comment. The sentence in the Discussion dealing with the recovery of CD8+ T cell function after DAA treatment has been modified according to his/her suggestions. However, a note of caution as to the actual effect of DAAs on T cell function has been maintained, mainly because of the lack of any direct experimental evidence in humans indicating that decline of viral antigen per se can actually lead to functional T cell restoration, which, instead, may well be indirectly related to other beneficial effects of DAA therapy, including a decline of liver inflammation and/or modulation of innate immunity, rather than just to HCV antigen decline.

6.- Along the same line, considering the current publications, the phenotypical characteristics of HCV-specific CD8+ T cells after DAA-mediated therapy were not addressed satisfactorily and the influence of the heterogeneity within the HCV-specific CD8+ T cell population was not addressed. This needs at least to be discussed.

We thank the Reviewer for this additional point. Although very difficult, if not impossible, to address experimentally due to the extremely limited availability of HCV-specific CD8+ T cells, the issue of the heterogeneity of these cells and its possible influence on the final outcome of DAA treatment has been mentioned in the revised manuscript.

- Minor points:

- In legend of figure 2c there is a mistake: Cells were stimulated overnight with either HCVNS3 peptides (red) or control (FLU-, CMV- and EBV-specific) peptides (blue), or were not unstimulated (green). This should clearly mean “not stimulated”.

We apologize for this typo, which has been amended in the revised text.

- In figure 2d, the ex vivo data points of the acute HCV chronic evolving patients seems to be completely different from the same data set of the first draft of the publication. Please check for mistakes regarding graph generation.

We thank the Reviewer for pointing out this formatting mistake, which was due to the high number of changes that were introduced during the extensive first revision of our manuscript. This and other formatting inconsistencies have been fixed throughout the revised manuscript.

Reviewers' comments:

>Reviewer #2 (Remarks to the Author):

I greatly appreciate the massive efforts by the authors to clarify previously raised issues and especially to add more data to this already extremely complex manuscript. Several of the issues have been successfully addressed, but substantial questions remain about some of the data, their presentation and the analysis and interpretation of the data. Some new questions have also come up based on additional data that were provided in the revision.

Below are my most significant concerns:

1) Some of the statistical methods remain unexplained, most notably in figures 2 and 3 the Wilcoxon paired sample test. It is unclear to me how cross sectional data from persisting versus resolving infection can be analyzed with a paired sample test. This inappropriate use of a paired difference test explains why many of the results appear rather similar from looking at the graphs, yet supposedly have highly significant differences. This issue was raised previously and was not addressed.

We fully acknowledge the Reviewer's concerns regarding the statistical methods previously used to analyze the data presented in Figures 2 and 3. Following-up to the indications of Reviewer #5, a statistician engaged by the Editor to evaluate the appropriateness of the statistical methods employed in our work, we have now double-checked and hopefully improved statistical analyses. Regarding the specific point raised by the Reviewer, a heterogeneity test for multiple group comparisons (Kruskal Wallis one-way analysis of variance with correction for multiple comparisons by the Dunn's Multiple Comparison test) is now used to analyze the data. This has been applied to the results reported in the new Figures 2 and 3, but also in Figures 4, 6 and 8 and in Supplementary Figures 4d, 9 and 10. Statistical significance of most data was confirmed by this analysis, and only for *in vitro* expanded CD8+ cells, 2 out of 8 sets of measurements lost significance, although a trend consistent with the original evaluation was maintained. Please note that to address the Reviewers' requests new patient data for protein phosphorylation, glucose uptake and GLUT1 expression levels have been added to these revised figures.

In particular, the previous Figures 2 and 3 (now Figures 2, 3 and 4) have completely been reorganized according to the suggestions of Reviewer #1: the previous Figure 2 was split into the new Figures 2 and 3 and panel e of the previous supplementary Figure 3 has been moved to the main Figure 2 (panel e); panel f of the previous supplementary Figure 3 is now panel e of the new Figure 3. Because of this extensive reorganization and inherent space limitations, we propose to delete the *in vitro* 10-days stimulation data previously shown in Figures 2 and 3. This will make the new Figures 2, 3 and 4 more readable, while focusing data presentation on *ex vivo* analyses, which more closely reflect the *in vivo* situation and represent the most challenging (and, we believe, informative) part of our work. All the *in vitro* data we would like to delete from the further revised manuscript, have been re-analyzed with the above described statistics and grouped in the figure below (**Reviewer #2, Figure 1**) which has been attached to this letter for Referee's and Editor's perusal.

Reviewer #2, Figure 1: **a.** Representative examples of virus-specific CD8⁺ T cells, derived from chronically evolving and self-limited acute HCV patients and from a healthy control, stained with HLA-A2+ dexamers after 10 days of HCV- or FLU- peptide specific stimulation. **b.** Glucose uptake, measured through the incorporation of the glucose analog 2-NBDG (MFI), and total Glut1 expression (MFI) in virus-specific CD8⁺ T cells from chronically evolving and self-limited acute HCV patients and from healthy control subjects stimulated as in **a.** **c.** Percentage of mitochondrial depolarized, virus-specific CD8⁺ T cells, detected with HLA-A2+ dexamers after 10 days of culture as in **a,** determined by staining with the mitochondrial membrane potential (MMP) sensitive dye JC-1 (*left*). Dextramer-positive, virus-specific depolarized cells were quantified by subtracting the percentage of FL1^{high}/FL2^{low} cells, determined by JC-1 staining of the unstimulated samples, from the same percentage determined on the corresponding cellular subsets after a 4-hours re-stimulation. Mitochondrial superoxide levels, shown on the *right*, were determined with the MitoSOX Red dye. **d.** Phospho-ATM (Ser1981) intracellular staining of dextramer positive virus-specific CD8⁺ T cells from chronically evolving and self-limited acute HCV patients and from healthy controls after 10 days culture (with HLA-A2-restricted HCV- or FLU-peptides). Intracellular staining for total p53 (**e**), phospho-p53 (Ser15) (**f**) and phospho-p38 (Thr180) (**g**) were performed as in **d.** Data in panels **b, c** (*right-side* graphs) and in panels **d** to **g** are reported as median fluorescence intensity (MFI) values, with the actual median values indicated by horizontal lines. Differences between multiple groups have been evaluated with the Kruskal-Wallis non-parametric test. *P*-values (approximated to 3 significant digits as suggested by Reviewer #5) have been corrected for pair-wise multiple group comparisons, according to the Dunn's method.

2) The sequence analysis has only partially resolved the question about potential impact of viral escape. No sequencing data was provided for two of the 4 epitopes used in the transcriptomics analysis, though these two are known to escape more often than the other two that were studied. It also seems that none of the many other patients that were studied were sequenced.

We addressed this issue in our previous response to Reviewer #1, who was apparently satisfied by our explanation. In particular, we honestly described the results of all analyses (including sequencing of viral RNA retrieved from 5 out of 8 chronically-evolving acute patients at the T1 time-point and 4 out of 7 chronic patients at the T2 time-point) we were able to perform on leftover patient sera. No mutations were detected in the HCV NS3 CINGVCWTV and HCV NS4 VLSDFKTWL epitopes, which were the only ones, among those used to purify virus-specific CD8+ T cells, that were actually able to stimulate expansion and cytokine production by HCV-specific CD8+ T cells *in vitro* (as shown by dextramer staining and intracellular cytokine staining), thus ruling out viral escape at least for these particular epitopes and patients. As again indicated by dextramer specific staining and by the ability to stimulate cytokine production, only the above mentioned epitopes were recognized also by the remaining 6 patients who were not subjected to sequence analysis. These observations, along with the high level of sequence conservation displayed by these epitopes (substitution rates of 0.89% and 0.02% for HCV NS3 and HCV NS4, respectively), make the possible existence of escape mutations extremely unlikely, thus indicating that the transcriptional signature we detected in our cohorts of early and late chronic patients actually reflects HCV-specific CD8+ T cell exhaustion with no impact of T cell escape on it.

It should also be noted that most of our functional analyses relied on CD8+ T cell stimulation with panels of overlapping peptides spanning the entire HCV NS3 sequence (a total of 130, 15-mer overlapping peptides), thus making again extremely unrealistic the possible occurrence of escape mutations involving all the different peptides recognized within this full-coverage peptide panel.

As a general comment, we would also like to point out that a clear-cut dichotomy between exhaustion and escape -namely, the concept that CTL escape mutations within an individual CTL epitope necessarily preclude the activation of the exhaustion program because of the lack of an ongoing antigen triggering- is not so straightforward to envisage and still quite debated. Based on general knowledge derived not only from HCV but also from other viruses with a high propensity to mutate, our successful CD8+ T cell sorting with “wild-type” peptides dextramers implies that HCV-specific CD8+ cells are present and expanded in the patients’ circulation due to persistent triggering by either the original wild-type epitope (if mutations have not occurred) or by a mutated variant epitope that would only partially abrogate CD8+ T cell recognition. In either case, ongoing prototype (i.e., “wild-type-specific”) TCR triggering may start intracellular reprogramming associated with the acquisition of an exhausted phenotype, regardless of the occurrence of mutational events.

Alternatively, if one assumes that an escape mutation completely abrogates epitope recognition by the prototype TCR but can induce the expansion of “escape-specific” CD8+ T cells which can still be picked-up by our dextramers, persistent “escape-specific” CD8+ T cell triggering would still have the potential to induce progression towards exhaustion.

Finally, the possibility that the isolation of prototype epitope-specific CD8+ T cells persisting after selection of a mutated epitope is the result of the so called “original antigenic sin” mechanism should also be taken into consideration. In this case, cross-reactive recognition of the antigenic variant by prototype-specific effector/memory CD8+ T cell clones would keep them persistently stimulated, and thus still prone to exhaustion, even after selection of the variant epitope. By still poorly understood mechanisms, the “original antigenic sin” phenomenon might also interfere with the development of the naïve immune response against antigenic variants,

thus resulting in a repertoire poorly able to respond to the mutated epitope but still represented by an expanded prototype-specific T cell pool.

In summary, based on these arguments, we strongly believe that exhaustion and escape can broadly overlap and that the emergence of mutations in a given epitope does not exclude the possibility that the same epitope can still promote prototype- or variant-specific CD8+ T cell exhaustion.

3a) The analysis of the transcriptional data is still hard to follow and interpret. Notably I cannot see how the results shown in figure 1d would point specifically towards glycolysis and glucose transport as stated in the manuscript. These pathways are also not dominating the subsequent gene enrichment analysis. Overall I still fail to see how the transcriptomics data actually led towards the subsequent experiments.

The Reviewer raises concerns on transcriptome data presentation and, particularly, on how these data led to the selection of specific pathways for subsequent functional analysis. This prompted us to critically reassess transcriptome data presentation and analysis.

In the original manuscript and in the previous revision, Figure 1d displayed the main pathways identified as dysregulated in chronically-evolving acute patients. These included CD28-mediated co-stimulation and mitochondrial fatty acid beta oxidation, which comprise a number of genes involved in T cell-specific glycolysis-regulatory signaling and in important mitochondrial functions, respectively. As revealed by topology-based analysis (now better described in the revised 'Methods' section), the latter include genes such as those coding for fatty acid uptake/oxidation components (PCCA, PCCB, and ACACB) and TCA cycle enzymes such as Citrate synthase, Malate Dehydrogenase (MDH1 and MDH2) and Malonyl-CoA Decarboxylase (MLYCD). Upregulation of these genes in acute chronically evolving patients was interpreted as an indication of a mitochondrion-centered compensatory metabolic reprogramming in response to starvation stress (glucose deprivation) conditions. In accordance with this interpretation, the dysregulated metal ion Solute Carrier (SLC) pathway (also displayed in Fig. 1d) includes the glucose transporters GLUT1 and GLUT4 (SLC2A1 and SLC2A4 in Supplementary Table 1) that are downregulated in chronically-evolving with respect to self-limited acute patients, which may be consistent with a defective T cell activation. Indeed, for an appropriate T cell induction, a functional cell surface trafficking and an up-regulation of the Glut1 glucose transporter are required to support the glycolytic activity, which is essential to meet the energy demands for effector T cell function development (Jacobs et al, J Immunol. 2008 Apr 1;180(7):4476-86; Peng M et al Science. 2016 Oct 28;354(6311):481-484. Epub 2016 Sep 29; Chang CH et al, Cell. 2013 Jun 6;153(6):1239-51. doi: 10.1016/j.cell.2013.05.016). Other genes related to glucose homeostasis, also derived from topology-based analysis, were listed in Supplementary Table 1. These included genes coding for the gluconeogenesis enzymes Glucose-6-Phosphatase (G6PC and G6PC3) and Phosphoenolpyruvate Carboxykinase (PCK2), both of which are involved in the production of glycolytic intermediates under low glucose conditions (please, see the graphical abstract in Vincent EE et al., 'Mitochondrial Phosphoenolpyruvate Carboxykinase Regulates Metabolic Adaptation and Enables Glucose-Independent Tumor Growth', Mol Cell 2015, 60:195-207).

We acknowledge that the previous topology-derived pathway designations were somewhat cryptic with regard to the specific genes and functions they described. For this reason, we have revised Figure 1, where panel 1e now better illustrates the results of GSEA analysis (previously shown in Supplementary Figure 2a), which clearly identify mitochondria, DNA damage and glycolysis-related mTOR and TCR signaling components (comprised within the "Cell signaling" category) as important dysregulated pathways. Moreover, the new Supplementary Figure 3 illustrates the concordance of our results, obtained by applying GSEA to Kegg metabolic pathways in chronic-evolving vs. self-limited acute patients, with those reported for the same set of genes in the LCMV model of chronic infection (early exhausted vs. functionally efficient LCMV-specific lymphocytes from Bengsch B. et al,

Immunity 2016, 45:358-73; please see **Reviewer #2, Figure 2** for an extract of the Bengsch et al data utilized for this concordance analysis). These new results confirm an early metabolic dysregulation of HCV-specific CD8+ T cells involving both mitochondria and glycolysis-related functions, associated with an intracellular glucose deficiency ('glucose deprivation' in Supplementary Figure 2a of Bengsch et al) that is functionally supported by three independent lines of experimental evidence: namely, reduced expression of the GLUT1 transporter at both the mRNA and the protein level (Figure 2c); impaired glucose uptake (Figure 2b) and reduced glucose consumption by virus-specific CD8+ T cells (Supplementary Figure 4a).

Reviewer #2, Figure 2. Concordant dysregulation of metabolism-related genes in early LCMV- and HCV-specific exhausted CD8+ T cells. **a.** A heat-map representation of leading-edge genes driving gene set enrichment of KEGG metabolic pathways at

day 8 post-infection, derived from microarray profiling of LCMV clone 13-specific vs. LCMV Armstrong strain-specific CD8+ T cells, is shown on the *left* (modified from Fig. 1d in Bengsch et al., Immunity 2016). Concordant enrichment of up-regulated and down-regulated metabolism-related genes identified in Bengsch et al. with the transcriptome profiling of HCV-specific CD8+ T cells from chronically-evolving vs self-limited acute patients is shown on the *right*. **b.** Bar plot representation of enrichment scores (ES ; up-regulated gene set in *red*; down-regulated gene set in *blue*) derived from GSEA analysis of the indicated energy-related KEGG metabolic pathways in the early LCMV infection mouse model (extracted from Fig. 1c of Bengsch et al. 2016) (*left panel*). Glucose deprivation signature (as defined by Ho et al., 2015), derived from GSEA analysis of transcriptome data from LCMV-specific CD8+ T cells (LCMV Arm and clone 13 infection; extracted from Fig. S2a of Bengsch et al., Immunity 2016) is shown in the middle panel, where positive ES values indicate gene set enrichment in clone 13 infection. Shown in the *right* panel are the results obtained from GSEA analysis of the indicated KEGG metabolic pathways applied to transcriptome data derived from the comparison of T1/early chronically evolving with self-limited acute HCV patients, displayed with the same graphical representation utilized for the LCMV data shown in the *left-side* panel. **c.** Correlation analysis (Pearson correlation coefficient) of the expression levels of genes comprised within energy-related KEGG pathways (glycolysis and mitochondrial oxidative phosphorylation) and the glucose deprivation signature (as defined by Ho et al., 2015), derived from transcriptome analysis of mouse LCMV infection (early exhausted LCMV clone 13-specific vs. LCMV Armstrong strain-specific CD8+ T cells at day 8 post-infection; data from Doering et al, 2012, publicly available via GSE41867) and human HCV infection (T1/early HCV-specific CD8+ T cells from acute chronically-evolving vs. self-limited patients).

Transcriptome data such as the ones briefly summarized above formed the basis for our working hypothesis that CD8+ T cells from chronically evolving acute patients are in shortage of glucose, which is required to meet the energy and biosynthetic demands of T cell expansion. The latter process, centered on the pentose phosphate pathway and particularly PCK2 activation, under low glucose conditions, is known to support the production of nucleotide and amino acid precursors that are necessary for cell proliferation (please see the previously mentioned paper by Vincent EE. et al, as an example).

Other transcriptome analysis data are in accordance with, and actually support, the above working hypothesis. For example, the upregulation of several mediators/effectors of intracellular signaling downstream to CD28 co-stimulation, including the CD28, AKT1, mTOR, RICTOR and MLST8 (subunits of mTORC) genes. Since this signaling cascade is known to cause an increase in glycolytic activity and Glut1-mediated glucose uptake by T cells, we reasoned that its upregulation in CD8+ T cells from chronic evolving patients could be indicative of a compensatory response aimed at stimulating an otherwise depressed glycolytic activity. A similar scenario, with dysfunctional mitochondria, reduced GLUT1 expression and glucose uptake, and a concomitant upregulation of the CD28 signaling pathway has been described previously in the LCMV model of early CD8+ T cell exhaustion (Bengsch B. et al., Immunity 2016, 45:358-73).

3b) In addition, the newly provided gene expression lists raise some additional questions, the most obvious why PD-1, TIGIT and CTLA-4, while upregulated in chronic versus resolving patients at T1, show an inverse pattern at T2 (being higher expressed in resolved patients). This is rather different from what we know about the expression of these inhibitory receptors that are known to be most differentially regulated in exhausted (late chronic) cells versus memory (resolved infection) T cells. To me this raises serious questions about the validity of the gene expression data.

Regarding PD-1 specifically, we would like to point out the wide heterogeneity of reported protein expression levels in chronic vs. resolved patients, with a broad range of variation not only in chronic patients (Rutebemberwa et al; Golden-Mason et al; please see panels A and B of **Reviewer #2, Figure 3** below) but also in resolvers (Kasproicz et al; Missale et al; please see panels C and D of **Reviewer #2, Figure 3**). In particular, the results of Rutebemberwa et al and Golden-Mason et al clearly delineate the existence of two distinct subsets of chronic

patients, with PD-1 MFI values either superimposable to, or slightly higher than those of resolved patients. Notably, the overall difference in PD-1 expression levels measured in chronic vs. resolved subjects was found to be significant in some (Rutebemberwa et al; Golden-Mason et al; see panels A and B) but not in other (e.g., Kasprovicz et al; panel C) studies.

A. Rutebemberwa et al (J. Immunol 2008; Cox's group)

B. Golden-Mason et al. (J Virol 2007; Rosen's group)

C. Kasprovicz et al. (J Virol 2008; Lauer's group)

D. Missale et al. (Gut 2012; Ferrari's group)

Reviewer #2, Figure 3: **A.** PD-1 expression levels (MFI) in HCV-specific T cells from HCV chronic patients or spontaneous HCV resolvers and in control CD8+ T cells (taken from Figure 2 in Rutebemberwa et al; J. Immunol 2008). **B.** PD-1 expression levels (MFI expressed as percentage of pentamer-positive CD8+ T cells) in HCV chronic and resolved HCV patients (taken from Figure 1 in Golden-Mason et al; J Virol 2007). **C.** PD-1 expression levels in HCV-specific CD8+ cells from long-term chronically infected patients and resolvers (PD-1 expression is displayed as percent PD-1 positive HCV-specific CD8+ cells in the *left panel* and as MFI values in the *right panel*) (taken from Figure 1, Kasprovicz et al; J Virol 2008). **D.** Percentage of PD-1-positive HCV- FLU- and CMV-specific CD8+ cells in patients with spontaneous resolution of infection (rAH-C) or with resolution after therapy (taken from Figure 5 in Missale et al; Gut 2012).

Likewise, no significant differences are reported in the few available human studies in which PD-1 transcript levels in exhausted and functionally efficient virus-specific CD8+ T cells have been determined. For example, PD-1 and TIGIT are not listed among the differentially expressed genes that emerged from the comparison between chronically evolving and self-limited acute HCV patients (Wolski et al., Immunity 2017, 47:648-663) and between HIV progressors and controllers (Quigley et al. Nat Med 2010, 16:1147-51).

In accordance with the above described variability and lack of clear-cut differences in PD-1 expression levels between chronic and resolved infections, RT-Real-Time PCR validation experiments largely confirmed microarray gene expression data, with no significant variation of neither PD-1, nor CTLA-4 transcript abundance in chronic vs. resolved patients. Additional PCR experiments were conducted to assess the concordance of expression of other genes, as reported in our response to point 7.

As to the overall validity of our transcriptome analysis, which is again disputed by the Reviewer, in addition to the side-by-side comparison with the results on metabolism/signaling related genes obtained in the LCMV mouse model of chronic infection described in our response to point 3b, we further compared the results of our GSEA analysis with those reported for other T cell exhaustion-related pathologies. As shown in Supplementary Figure 14 (modified from previously Supplementary Figure 10), significant correlations were again detected between genes and pathways that were identified as differentially expressed in our study and in other T-cell exhaustion-related infections.

4) Regarding the seahorse analyses the authors have provided data about the size of the antigen-specific populations for both HCV- and CEF-specific cells. While it is correct that HCV-specific responses are indeed also detectable, they are also with few exceptions less than 0.5% of total CD8s, which is about one log lower than the CEF populations. In my opinion this can fully explain the absence of reactivity for HCV responses in this assay.

The possibility that different frequencies of virus-specific CD8+ T cells within the total CD8 population may affect the Seahorse results and that Seahorse analysis may not be sensitive enough at the low, virus-specific CD8+ T cell

frequencies commonly found in HCV infected patients, was also our initial concern. However, different lines of experimental evidence indicate that Seahorse analysis has indeed a sensitivity sufficient to distinguish between different metabolic conditions even at frequencies of virus-specific lymphocytes lower than 0.5% of the total number of CD8+ T cells (i.e., a frequency comparable to that measured in the majority of patient CD8+ cells populations).

First, consistent and reproducible metabolic activity differences were observed between HCV-specific and control CD8+ T cells even after data normalization to baseline values, which makes the Seahorse results independent from the relative number and activation level of the virus-specific cells present in the total CD8+ T cell population (please, see also our response to point 1 of Reviewer #1).

Second, the comparison of early self-limited with chronically evolving acute infection at the T1 time-point indicates opposite metabolic profiles for the two patients' cohorts (please see panels d vs. e in the new Figure 2 and d vs. e in the new Figure 3). It is true that most self-limited patients display higher HCV-specific CD8 T cell frequencies than chronically evolving patients, yet distinctively different metabolic profiles were detected in patients with different outcomes of infection (self-limited vs chronically evolving) but similar frequencies of HCV-specific CD8 T cells (please, see patients #1 vs. #7 in the Supplementary Fig. 5). In our opinion, this rules out the possibility that metabolic differences merely reflect differences in HCV-specific CD8+ T cell frequency.

Third, when similar frequencies of IFN- γ producing CD8+ cells were detected after overnight stimulation of individual patients lymphocytes with control and HCV-specific peptides, a better metabolic performance was again detected in control compared to HCV peptide stimulated cells (please see patient #1 in the Supplementary Fig. 5). In our opinion, this represents an additional indication of a real qualitative metabolic difference, totally unrelated to the relative size of the virus-specific CD8+ T cell pool within the overall CD8 T cell population.

Fourth, we analyzed the metabolic profile of FLU matrix 58-66-specific CD8+ T cells detected by dextramer staining in a healthy subject at a frequency (0.44%) comparable to that commonly found in our acute HCV patients. To further assess Seahorse sensitivity, we also diluted the stimulated cells with an equal number of unstimulated cells, in order to lower the frequency of stimulated FLU-specific CD8+ T cells to 0.22%. As apparent in the plots shown below for Reviewer's and Editor's evaluation (**Reviewer #2, Figure 4**), Seahorse analysis has enough sensitivity to reliably detect metabolic differences between stimulated (*blue* and *light blue*) and unstimulated (*green*) cells down to frequencies lower than 0.5% of total CD8+ T cells (i.e., frequencies comparable to, or lower than, those of HCV-specific cells commonly found in total CD8 cells from our patients populations).

Reviewer #2, Figure 4: Circulating FLU-specific CD8 T cells were detected in a healthy subject at a frequency of 0.44% of the total CD8+ T cells by staining with the HLA-A2 dextramer containing the matrix-epitope 58-66 (*left panel*). They were used either undiluted or diluted with an equal number of unstimulated CD8 T cells, in order to measure ECAR and OCR at FLU-specific CD8 T cell frequencies of 0.44% and 0.22% (i.e., T cell frequencies comparable to, or lower than, those detected in our HCV patients cohorts). The graphs on the *right* show the metabolic flux profiles of purified CD8+ T cells either unstimulated (*green*) or stimulated overnight with the matrix 58-66 peptide at a frequency of 0.44% (*dark blue*) or 0.22% (*light blue*). Data are the mean \pm SEM of two biological replicates.

5) I remain unconvinced by most of the data in figure 3. If the appropriate unpaired test is used, will the differences remain significant? In addition, while the authors are correct that phosphoflow experiments have been directly compared between different outcomes in LCMV using inbred mice, this seems more questionable to do in a human cohort, given the high genetic variability. For this reason I still think one needs the reference of unstimulated samples for each patient in these assays.

As mentioned above in our response to point #1, the Kruskal Wallis one-way analysis of variance test with correction for multiple comparisons by the Dunn's Multiple Comparison test has been used to reanalyze the data and the results, which are largely consistent with those produced by the previous analysis, are displayed in the revised Figures 2, 3, 4 and in the Supplementary Figures 9 and 10.

Regarding phosphorylation assays, we agree with the Reviewer on the conceivably higher genetic variability of human patients. To address the Reviewer's concern, unstimulated and anti-CD3/anti-CD28 stimulated cells have been studied in additional patients and the results are illustrated in the new Figure 4. These new data confirm that the expression of phosphorylated proteins (ATM, p53 and p38) is greater in HCV-specific CD8+ cells from chronically evolving acute patients compared to HCV-specific CD8+ cells from self-limited patients and to FLU-specific CD8 cells from healthy subjects, as reported in the original manuscript where only stimulated samples were analyzed.

6) The new data in figure S4 are not useful to compare with the results in figure 4 as they are shown in a completely different format. Figure 4 shows fold changes per patient, figure S4 aggregate results for all samples.

We thank the Reviewer for this comment. Accordingly, the data in the new Figure 5 (previously Figure 4), Figures S4 and S5 (derived and expanded from the previous Figure S4 in order to incorporate all the new data requested by Reviewer #1) are now presented in the same graphical format.

7) The data in figure S5b for PD-1 and CTLA-4 directly contradict the microarray results in table 2. In s5b expression is higher in chronic patients and the opposite is shown in the microarray data, as already mentioned under 3)

Regarding this remark, partly addressed also in our response to point 3b, we would like to recall that no significant variation in PD-1 expression levels was detected by RT-PCR (fold-change ~1), thus confirming the microarray results as to the lack of any significant difference in PD-1 expression levels between chronic and resolved patients. The fold-change for CTLA-4 measured by RT-PCR was slightly higher than that derived from microarray analysis but again the increase in chronic compared to resolved patients was not significant. Although apparently in contrast with the general notion of high PD-1 levels in chronic patients, these data are in line with other results obtained at the protein level (please see, for example, Kasprovicz et al. J of Virol 2008) showing that PD-1 expression can remain elevated also in the resolution phase of acute hepatitis C and that “most cells continue to express PD-1 in resolved and chronic stages of infection”, as stated by the authors on page ... of their manuscript

To further assess the reliability of our microarray data, we performed additional Real Time qPCRs on nine different subjects targeting four genes related to mitochondrial and proteasomal functions. As shown in the figure below (**Reviewer #2, Figure 5**), attached to this letter for Reviewer’s evaluation, a statistically significant correlation was detected with the Spearman’s rank-order correlation test between expression levels determined by microarray (\log_2) and qPCR ($\log_2 2^{-\Delta Ct}$).

Reviewer #2, Figure 5: Validation of a subset of dysregulated (down-regulated in chronic vs. resolved) mitochondrion (NDUFA6, UQCRC1 and TIMM10) and proteasome (PSMD4) related genes in CD8 cell from nine different subjects. Gene expression levels derived from microarray (\log_2) and qPCR ($\log_2 2^{-\Delta Ct}$) analysis are shown as Spearman’s correlation plots.

8) A major new concern comes from the new supplemental data in figures S4/8/9. Here the ICS data show higher percentages for untreated cells of HCV-specific CD8 T cells in chronic compared to resolver patients at T2 directly *ex vivo* and even more stunningly post *in vitro* expansion. This is puzzling, as all published literature finds much stronger T cell responses directly *ex vivo* and also better expansion of T cells in resolved versus chronic infection. In addition, I am somewhat surprised by the fact that many somewhat modest looking differences in chronic patients are all highly significant, but none of the differences the graphs seem to indicate for resolvers. Why where that data not shown in the same way as in figure 4e/f.

We acknowledge that the different modes of data presentation utilized in the previous Supplementary Figures S8 and S9 (bar plots with mean \pm SEM and whisker plots with median values, respectively) made it difficult to evaluate the percentage of cytokine production in chronic vs. resolved patients. Furthermore, the y-axes in Figure S9a had two different scales for *ex vivo* (*left-side*) and *in vitro* culture (*right-side*) experiments, which further complicated the comparison of the data represented in the two different figures. We apologize for this heterogeneity in data presentation, which has now been corrected. In particular, panel a of the previous supplementary Figure S8 (now Supplementary Figure 9) has been deleted, because the same data are reported in a dot-graph format in panels d to l of Figure 7 (previously Figure 6). However, the data shown in the previous Supplementary Figure S8a are attached below in a whisker format for Reviewer's evaluation (please see **Reviewer #2, Figure 6**).

Reviewer #2, Figure 6: Impact of HMT inhibitors on anti-viral cytokine production by T1/early and T2/late exhausted HCV-specific CD8+ T cells. PBMCs from chronically evolving (T1/early) or chronic (T2/late) HCV patients were stimulated for 40 h or 10 days of culture with HCV-NS3 peptides, in the presence or absence of: the EZH2 HMT inhibitors GSK126 (GSK) and EPZ005687 (EPZ) (red bars); the EHMT2/ G9a inhibitors UNC0638 (UNC) and BIX01294 (BIX) (green bars); the p53 inhibitor pifithrin- α (blue bars). Cells were co-stained for CD8 and IFN- γ , IL2 and TNF α . Results are displayed as Tukey whisker plots with median values of the percentage of cytokine-positive CD8+ T cells shown inside each bar by horizontal lines. Statistical analysis (treated vs. untreated) was performed with the two-tailed Wilcoxon-matched-paired test.

Further to this point, fold-changes in supplementary Figure 10 (previously Figure S9) are now represented as dots for consistency with the data presentation utilized in the new Figure 7. Since this new format does not allow to

visualize untreated samples in the different study groups, we have assembled a figure, attached below for Referee’s evaluation, showing the values measured in stimulated and untreated cells from all patient categories. As shown below (**Reviewer #2, Figure 7**), cytokine production in our study was generally higher in self-limited compared to chronically evolving acute infections at the T1 time-point and in resolved compared to chronic patients at the T2 time-point.

Reviewer #2, Figure 7: PBMCs from chronically evolving and self-limited (T1/early) or chronic and resolved (T2/late) HCV patients were stimulated for 40 h or 10 days of culture with HCV-NS3 peptides and co-stained for CD8 and IFN-γ, IL2 and TNFα. Results are displayed as Tukey whisker plots with median values of the percentage of cytokine positive CD8+ T cells indicated within each bar. Statistics with Mann–Whitney test.

In the evaluation of these data, the Reviewer should kindly consider the following points.

- A. In the comparison of our *ex-vivo* data with those presented in other studies, it should be noted that our *ex-vivo* analysis is based on a 40-hours PBMC stimulation, which should perhaps be better defined as ‘short-term stimulation’ rather than ‘*ex-vivo* analysis’. This modified experimental set-up has been chosen to increase the likelihood of inducing the production of measurable cytokine levels in response to the

different treatments also in CD8+ T cells from chronic patients. In fact, as correctly stated by the Reviewer, the latter cells often fail to respond and are generally negative after a standard 4-hour stimulation.

- B. Six new biological replicates have been added in order to increase the number of resolved (T2) patients which was originally quite low (6 patients).
- C. Chronic and resolved patients have previously been shown to display widely variable levels of cytokine production and cytotoxic activity, especially upon long-term (7-10 days) *in vitro* stimulation, and published data are not entirely concordant regarding the detection of significant differences between chronic patients and spontaneous resolvers. Please see, for example, the results reported in the study by Chang et al, Hepatology 2001 (**Reviewer #2, Figure 8, panel A**) showing a better IFN- γ production and cytotoxic activity by HCV-specific CD8+ T cells from chronic patients compared to resolvers and those reported in the study by Missale et al, Gut 2012 (**Reviewer #2, Figure 8, panel B**) to see how variable antiviral responses can be in spontaneous as well as in pharmacologically treated resolved infections.

Reviewer #2, Figure 8: Cytotoxic activity and cytokine production by HCV-specific CD8 T cells. **A.** Specific cytotoxicity of virus-specific CD8+ T cell lines measured at an E/T ratio of 50 and plotted against IFN- γ production. HCV-specific T cell functions were determined after *in vitro* expansion and staining for intracellular IFN- γ after a 5-hour incubation with and without specific NS3-HCV peptides (taken from Figure 5 in Chang et al, Hepatology 2001). **B.** Cytokine production by virus-specific CD8 cells of different specificities derived from spontaneous resolvers (*self-limited*) and from pharmacologically treated acute (*rAH-C*) or chronic (*rCHC*) patients (taken from Figure 2 in Missale et al, Gut 2012).

9) It would be critical to see more raw data and more detailed results for the assays shown in figures 6 and 7. In the chronic stage of HCV infection the direct ex vivo ICS assay is typically negative for interferon-gamma and even more so for IL-2 in practically all patients. What is shown in figure 6c is feasible in acute patients, but is decidedly

not a representative results for this assay in chronic (T2) patients. How many PBMC were studied and how big were the clouds of functional cells?

As mentioned in our response to the previous point, anti-viral responses in chronic patients can be widely variable and the dot-plots reported in panel C of the previous Figure 6 refer to a chronic patient with one of the best responses.

As requested, additional raw data have been added in Supplementary Fig. 11. *Ex-vivo* and 10-days culture experiments were always started with PBMC numbers ranging from 400,000 to 600,000, and depending on cell growth, 100,000 to 300,000 lymphocytes were usually recovered after intracellular cytokine staining. As mentioned above (point A), *ex-vivo* stimulation was prolonged to 40 hours.

We also carefully re-checked all clinical data to further verify chronic infection diagnosis, which was confirmed in all enrolled patients.

>Reviewer #3 (Remarks to the Author):

The authors addressed my previous concerns and appear to have addressed most or all the concerns of the other reviewers. Thus it is acceptable for publication.

>Reviewer #4 (Remarks to the Author)

None

We are grateful to Reviewers #3 and #4 for their positive assessment of our work.

Reviewers' comments:

Reviewer #5 (Remarks to the Author):

1. General comments.

This is not an easy paper to review from a data analytic viewpoint. There are a huge number of analyses presented, and, in general I do not think these analyses are described very clearly. The underlying essence gets completely lost in a mass of experimental technical detail.

In general, the sample sizes are small and given the likely experimental variation in the variables being measured it seems unlikely that the findings are statistically robust.

Following-up to the first remark of the Reviewer, we tried to improve the clarity of data presentation by reorganizing the content and format of many figures, especially new Figures 1-4, 6, 8, S4, S9 and S10.

Regarding the 'small sample size' specifically, we agree with the Reviewer's comment but would like to point out that recruitment of acute HCV patients with different outcomes of infection is nowadays extremely difficult because present guidelines recommend early pharmacological therapy of acute HCV patients prior to the definition of the infection outcome. For this reason, the recruitment of new patients with chronic evolution of infection has become extremely difficult, if not impossible. Additional limitations to the recruitment of larger patients' cohorts are due to:

- i) the need of selecting HLA-A2 positive patients for some specific studies, such as metabolic and epigenetic assays which must be performed on virus-specific dextramer positive CD8+ T cells;
- ii) the large numbers of PBMC required by the experimental design of our study, because each assay (requiring at least 4 million of PBMC) must be performed separately due the different staining procedures utilized for individual measurements (e.g. glucose uptake, GLUT1 expression, JC1, ROS and proteasome stainings).

Despite these strong limitations, we tried to expand as much as possible the number of patients employed for the different assays in order to improve the reliability of the data and the robustness of statistical analysis. Overall, the total number of tested subjects has been increased from 207 in the original manuscript to 243 in the present version. Given the above explained limitations, this is an extremely large number of patients, certainly much larger than those employed in previously published studies based on complex laboratory analyses applied to similar cohorts of patients.

In addition, as also suggested by the Reviewer, we re-assessed the appropriateness and stringency of most statistical analyses using different statistical methods. In particular, for the comparison of different patient groups, we applied a heterogeneity test, namely the non-parametric Kruskal Wallis one-way analysis of variance test with correction for multiple comparisons by the Dunn's Multiple Comparison test, according to the Reviewer's indications. To assess possible differences in responses across multiple treatments, we applied the non-parametric Friedman test. Even with this more stringent statistical analysis, most (albeit not all) results reported in the previous version of our manuscript have been confirmed.

Sample size was also considered in transcriptome analysis. In fact, when performing GSEA, the FDR (False Discovery Rate) was estimated using the option "gene set permutation" rather than "phenotype permutation", which is recommended when the sample size is lower than 7, in order to obtain a more stringent assessment of significance. An FDR cutoff of 0.1, rather than 0.25 was also chosen accordingly.

Many P-values are reported to 4 significant figures. This is not appropriate precision.

According to the reviewer's comment, p-values with 4 significant figures have been corrected.

2. The description of the statistical methods is very limited. Reviewer #2 raised this issue (point #3) but clearly it has not been adequately addressed. For example in first paragraph of p8 a topology-based analysis is mentioned, but there is almost no description in the statistical methods of what that actually means. I cannot understand how this has generated a q-value.

We apologize for the insufficiently detailed description of transcriptome data analysis. This was based on the published methods, referenced in the 'Methods' section, that were applied to our data using default parameters unless otherwise specified. However, following-up to the Reviewer's request, a more detailed description of the methods we used for data-mining has been incorporated in the revised manuscript (please see the new 'Methods' section).

Regarding 'topology-based' analysis specifically - a data-mining method originally devised by one of our co-authors (CR; Martini et al, Nucleic Acids Res 2013; Sales et al Nucleic Acids Res 2013)- this relies on pathway structure, extracted and analyzed with the Graphite Web tool (<https://doi.org/10.1093/nar/gkt386>), to compare groups of genes in terms of means and covariance, using biological information derived from the KEGG and Reactome databases. As explained in detail in the revised 'Methods' section, Topological Pathway Analysis, as applied to gene expression data, uses graphical theory and pathway structure to compare different experimental groups in terms of means and covariance among genes. Using biologically driven rules, the resulting pathway structure is converted into a network graph, where nodes and edges represent genes and gene connections, respectively. Under the assumption of a Gaussian distribution and taking into account the constraints imposed by the graph structure, mean and covariance matrices are then estimated and comparatively assessed using the likelihood ratio test. In order to identify the key portions of the dysregulated pathways, graphs are decomposed into cliques (small connected components) and the same statistical tests applied to the means and the covariance matrices are used to select the most significant chain of cliques within each pathway. The resulting p -values are then corrected for multiple testing with the Benjamini-Hochberg method, thereby generating a q -value.

The reason why we used topological analysis for the initial data-mining, is that it offers some significant advantages compared to more classical, univariate inferential methods: i) it uses a multivariate approach, thus shifting the focus from a single gene to a gene-set (pathway or group of cliques); ii) it takes into account the structure of the relationship among genes, thus increasing the overall power of the analysis; and iii) it highlights differences not only in the mean but also in the covariance structure. In fact, a difference in mean expression levels does not necessarily result in a change of the interaction strength among genes. In such a case, we would have pathways with significantly altered mean expression levels but unaltered biological interactions. On the contrary, if transcript abundance ratios are altered, we would have a significant alteration not only of their mean expression levels, but also of the strength of their connections, thus pointing to dysregulated pathways with an altered functionality.

Similarly, the description of the gene-set enrichment analysis is inadequate

We apologize for the rather narrow description of Gene Set Enrichment Analysis (GSEA), which is a widely used data-mining method, and for that reason we extensively referred to previous work describing this type of analysis. However, following up to the Reviewer's criticism, a more detailed description is now provided in the revised

'Methods' section. Moreover, we would like to point out that GSEA, a non-parametric method based on the Kolmogorov–Smirnov (K–S) test: i) ranks genes based on a statistical evaluation (e.g., t-test, signal to noise) of their differential expression in two different experimental conditions; ii) it is typically run against a large number of predefined gene-sets (e.g., the Molecular Signatures Database in the case of our analysis); and iii) it can be applied to transcriptome analysis data without a predetermined fold-change cut-off.

In particular, GSEA allows to determine whether a pre-defined set of genes shows statistically significant differences in expression between two biological conditions (such as the comparison between the different infection outcomes associated to the T1/early and T2/late time-points). To this end, genes comprised within the transcriptome data set are ranked according to the correlation between their expression levels and the specific comparison under examination (e.g. T1/early chronic vs T1/early self-limited) using any suitable metric. For each gene comprised within the ranked list, a running-sum statistics is augmented whenever a gene of the query set is encountered (or decreased otherwise) by assigning a score to each gene. Finally, the enriched score (ES) is computed as the maximum deviation from zero of the running-sum statistics. Statistical significance of the ES is then estimated with the use of a permutation test procedure.

3. Reviewer #1, point #5.

The relevant section is p7, para 2. The first sentence states that “Principal component analysis of ANOVA-filtered expression data”. This analysis is not described in the statistical methods and it is unclear to me what filtering has been applied. There are four patient groups with 5, 8, 4 and 7 patients. Parametric statistical methods are unlikely to be appropriate with such small numbers.

As stated in the 'Methods' section of the previous version of our manuscript: “The GeneSpring GX v11.5 software package (Agilent Technologies) was used for quality control checks, data normalization by the quantile method and initial microarray data analysis. Probes detectable in at least two-thirds of replicates for each condition were retained for further analysis.” This is the filtering method we used to select reliable probes (among those detected at above-background levels) for subsequent analysis. It was also mentioned that “ANOVA with Benjamini–Hochberg correction for multiple testing ($FDR \leq 0.05$) was used to select differentially expressed genes among all conditions, that were visualized as hierarchical clustering and used for principal component analysis (PCA)”.

To address the Reviewer's concern about an insufficient description of this analysis, we further detailed PCA in the revised 'Methods' section and added the above technical specifications to the legend of Figure 1.

Regarding the appropriateness of parametric statistical methods, we would like to point out the following considerations: i) ANOVA was only used to filter-out non-variable probes, thus in this case the assumption of a Gaussian distribution is irrelevant to the test; ii) as mentioned above, GSEA is based on Kolmogorov-Smirnov statistics, which relies on a permutational approach and therefore does not involve any distributional assumption; iii) Topological pathway analysis also relies on a permutational, rather than a distributional approach.

Therefore, all the main transcriptomic analyses were performed using non-parametric statistical methods.

4. Figure 2/3. When comparing three groups it is inappropriate to compare group 1 with group 2 and then to compare group 1 with group 3. A general heterogeneity test for between group differences should be applied.

We thank the Reviewer for this suggestion, which has been implemented with the use of a heterogeneity test for multiple group comparisons, that has been applied to the data in Figures 2, 3 and 4.

We acknowledge the inappropriate use of the Wilcoxon paired sample test (please, see also our response to point # 1 of reviewer 2) and thank the Reviewer for this suggestion, which has been implemented with the use of a heterogeneity test for multiple group comparisons, applied to the data in Figures 2, 3 and 4. As mentioned above, in the revised manuscript, the latter test has been used in place of the Wilcoxon paired sample test.

Of all the P-values reported in these figures $P=0.0039$ on seven occasions. This seems unlikely - though possible - with a rank based test (i.e. they would need to have observed the same ranking of different outcomes between the two groups on seven occasions).

As suggested by the Reviewer, in the last revision of the manuscript the heterogeneity test for multiple group comparisons has been used in place of the Wilcoxon paired sample test.

Relative to the specific question raised by the Reviewer, who finds unlikely (though possible) that we obtained the same p-value (0.0039) on seven occasions, we agree with this concern; we note, however, that all p-values equal to 0.0039 were derived from Wilcoxon paired tests applied to nine data pairs. This test calculates differences by pair according to two options (i.e., positive or negative differences). Having nine pairs, the possible number of different sums is $2^9 = 512$, where 512 is the number of different values of the sums of the ranks. The most extreme case is one in which all the differences (and thus all the ranks) are positive or negative, which would correspond to an approximated p-value of 0.0039 (i.e., $2/512$). This means that for all comparisons yielding such a p-value, all the values of the sums of the ranks of a group are larger (or smaller) than the corresponding values of the other group. Along this view, it seems plausible (or at least possible) that in seven comparisons, group #1 was always greater than group #2 or vice versa.

5. Summary. I would not quibble very much with the individual statistical tests that have been applied, but the description of each of these and the inferences drawn from the findings of each needs to be clearer.

We really hope that with the extended description of the methods utilized for transcriptome data analysis, together with the reconsideration of statistical analysis along the lines indicated by the Reviewer and further clarifications added to the revised text, we have satisfactorily addressed the reviewer's concerns.

REVIEWERS' COMMENTS:

Reviewer #1 (Remarks to the Author):

The main issues raised in the previous review were very well addressed by the authors. Nevertheless it would be important to address some limitations of the study in the abstract and discussion.

Abstract:

The abstract should be written more precisely as it currently gives rise to misinterpretations.

1)

For example, the authors use wording of “early” and “late” exhaustion to rather describe the analysis of HCV-specific CD8 T cells during early and late infection. This may be confusing to some readers that would link use of “early” and “late” wording with regards to exhaustion rather to aspects related to the differentiation program of exhausted T cells. Typically such analysis are performed in experimental settings that are tightly controlled, such as in mouse models with invariant TCRs. In this study, as in many human analyses, it is not clear that the populations of virus-specific CD8 T cells analyzed early and late share the same origin. This would be suggested if they all share the same clonality, i.e. TCR sequence which was not analyzed. Instead I suggest to refer with “early” and “late” to the timing post viral infection.

2)

The authors write in the abstract:

“Here we show that early exhausted CD8+ T cells are marked by a general transcriptional upregulation associated with impaired glycolytic and mitochondrial functions which are causally linked to an abnormally enhanced ATM and p53 signaling.”

It has to be noted that no causal relationship of the transcriptional and metabolic changes with ATM and p53 signaling in early (i.e. early timepoint after infection) exhausted T cells are shown in this study. The authors even mention the lack of (formal) evidence on p23. I would also be cautious about subjective wording like “abnormally” when levels of ATM and p53 signaling are clearly high but do not show orders of magnitude of difference compared to controls.

What the authors rather show is evidence that in early chronic HCV infection, exhausted virus-specific CD8+ T cells are marked by an upregulation of transcription associated with impaired glycolytic and mitochondrial functions that are linked to enhanced ATM and p53 signaling.

3)

The authors write in the abstract:

“Late exhaustion is instead characterized by transition to a broad gene downregulation associated with a wide metabolic and anti-viral function impairment, which can be rescued by histone methyltransferase inhibitors.”

Similarly to 1) and 2) this might be more precisely reflected by wording such as

“After evolution to chronic infection, exhaustion of HCV-specific T cell responses is instead characterized by broad gene downregulation associated with a wide metabolic and anti-viral function impairment, which can be augmented by histone methyltransferase inhibitors .”

Discussion

Weaknesses of the study inherent to the experimental protocols should be discussed.

For example, pre-stimulation of T cells in the experimental protocols can cause selective loss of highly exhausted T cells prone to apoptosis from the analysis. It can also cause metabolic reprogramming. The use of CD3 and CD28 stimulation may act differently on cells with variable CD28 expression, benefitting CD28+ cells etc. Seahorse analysis between patient samples is hard to cross-compare due to potential batch effects. The authors have tried to address some of the issues inherent to their approach using controls but these caveats need to be pointed out.

Reviewer #2 (Remarks to the Author):

I do not have any more questions or suggestions for the authors.

Reviewer #5 (Remarks to the Author):

The main statistical issues that I raised have now been addressed. The manuscript is still a very complex and difficult read.

ABSTRACT

The abstract should be written more precisely as it currently gives rise to misinterpretations.

1) For example, the authors use wording of “early” and “late” exhaustion to rather describe the analysis of HCV-specific CD8 T cells during early and late infection. This may be confusing to some readers that would link use of “early” and “late” wording with regards to exhaustion rather to aspects related to the differentiation program of exhausted T cells. Typically, such analysis are performed in experimental settings that are tightly controlled, such as in mouse models with invariant TCRs. In this study, as in many human analyses, it is not clear that the populations of virus-specific CD8 T cells analyzed early and late share the same origin. This would be suggested if they all share the same clonality, i.e. TCR sequence which was not analyzed. Instead I suggest to refer with “early” and “late” to the timing post viral infection.

2) The authors write in the abstract:

“Here we show that early exhausted CD8+ T cells are marked by a general transcriptional upregulation associated with impaired glycolytic and mitochondrial functions which are causally linked to an abnormally enhanced ATM and p53 signaling.”

It has to be noted that no causal relationship of the transcriptional and metabolic changes with ATM and p53 signaling in early (i.e. early timepoint after infection) exhausted T cells are shown in this study. The authors even mention the lack of (formal) evidence on p23. I would also be cautious about subjective wording like “abnormally” when levels of ATM and p53 signaling are clearly high but do not show orders of magnitude of difference compared to controls.

What the authors rather show is evidence that in early chronic HCV infection, exhausted virus-specific CD8+ T cells are marked by an upregulation of transcription associated with impaired glycolytic and mitochondrial functions that are linked to enhanced ATM and p53 signaling.

3) The authors write in the abstract:

“Late exhaustion is instead characterized by transition to a broad gene downregulation associated with a wide metabolic and anti-viral function impairment, which can be rescued by histone methyltransferase inhibitors.”

Similarly to 1) and 2) this might be more precisely reflected by wording such as

“After evolution to chronic infection, exhaustion of HCV-specific T cell responses is instead characterized by broad gene downregulation associated with a wide metabolic and anti-viral function impairment, which can be augmented by histone methyltransferase inhibitors.”

According to the Reviewer's requests, we made all changes to the abstract she/he suggested in points 1 and 2. Regarding point #3, the sentence pointed out as not sufficiently clear by the Reviewer has been rephrased as suggested, except for the word, "augmented", which was not included in the reworded sentence because it would totally alter the main message of our study. Indeed, far from 'augmenting' exhaustion, histone methyltransferase inhibitors actually restore the T cell function.

DISCUSSION

Weaknesses of the study inherent to the experimental protocols should be discussed.

For example, pre-stimulation of T cells in the experimental protocols can cause selective loss of highly exhausted T cells prone to apoptosis from the analysis. It can also cause metabolic reprogramming. The use of CD3 and CD28 stimulation may act differently on cells with variable CD28 expression, benefitting CD28+ cells etc. Seahorse analysis between patient samples is hard to cross-compare due to potential batch effects. The authors have tried to address some of the issues inherent to their approach using controls but these caveats need to be pointed out.

- The possible pro-apoptotic effect of CD3/CD28 stimulation on highly exhausted (and prone to apoptotic death) T cells has been mentioned in the Discussion section on page 22 (sentence added to the text: "Moreover, the actual amount of pre-apoptotic cells might have been further lowered by overnight stimulation of dextramer-positive CD8 T cells with anti-CD3 and anti-CD28 antibodies, a treatment that is expected to drive apoptosis-committed T cells toward apoptotic death").
- The risk of metabolic reprogramming associated with CD3/CD28 stimulation as well as the cautionary countermeasure we have set-up and adopted in our study are now mentioned on page 20 of the revised 'Discussion'. Indeed, given the unfeasibility of using unstimulated HCV-specific CD8+ T cells directly ex vivo, because of their extremely low frequency, we made a great effort to set-up highly sensitive T cell functional assays that could be successfully applied to short-term (overnight) rather than long-term (typically, 7-10 days) cultured CD8+ T cells, as commonly used in human studies of virus-specific T cells. The following sentence has been added to the revised 'Discussion': "Short-term culture conditions in the presence of either HCV-specific peptides or anti-CD23/anti-CD28 antibodies were set-up in order to circumvent the strong limitations inherent to the extremely low frequency of HCV-specific CD8+ T cells. These conditions, which differ from the much more prolonged stimulation commonly applied in human studies of virus-specific CD8+ T cells, were also aimed to limit possible artifactual metabolic alterations associated with long-term culture and in vitro expansion").

- Further details on Seahorse analysis, and particularly the fact that the most relevant comparisons were performed between different stimuli applied to CD8+ T cells from each individual patient and were always analyzed in the same experimental session, are now provided at page 38. As mentioned in the revised manuscript, we believe that this kind of experimental design, together with the internal normalization of maximal to basal OCR and ECAR values that was applied to each sample, makes extremely unlikely the possible occurrence of a 'batch effect'. This view is strongly supported by the identical patterns of metabolic responses we detected within each cohort of self-limited or chronically evolving acute patients. The following sentence has been added to the revised text: "Seahorse experiments were performed on total CD8+ T cells purified with the CD8+ T Cell Isolation Kit (Miltenyi Biotec). Multiple stimuli (i.e, Flu/EBV/CMV and NS3-specific HCV peptides, plus an unstimulated control) were applied to CD8+ T cells from each individual patient and each culture condition was analyzed in the same experimental session in order to limit possible batch effects. These were further minimized through the internal normalization of maximal to basal OCR and ECAR values that was applied to each sample".